**A record of Neogene seawater $\delta^{11}$B reconstructed from paired $\delta^{11}$B**
**analyses on benthic and planktic foraminifera.**
Greenop Rosanna[1,2*], Hain, Mathis P.[1], Sosdian, Sindia M.[3], Oliver, Kevin I.C.[1],
Goodwin, Philip[1], Chalk, Thomas B.[1,4], Lear, Caroline H.[3], Wilson, Paul A.[1], Foster,
Gavin L.[1],
[*]Corresponding author
[1] *Ocean and Earth Science, National Oceanography Centre Southampton, University*
*of Southampton, Waterfront Campus, European Way, Southampton SO14 3ZH, UK*
[2]*School of Geography & Geosciences, Irvine Building, University of St Andrews,*
*North Street, St Andrews, KY16 9AL, UK*
[3]*School of Earth & Ocean Sciences, Cardiff University, Cardiff, CF10 3AT, UK*
[4] *Department of Physical Oceanography, Woods Hole Oceanographic Institution,*
*Woods Hole, Massachusetts, USA*
**Abstract:**
The boron isotope composition ($\delta^{11}$B) of foraminiferal calcite, which reflects
seawater pH, is a well-established proxy for reconstructing past seawater carbonate
chemistry and, in the case of planktic foraminifera, past atmospheric $CO_2$. However,
to translate $\delta^{11}$B measurements determined in calcareous fossils into pH we need to
know the boron isotopic composition of the seawater in which they grew ($\delta^{11}$B$_{sw}$).
While a number of $\delta^{11}$B$_{sw}$ reconstructions exist, more work is needed to build
confidence in our knowledge of this important parameter. Here we present a new
Neogene $\delta^{11}$B$_{sw}$ record based on the $\delta^{11}$B difference between paired measurements of
planktic and benthic foraminifera and an estimate of the coeval water column pH
gradient derived from planktic/benthic $\delta^{13}$C data.  To underscore this approach we
present extensive tests using the CYCLOPS and GENIE carbon cycle models to
demonstrate that the planktic/benthic $\Delta$pH/$\Delta\delta^{13}$C relationship is relatively insensitive
to ocean and carbon cycle changes. In keeping with previously published records, our
reconstruction suggests that $\delta^{11}B_{sw}$ was ~ 37.5 ‰ during the early and middle
Miocene and rapidly increased from ~12 to 5 Ma to reach a plateau near the modern
value of 39.61 ‰. A similar pattern of change is evident in the seawater composition
of the Mg, Li and Ca stable isotope systems. Concurrent shifts in the seawater
isotopic composition of all four of these elements during the late Miocene are
suggestive of a common forcing mechanism. Based on the observed direction of
change we hypothesise that an increase in secondary mineral formation during
continental weathering may have affected the isotopic composition of the riverine
input to the ocean since ~12-15 Ma.

## 1. Introduction

Key to determining the relationship between $CO_2$ and climate in the geological past is
the calculation of reliable estimates of absolute $CO_2$ through time. In recent years the
boron isotope composition ($\delta^{11}B$) of foraminiferal calcite has become a high-profile
tool for reconstructing $CO_2$ beyond the last 800 kyrs and throughout the Cenozoic Era
(Foster, 2008; Hönisch et al., 2009; Pearson et al., 2009; Bartoli et al., 2009; Foster et
al., 2012; Badger et al., 2013; Henehan et al., 2013; Greenop et al., 2014; Martínez-
Botí, et al., 2015a). Yet long-term change in the boron isotope composition of
seawater ($\delta^{11}B_{sw}$) is currently poorly constrained and represents a major source of the
uncertainty associated with $\delta^{11}B$-determined $CO_2$ estimates (e.g. Pearson et al., 2009).
In the modern ocean boron is a conservative element with a spatially invariant
isotope ratio (39.61‰; Foster et al., 2010), but this value is subject to change through
geological time. The residence time of boron in the ocean is estimated to lie between
11 and 17 Myrs (Lemarchand et al., 2000). Therefore we can expect the uncertainty
associated with $\delta^{11}B_{sw}$ to be an important factor in $CO_2$ estimates beyond the late
Pliocene (~ 4-5 Ma, Palmer et al., 1998; Lemarchand et al., 2000; Pearson et al.,
2009; Foster et al., 2012; Anagnostou et al. 2016).
The ocean boron budget and its isotopic composition are controlled by a number of
inputs and outputs (Fig. 1). However, because the magnitude of the boron fluxes
between land, the ocean and the atmosphere in the modern are still poorly
understood, the residence time and changes in both concentration ($[B]_{sw}$) and isotopic
composition ($\delta^{11}B_{sw}$) through time remain uncertain. The main inputs of B into the
ocean are silicate weathering, and to a lesser extent evaporite and carbonate
weathering, delivered to the ocean by rivers (Lemarchand et al., 2000; Rose et al.,
2000; Lemarchand and Gaillardet, 2006), hydrothermal vents (You et al., 1993) and
fluid expelled from accretionary prisms (Smith et al., 1995).  The major loss terms
are low temperature oceanic crust alteration (Smith et al., 1995), adsorption onto
sediments (Spivack and Edmond, 1987) and co-precipitation into carbonates
(Hemming and Hanson, 1992). In the case of all three outputs the light $^{10}B$ isotope is
preferentially removed relative to $^{11}B$, such that the seawater $^{11}B/^{10}B$ ratio ( $\delta^{11}B_{sw}$,
39.61‰) is significantly greater than that of the cumulative inputs ($\delta^{11}B$ of ~10.4‰;
Lemarchand et al., 2000).  Our understanding of the modern boron fluxes outlined
above, and illustrated in Fig. 1, implies a significant imbalance between inputs and
outputs and consequently the poorly constrained ocean-atmosphere boron fluxes may
also be an important part of the ocean's modern boron mass balance  (Park and
Schlesinger, 2002). Here, however, we follow Lemarchand et al., (2000) in assuming
that atmospheric fluxes are unlikely to have varied significantly on geological
timescales and therefore will not be discussed further in reference to the Neogene
$\delta^{11}B_{sw}$ record we present.

Unlike many other isotopic systems (e.g. $\delta^{7}Li_{sw}$, $\delta^{26}Mg_{sw}$, $\delta^{44/40}Ca_{sw}$, $^{87}Sr/^{86}Sr$), to date,
no direct archive has been documented for $\delta^{11}B_{sw}$. This is a result of the pH-
dependent boron speciation in seawater upon which the $\delta^{11}B$-pH proxy is based
(Hemming & Hanson 1992) that imparts a pH dependency on the $\delta^{11}B$ of all marine
precipitates so far examined. Empirical reconstructions of $\delta^{11}B_{sw}$ must therefore use
"indirect" approaches. So far four approaches have been applied to the problem (Fig.
2): (1) geochemical modeling (Lemarchand et al., 2000), (2) $\delta^{11}B$ analysis of halites
(Paris et al., 2010), (3) measurements of benthic foraminiferal $\delta^{11}B$ coupled to
various assumptions about past changes in ocean pH (Raitzsch and Hönisch, 2013),
and (4) measurements of $\delta^{11}B$ in surface and thermocline dwelling foraminifera
coupled with additional information on the pH gradient of the surface ocean (Palmer
et al., 1998; Pearson and Palmer 1999, Pearson and Palmer 2000; Anagnostou et al.,
2016). Geochemical modelling of the changes in the flux of boron into and out of the
ocean through time has been used to suggest that $\delta^{11}B_{sw}$ increased from 37‰ at 60
Ma to 40‰ ± 1‰ today, driven by a combination of processes including changing
boron continental discharge (Lemarchand et al., 2000). In the case of approach 2,
while modern natural halites reflect $\delta^{11}B_{sw}$ (39.7 ‰) with no apparent fractionation,
measurement of $\delta^{11}B$ in ancient halites yield isotopic ratios that are significantly
lower than all other approaches (Fig. 2; Paris et al., 2010), with implausible
variability among samples of the same age (7‰ range), thereby casting doubt over
the reliability of this approach (Raitzsch and Hönisch, 2013). In the case of approach
3, $\delta^{11}B_{sw}$ is calculated from globally distributed benthic $\delta^{11}B$ data with an imposed
degree of deep-ocean pH change (Fig. 2; Raitzsch and Hönisch, 2013). This method
hinges on two key assumptions: (a) a near linear surface water pH increase of 0.39
over the past 50 Myrs taken from the average pH output from a number of modeling
studies (Berner and Kothavala, 2001; Tyrrell and Zeebe, 2004; Ridgwell, 2005), and
(b) a prescribed constant surface-to-deep ocean pH gradient of 0.3 (Tyrrell and
Zeebe, 2004, and modern observations). The modeled surface pH and estimated fixed
pH gradient is then used to estimate deep ocean pH, and then convert benthic
foraminiferal $\delta^{11}B$ measurements to $\delta^{11}B_{sw}$. This approach yields broadly similar
results to geochemical modeling (Fig. 2). The fourth approach exploits the non-linear
relationship between $\delta^{11}B$ and pH alongside estimated pH gradients in the ocean to
constrain $\delta^{11}B_{sw}$ (Palmer et al., 1998; Pearson and Palmer 1999, Pearson and Palmer
2000) and is the basis of the approach used in this study. The advantage of this
method is that $\delta^{11}B_{sw}$ can be reconstructed empirically without relying on *a priori*
absolute-pH constraints. The non-linear relationship between $\delta^{11}B$ and pH means that
the pH difference between two $\delta^{11}B$ data points varies as a function of $\delta^{11}B_{sw}$ (Fig. 3).
Consequently, if the size of the pH gradient can be estimated then there is only one
$\delta^{11}B_{sw}$ value that is consistent with the foraminiferal $\delta^{11}B$ measurements and the
specified pH gradient irrespective of the absolute pH (Fig. 3c). Previously this
approach has been applied to pH variations in the surface ocean and used in studies
of Cenozoic $p$CO$_2$ to account for changes in $\delta^{11}B_{sw}$ (determined using $\delta^{11}B$ in surface
and thermocline-dwelling foraminifera) (Fig. 2) (Palmer et al., 1998; Pearson and
Palmer 1999, Pearson and Palmer 2000; Anagnostou et al., 2016). This approach uses
a constant pH gradient between the surface and some depth proximal to the oxygen
minimum zone and the boron isotope values of a mixed layer dwelling species and
thermocline dweller to calculate a value for $\delta^{11}B_{sw}$ (Pearson and Palmer, 1999). The
resulting record suggests that $\delta^{11}B_{sw}$ varies between 37.7‰ and 39.4‰ through the
Neogene (Fig. 2) (Pearson and Palmer, 2000).
The same method, but using planktic-benthic instead of surface planktic-thermocline
planktic $\delta^{11}B$ gradients to calculate $\delta^{11}B_{sw}$, was recently applied to the middle
Miocene where it yielded a $\delta^{11}B_{sw}$ of 37.6 $^{+0.4}_{-0.5}$ ‰ (Foster et al., 2012). A further
modification to the method of Pearson and Palmer (1999) was also proposed in that
study wherein $\delta^{13}C$ in foraminiferal calcite was used to estimate the surface-to-deep
pH gradient (Foster et al., 2012). Here, we reconstruct $\delta^{11}B_{sw}$ for the last 23 Ma, the
Neogene, based on this modified approach. We undertake extensive sensitivity tests
using both the CYCLOPS carbon cycle box model and the GENIE Earth system
model to define the plausible range in the relationship between surface/deep pH
difference and $\delta^{13}C$ difference, which is an essential parameter for this approach.
Finally, we employ a Monte Carlo approach for comprehensive propagation of
uncertainty in all input parameters and we focus on reconstructing $\delta^{11}B_{sw}$ – the
implications of our work for understanding the evolution of Neogene ocean pH and
atmospheric $p\mathrm{CO_2}$ will be documented elsewhere.
## 2. Methods
### 2.1 Site Locations and Age Models
Foraminifera from four sites are used to construct the planktic-benthic $\delta^{11}B$ pairs;
Ocean Drilling Program, ODP, Site 758 and ODP Site 999 for the Pleistocene and
Pliocene samples and ODP Site 926 and Site 761 for the Miocene (Fig. 4) (this study;
Foster et al., 2012; Martìnez-Botì et al., 2015a, and a follow up study by Sosdian et
al.,). We also incorporate the middle Miocene planktic-benthic pair from Site 761 in
Foster et al. (2012). To place all data from all sites on a single age model we use the
nanno and planktic foraminifera stratigraphy from sites 999, 926 and 761 (Shipboard
Scientific Party, 1997; Shipboard Scientific Party, 1995; Zeeden et al., 2013;
Holbourn et al., 2004) updated to GTS2012 (Gradstein et al., 2012). At Site 758 the
magnetostratigraphy (Shipboard Scientific Party, 1989) is used and updated to
GTS2012 (Gradstein et al., 2012).

## 2.2 Boron Isotope Analysis and pH Calculation

The boron isotope measurements (expressed in delta notation as $\delta^{11}B$ – permil variation) were made relative to the boric acid standard SRM 951; (Catanzaro et al., 1970). Boron was first separated from the Ca matrix prior to analysis using the boron specific resin Amberlite IRA 743 following Foster et al. (2013). The boron isotopic composition was then determined using a sample-standard bracketing routine on a ThermoFisher Scientific Neptune multicollector inductively coupled plasma mass spectrometer (MC-ICPMS) at the University of Southampton (following Foster et al., 2013). The relationship between $\delta^{11}B$ of $CaCO_3$ and pH is very closely approximated by the following equation:

$$pH = pK_B^* - \log\left(-\frac{\delta^{11}B_{SW} - \delta^{11}B_{CaCO_3}}{\delta^{11}B_{SW} - \propto_B . \delta^{11}B_{CaCO_3} - 1000 . (\propto_B - 1)}\right) \qquad (1)$$

Where $pK_B^*$ is the equilibrium constant, dependent on salinity, temperature, pressure and seawater major ion composition (i.e., [Ca] and [Mg]), $\propto_B$ is the fractionation factor between the two boron species and $\delta^{11}B_{sw}$ is the boron isotope composition of seawater. Here we use the fractionation factor of 1.0272, calculated from spectrophotometric measurements (Klochko et al., 2006). No temperature correction was applied as a number of recent studies suggest that it is not significant over our investigated temperature range (Rae et al. 2011; Henehan et al., 2013; Martínez-Botí et al. (2015b); Kaczmarek et al. 2016). Although the $\delta^{11}B$ of foraminifera correlates well with pH and hence $[CO_2]_{aq}$, the $\delta^{11}B_{calcite}$ is often not exactly equal to $\delta^{11}B_{borate}$ (Sanyal et al., 2001; Foster, 2008; Henehan et al., 2013). The planktic species used to construct the benthic-planktic pairs changes through time, as a single species is not available for the entire Neogene (this study; Foster et al., 2012; Martìnez-Botì et al., 2015a, and a follow up study by Sosdian et al.). Here *Globigerinoides ruber* is used for 0 to 3 Ma, *Trilobatus sacculifer* (formally *Globigerinoides sacculifer* and including *Trilobatus trilobus*; Hembleden et al., 1987; Spezzaferri et al., 2015) for 0 to 20 Ma and *Globigerina praebulloides* for 22 to 23 Ma. The calibration for *G. ruber* (300-355μm) is derived from culturing data supported by core top data (Henehan et al., 2013). The *T. sacculifer* calibration (300-355μm) is from a follow up

study by Sosdian et al. where the *T. sacculifer* calibration of Sanyal et al., (2001) is
used with a modified intercept so that it passes through the core top value for *T.*
*sacculifer* (300–355 μm) from ODP Hole 999A (Seki et al., 2010). Unlike the
asymbiotic modern *G. bulloides, G. praebulloides* appears to be symbiotic at least in
the latest Oligocene (Pearson and Wade, 2009). Therefore, we apply the *T. sacculifer*
(300-355μm) calibration to this species. For *T. sacculifer* (500-600μm) between 0
and 1 Ma, we use the calibration from Martìnez-Botì et al. (2015b) where the
calibration of Sanyal et al. (2001) measured using NTIMS is corrected for the offset
between MC-ICPMS and NTIMS using a comparison of core-top *T. sacculifer*
measured by the two different methods from adjacent sites (Foster, 2008; Sanyal et
al., 1995). In order to constrain deep-water pH, analysis was conducted on benthic
foraminifera *Cibicidoides wuellerstorfi* or *Cibicidoides mundulus* depending on
which species were most abundant in each sample. The $\delta^{11}B$ of both *Cibicidoides*
species shows no offset from the theoretical $\delta^{11}B$ of the borate ion and therefore no
calibration is needed to adjust for species-specific offsets (Rae et al., 2011).
As mentioned above, in addition to $\delta^{11}B_{calcite}$, temperature, salinity, water depth
(pressure) and seawater major ion composition are also needed to calculate pH from
$\delta^{11}B$. We use the MyAMI specific ion interaction model (Hain et al., 2015) to
calculate the appropriate equilibrium constants based on existing [Ca] and [Mg]
reconstructions (Horita et al., 2002; Brennan et al., 2013). Sea surface temperature
(SST) is calculated from tandem Mg/Ca analyses on an aliquot of the $\delta^{11}B$ sample
(with a conservative 2σ uncertainty of 2°C). Adjustments were made for changes in
Mg/Ca$_{sw}$ using the records of Horita et al. (2002) and Brennan et al. (2013), and
correcting for changes in dependence on Mg/Ca$_{sw}$ following Evans and Müller (2012)
using H = 0.41 calculated from *T. sacculifer* (where H describes the power
relationship between test Mg/Ca incorporation and Mg/Ca$_{sw}$; Delaney et al., 1985;
Hasiuk and Lohmann, 2010; Evans and Müller, 2012) using the equations:

$$Mg/Ca_{sw.c} = (Mg/Ca_{sw.a} \ / Mg/Ca_{sw.m})^{0.41} \qquad (2)$$

Where Mg/Ca$_{sw.c}$ is the correction factor applied to the temperature equation for
changing Mg/Ca$_{sw}$, Mg/Ca$_{sw.a}$ is the estimated Mg/Ca$_{sw}$ for the age of the sample and
Mg/Ca$_{sw.m}$ is modern Mg/Ca$_{sw}$. Temperature is then calculated using the generic
planktic foraminifera calibration of Anand et al. (2003) and including a correction
factor for $Mg/Ca_{sw}$.

$$Temperature = ln(Mg/Ca_{test}/(0.38 * Mg/Ca_{sw.c}))/0.09 \qquad (3)$$

Mg/Ca analysis was conducted on a small aliquot of the sample dissolved for isotope
analysis at the University of Southampton using a ThermoFisher Scientific Element 2
XR. Al/Ca was also measured to assess the competency of the sample cleaning.
Because of complications with the Mg/Ca-temperature proxy in *Cibicidoides* species
(Elderfield et al., 2006), bottom water temperatures (BWTs) are estimated here by
taking the global secular temperature change from the Mg/Ca temperature
compilation of Cramer et al. (2011), using the calibration of Lear et al. (2010) and
applying this change to the modern bottom water temperature at each site taken from
the nearest GLODAP site (with a conservative 2σ uncertainty of 2°C). Salinity is held
constant at modern values determined from the nearest GLODAP site (2σ uncertainty
of 2 ‰ uncertainty) for the entire record. Note that temperature and salinity have
little influence on the calculated pH and the uncertainty in $\delta^{11}B_{sw}$ is dominated by the
uncertainty in the $\delta^{11}B$ measurement and the estimate of the pH gradient.
The majority of the $\delta^{13}C$ data were measured at Cardiff University on a
ThermoFinnigan MAT 252 coupled with a Kiel III carbonate device for automated
sample preparation. Additional samples were measured on a gas source mass
spectrometer Europa GEO 20-20, University of Southampton equipped with
automated carbonate preparation device and on a Finnigan MAT 253 gas isotope
ratio mass spectrometer connected to a Kiel IV automated carbonate preparation
device at the Zentrum für Marine Tropenökologie (ZMT), Bremen. The Pliocene
benthic $\delta^{13}C$ from Site 999 were taken from the nearest sample in Haug and
Tiedemann, (1998). In almost all cases $\delta^{13}C$ was analysed on the same foraminiferal
species as $\delta^{11}B$ and Mg/Ca (38/44 samples). Where this was not possible another
surface dweller/benthic foraminifera was used from the same depth habitat. *C.*
*wuellerstorfi* or *C. mundulus* were measured in all cases for benthic $\delta^{13}C$. Stable
isotope results are reported relative to the Vienna Peedee belemnite (VPDB)
standard. We use a carbon isotope vital effect for *G. ruber* (+0.94 ‰; Spero et al.,
2003), *T. sacculifer*/*G. praebulloides* (+0.46 ‰; Spero et al., 2003; Al-Rousan et al.,
2004;), *C. mundulus* (+0.47 ‰; McCorkle et al., 1997) and *C. wuellerstorfi* (+0.1 ‰;

245 McCorkle et al., 1997) to calculate the $\delta^{13}C$ of dissolved inorganic carbon (DIC).

246 **2.3 Carbon isotopes as a proxy for vertical ocean pH gradient**

247 The use of $\delta^{13}C$ in foraminiferal calcite to estimate the surface to deep pH gradient

248 requires knowledge of the slope of the pH-$\delta^{13}C$ relationship in the past. In this section

249 we briefly outline the main factors that contribute to the pH-$\delta^{13}C$ relationship in order

250 to underpin our analysis of extensive carbon cycle model simulations.

251 The production, sinking and sequestration into the ocean interior of low-$\delta^{13}C$ organic

252 carbon via the soft-tissue component of the biological pump leads to a broad

253 correlation between $\delta^{13}C$, $[CO_3^{2-}]$ and macronutrients in the ocean (e.g., Hain et al.,

254 2014a). The remineralization of this organic matter decreases $\delta^{13}C$ and titrates $[CO_3^{2-}]$

255 thereby reducing pH, while nutrient concentrations are increased. In waters that have

256 experienced more soft tissue remineralization both pH and $\delta^{13}C$ will be lower (Fig.

257 5a,b), and this is the dominant reason for the positive slope between $\delta^{13}C$ and pH in

258 the modern ocean (e.g., Foster et al., 2012; Fig. 5c).

259 Another significant factor affecting the spatial distribution of both $\delta^{13}C$ and pH is

260 seawater temperature, which affects both the equilibrium solubility of DIC and the

261 equilibrium isotopic composition of DIC. Warmer ocean waters have decreased

262 equilibrium solubility of DIC and so increased local $[CO_3^{2-}]$ and pH (Goodwin and

263 Lauderdale, 2013), while warmer waters have relatively low equilibrium $\delta^{13}C$ values

264 (Lynch-Stieglitz et al, 1995). This means that a spatial gradient in temperature acts to

265 drive $\delta^{13}C$ and pH in opposite directions: warmer waters tend to have higher pH but

266 lower $\delta^{13}C$. These opposing temperature effects act to reduce the pH difference

267 between two points with greatly different temperature to below the value expected

268 based on $\delta^{13}C$ alone. In other words, when using $\delta^{13}C$ differences to estimate the pH

269 gradient between the warm low latitude surface and cold deep waters the appropriate

270 $\Delta$pH-$\Delta\delta^{13}C$ gradient will be less than expected when only considering the effect of

271 organic carbon production, sinking and sequestration. For this reason, in our

272 modeling analysis we focus on the warm-surface to cold-bottom $\Delta$pH/$\Delta\delta^{13}C$ rather

273 than the slope of the overall pH-$\delta^{13}C$ relationship, with the latter expected to be

274 greater than the former.

In the modern ocean, and for the preceding tens of millions of years, the two
dynamics described above are likely dominant in setting spatial variation in $\delta^{13}C$ and
pH (and $[CO_3^{2-}]$). However, other processes will have a minor effect on either pH or
$\delta^{13}C$. For instance, the dissolution of $CaCO_3$ shells increases $[CO_3^{2-}]$ and pH
(Broecker and Peng, 1982), but does not significantly affect $\delta^{13}C$ (Zeebe and Wolf-
Gladrow, 2001). Moreover, the long timescale of air/sea isotopic equilibration of $CO_2$
combined with kinetic isotope fractionation during net carbon transfer is an important
factor in setting the distribution of $\delta^{13}C$ on a global ocean scale (Galbraith et al.,
2015; Lynch-Stieglitz et al., 1995), while the effect of $CO_2$ disequilibrium on $[CO_3^{2-}]$
and pH is modest (Goodwin and Lauderdale, 2013).

**2.4 Modelling the pH to $\delta^{13}C$ relationship**
After correcting for the shift in $\delta^{13}C$ due to anthropogenic activity, or Suess effect
(Keeling 1979), modern global ocean observations demonstrate a near-linear
relationship between global ocean data of *in situ* seawater pH and $\delta^{13}C$ DIC with a
slope of $0.201 \pm 0.005$ ($2\sigma$) (Foster et al., 2012; Fig 5c.) This empirically determined
slope might well have been different in past oceans with very different nutrient
cycling, carbon chemistry and circulation compared to today, and it does not
appropriately represent the temperature effect described above (i.e., warm-surface to
cold-bottom water $\Delta pH/\Delta\delta^{13}C$). Here we use an ensemble approach with two
independent carbon cycle models to investigate changes in the $\Delta pH/\Delta\delta^{13}C$ regression.
Below we provide pertinent information on the GENIE and CYCLOPS model
experiments:
We use the Earth System model GENIE-1 (Edwards and Marsh, 2005; Ridgwell et al.
2007) to assess the robustness of the $\Delta pH$-to-$\Delta\delta^{13}C$ relationship and its sensitivity to
physical and biogeochemical ocean forcing. The configuration used here is closely
related to that of Holden et al. (2013), in which the controls on oceanic $\delta^{13}C$
distribution were assessed, with an energy and moisture balance in the atmosphere,
simple representations of land vegetation and sea ice, and frictional geostrophic
ocean physics. In each of 16 vertical levels in the ocean, increasing in thickness with
depth, there are 36x36 grid cells (10° in longitude and nominally 5° in latitude, with
higher resolution at low latitudes). Modern ocean bathymetry and land topography is
applied in all simulations. The ocean biogeochemical scheme (Ridgwell et al. 2007)
is based on conversion of DIC to organic carbon associated with phosphate uptake
with fixed P:C:O stoichiometry. Organic carbon and nutrients are remineralized
according to a remineralization profile with a pre-defined $e$-folding depth scale. This
depth scale, as well as the rain ratio of inorganic to organic carbon in sinking
particulate matter, is among the parameters examined in the sensitivity study. In these
simulations, there is no interaction with sediments. As a result of this, the steady state
solutions reported here are reached within the 5000-year simulations, but they are not
consistent with being in secular steady state with regard to the balance of continental
weathering and ocean $CaCO_3$ burial.
The sensitivity study consists of seven sets of experiments, each varying a single
model parameter relative to the control simulation with preindustrial atmospheric
$p CO_2$. This enables us to assess which processes, if any, are capable of altering the
oceanic relationship between $\Delta pH$ and $\Delta \delta^{13}C$ relationship, and the uncertainty in the
predictive skill of this relationship due to spatial variability. These experiments are
therefore exploratory in nature and intended to study plausible range rather than
determine magnitude of past changes. The seven parameters varied are (1) the ocean
alkalinity reservoir; (2) the ocean's carbon reservoir; (3) the parameter "S. Lim gas
exchange" which blocks air-sea gas exchange south of the stated latitude, significant
here because of the dependence of $\delta^{13}C$ on surface disequilibrium (Galbraith et al.,
2015); (4) inorganic to organic carbon rain ratio, controlling the relationship between
DIC and alkalinity distributions; (5) "Antarctic shelf FWF", a freshwater flux
adjustment (always switched off in control experiments with GENIE) facilitating the
formation of brine rich waters, which produces a high-salinity poorly-ventilated deep
ocean at high values; (6) "Atlantic-Pacific FWF", a freshwater flux adjustment
equivalent to freshwater hosing, leading to a shut-down of the Atlantic meridional
overturning circulation at low values; (7) remineralization depth-scale of sinking
organic matter, which affects the vertical gradient both of pH and $\delta^{13}C$. A wide range
of parameter values is chosen for each parameter in order to exceed any plausible
changes within the Cenozoic.
For the second exploration of the controls on the slope of the $\Delta pH$-$\Delta\delta^{13}C$ relationship
we use the CYCLOPS biogeochemical 18-box model that includes a dynamical
lysocline, a subantarctic zone surface box and a polar Antarctic zone box (Sigman et
al., 1998; Hain et al., 2010, 2014b). The very large model ensemble with 13,500
individual model scenarios is designed to capture the full plausible range of (a)
glacial/interglacial carbon cycle states by sampling the full solution space of Hain et
al. (2010), and (b) reconstructed secular changes in seawater [Ca] (calcium
concentration), carbonate compensation depth (CCD), weathering and atmospheric
$CO_2$ (Table 1). The following seven model parameters are systematically sampled to
set the 13,500 model scenarios: (1) shallow versus deep Atlantic meridional
overturning circulation represented by modern reference north Atlantic deep water
(NADW) versus peak glacial North Atlantic intermediate water (GNAIW)
circulation; (2) iron-driven changes in nutrient drawdown in the subantarctic zone of
the Southern Ocean; (3) changes in nutrient drawdown of the polar Antarctic; (4)
changes in vertical exchange between the deep Southern Ocean and the polar
Antarctic surface; (5) range in seawater [Ca] concentration from 1x to 1.5x modern as
per reconstructions (Horita et al., 2002); (6) Pacific CCD is set to the range of 4.4-4.9
km via changes in the weathering flux, as per sedimentological evidence (Pälike et
al., 2012); (7) atmospheric $CO_2$ is set from 200 ppm to 1000 ppm by changes in the
'weatherability' parameter of the silicate weathering mechanism. The ensemble spans
predicted bulk ocean DIC between 1500 and 4500 $\mu$mol/kg, a wide range of ocean
pH and $CaCO_3$ saturation states consistent with the open system weathering cycle,
and widely different states of the oceanic biological pump. All 13,500 model
scenarios are run for two million years after every single 'weatherability' adjustment,
part of the CCD inversion algorithm, guaranteeing the specified CCD depth and
steady state with regard to the balance of continental weathering and ocean $CaCO_3$
burial for the final solution (unlike the GENIE simulations $CaCO_3$ burial was entirely
neglected due to computational cost of the long model integrations it would require).
The inverse algorithm typically takes at least ten steps to conversion, resulting in
~300 billion simulated years for this ensemble. This range of modelling parameters
was chosen to exceed the range of carbonate system and ocean circulation changes
that can be expected for the Neogene based on records of [Ca] and [Mg] (Horita et
al., 2002), CCD changes (Pälike et al., 2012), atmospheric $CO_2$ (Beerling and Royer,
2011) and records of glacial-interglacial circulation change (Curry and Oppo, 2005).
**2.5 Assessing uncertainty**
$\delta^{11}B_{sw}$ uncertainty was calculated using a Monte Carlo approach where pH was
calculated for deep and surface waters at each time slice using a random sampling
(n=10000) of the various input parameters within their respective uncertainties as
represented by normal distributions. These uncertainties ($2\sigma$ uncertainty in
parentheses) are: temperature ($\pm 2$ °C), salinity ($\pm 2$ units on the practical salinity
scale) [Ca] ($\pm 4.5$ mmol/kg), [Mg], ($\pm 4.5$ mmol/kg), $\delta^{11}B_{planktic}$ ($\pm 0.15-0.42$ ‰) and
$\delta^{11}B_{benthic}$ ($\pm 0.21-0.61$ ‰). For the estimate of the surface to sea floor pH gradient we
use the central value of the $\Delta pH$-to-$\Delta\delta^{13}C$ relationship diagnosed from our
CYCLOPS and GENIE sensitivity experiments (i.e., 0.175/‰, see section 3.2 below)
and then we assign a $\pm 0.05$ uncertainty range with a uniform probability (rather than
a normal distribution) to the resulting surface to sea floor $\Delta pH$ estimate (see also
Table 2). Thus, the magnitude of this nominal uncertainty is equivalent to a 0.14/‰
to 0.21/‰ $\Delta pH/\Delta\delta^{13}C$ uncertainty range that spans the vast majority of our
CYCLOPS and GENIE simulations, and the prediction error (RMSE) of fitting a
linear relationship to the GENIE pH and $\delta^{13}C$ output (see section 3.2 below). The
uncertainty in the $\delta^{11}B$ measurements is calculated from the long-term reproducibility
of Japanese Geological Survey *Porites* coral standard (JCP; $\delta^{11}B=24.3$‰) at the
University of Southampton using the equations:
$$2\sigma = 2.25\,\exp^{-23.01[^{11}B]} + 0.28\,\exp^{-0.64[^{11}B]} \qquad (4)$$
$$2\sigma = 33450\,\exp^{-168.2[^{11}B]} + 0.311\,\exp^{-1.477[^{11}B]} \qquad (5)$$
where $[^{11}B]$ is the intensity of $^{11}B$ signal in volts and equation (4) and equation (5)
used with $10^{11}\,\Omega$ and $10^{12}\,\Omega$ resistors, respectively.
From the 10,000 Monte Carlo ensemble solutions of our 22 benthic-planktic pairs we
construct 10,000 randomized records of $\delta^{11}B_{sw}$ as a function of time. Each of these
randomized $\delta^{11}B_{sw}$ records are subjected to smoothing using the locally weighted
scatterplot smoothing (LOWESS) algorithm with a smoothing parameter (span) of
0.7. The purpose of the smoothing is to put some controls on the rate at which the
resulting individual Monte Carlo $\delta^{11}B_{sw}$ records are allowed to change, which in
reality is limited by the seawater boron mass balance (~0.1 ‰ per million years;
boron residence time is 11-17 million years; Lemarchand et al., 2000). Our choice of
smoothing parameter allows for some of the individual Monte Carlo records to
change as fast as ~1 ‰ per million years, although in reality the average rate of
change is much smaller than this (see section 3.3). Consequently this method
removes a significant amount of uncorrelated stochastic noise (resulting from the
uncertainty in our input parameters) while not smoothing away the underlying signal.
As a result of anomalously low $\delta^{11}B$ differences (< 1‰) between benthic and planktic
pairs, two pairs at 8.68 Ma and 19 Ma were discarded from the smoothing. It may be
possible that preservation is not so good within these intervals and the planktic
foraminifera are affected by partial dissolution (Seki et al., 2010). The spread of the
ensemble of smoothed $\delta^{11}B_{sw}$ curves represents the combination of the compounded,
propagated uncertainties of the various inputs (i.e., Monte Carlo sampling) with the
additional constraint of gradual $\delta^{11}B_{sw}$ change over geological time imposed by the
inputs and outputs of boron to the ocean and the total boron inventory (i.e., the
smoothing of individual Monte Carlo members. Various statistical properties (i.e.,
mean, median, standard deviation ($\sigma$), various quantiles) of this $\delta^{11}B_{sw}$ reconstruction
were evaluated from the ensemble of smoothed $\delta^{11}B_{sw}$ records. Generally, for any
given benthic-planktic pair the resulting $\delta^{11}B_{sw}$ estimates are not perfectly normally
distributed and thus we use the median as the metric for the central tendency (i.e.,
placement of marker in Figure 10).
**3. Results and Discussion**
**3.1 $\delta^{11}B$ benthic and planktic data**
Surface and deep-ocean, $\delta^{11}B$ broadly show a similar, but inverse, pattern to $\delta^{13}C$ and
temperature throughout the Neogene (Fig. 6). The $\delta^{11}B$ benthic record decreases from
~15 ‰ at 24 Ma to a minimum of 13.28 ‰ at 14 Ma before increasing to ~17 ‰ at
present day (Fig. 6). This pattern and the range of values in benthic foraminiferal $\delta^{11}B$
is in keeping with previously published Neogene $\delta^{11}B$ benthic records measured
using NTIMS (Raitzsch and Hönisch, 2013), suggesting that our deep-water $\delta^{11}B$
record is representative of large scale pH changes in the global ocean. While the
surface $\delta^{11}B_{planktic}$ remained relatively constant between 24 and 11 Ma at ~16 ‰, there
is a significant increase in $\delta^{11}B$ between the middle Miocene and present (values
increase to ~20 ‰) (Fig. 6b). The reconstructed surface water temperatures show a
long-term decrease through the Neogene from ~28°C to 24°C, aside from during the
Miocene Climatic Optimum (MCO) where maximum Neogene temperatures are
reached (Fig. 6c). Following Cramer et al. (2011) deep-water temperatures decrease
from ~12°C to 4°C at the present day and similarly show maximum temperatures in
the MCO. Surface and deep-water $\delta^{13}C_{DIC}$ both broadly decrease through the Neogene
and appear to covary on shorter timescales (Fig. 6e, f).

## 3.2 The relationship between $\delta^{13}C$ and pH gradients

In the global modern ocean data, after accounting for the anthropogenic carbon, the
empirical relationship between *in situ* pH and DIC $\delta^{13}C$ is well described by a linear
function with a slope of 0.201± 0.005 (2σ) (Fig. 5; Foster et al., 2012). However, this
slope is only defined by surface waters in the North Atlantic due to a current lack of
modern data where the impact of the Suess effect has been corrected (Olsen and
Ninneman, 2010). Consequently we are not currently able to determine the slope
between the warm-surface and cold-deep ocean in the modern ocean at our sites.
Instead, here we use the two modeling experiments to define this slope. In the control
GENIE experiment (green star; Fig. 7), the central value for the slope of the pH/$\delta^{13}C$
relationship is slightly greater than 0.2/‰ for the full 3D data regression (not shown)
and about 0.175/‰ for the warm-surface-to-cold-deep $\Delta pH$-to-$\Delta\delta^{13}C$ relationship
(Fig. 7) – consistent with theory for the effect of temperature gradients (see section
2.3). For both ways of analysing the GENIE output the prediction uncertainty of the
regressions, the root-mean-squared error (RMSE), is ~0.05/‰ under most conditions
(open red circles in Fig. 7), with the exception of cases where large changes in either
DIC or ALK yield somewhat larger changes in the relationship between pH and $\delta^{13}C$
(see below). In our CYCLOPS model ensemble, the central value of the slopes of the
full 3D pH/$\delta^{13}C$ regressions and of the warm-surface-to-cold-deep $\Delta pH/\Delta\delta^{13}C$ is
0.2047/‰ (1σ of 0.0196/‰; Fig.8a) and 0.1797/‰ (1σ of 0.0213/‰; Fig.8b),
respectively. If we restrict our analysis of the CYCLOPS ensemble to only the
Atlantic-basin warm-surface-to-cold-deep $\Delta pH/\Delta\delta^{13}C$, where most of our samples
come from, we find a relationship of only 0.1655/‰ (1σ of 0.0192/‰; Fig.8c). That
is, overall, we find near-perfect agreement between modern empirical data and our
GENIE and CYCLOPS experiments. Encouraged by this agreement we select the
warm-surface-to-cold-deep $\Delta pH/\Delta\delta^{13}C$ central value of 0.175/‰ to estimate the
surface/sea floor pH difference from the planktic/benthic foraminifera $\delta^{13}C$
difference. To account for our ignorance as to the accurate value of $\Delta pH/\Delta\delta^{13}C$ in the
modern ocean, its temporal changes over the course of the study interval and the
inherent prediction error from using a linear $\Delta pH$-to-$\Delta\delta^{13}C$ relationship, we assign a
nominal uniform uncertainty range of ±0.05 around the central $\Delta pH$ estimate for the
purpose of Monte Carlo uncertainty propagation. Our analysis also suggests that
where surface-to-thermocline planktic/planktic gradients are employed, the plausible
$\Delta pH/\Delta\delta^{13}C$ range should be significantly higher than applied here to account for the
relatively lower temperature difference. Based on the appropriate $\Delta pH/\Delta\delta^{13}C$
relationship we reconstruct a time varying surface-to-deep pH gradient, which
ranges between 0.14 and 0.35 pH units over our study interval (Fig. 9) and apply a
flat uncertainty of ± 0.05. The reconstructed pH gradient remains broadly within the
range of the modern values (0.19 to 0.3) although there is some evidence of multi-
million year scale variability (Fig. 9).
As a caveat to our usage of the $\Delta pH$-to-$\Delta\delta^{13}C$ relationship we point to changes of that
relationship that arise in our GENIE sensitivity experiments where carbon and
alkalinity inventories are manipulated, which can yield values outside of what is
plausible. We note that our CYCLOPS ensemble samples a very much wider range of
carbon and alkalinity inventories with $\Delta pH/\Delta\delta^{13}C$ remaining inside that range. While
CYCLOPS simulates the balance between weathering and $CaCO_3$ burial, which is
known to neutralize sudden carbon or alkalinity perturbations on timescales much
less than one million years, the configuration used for our GENIE simulations does
not and is therefore subject to states of ocean carbon chemistry that can safely be
ruled out for our study interval and likely for most of the Phanerozoic. The differing
outputs from CYCLOPS and GENIE in the DIC and ALK experiments shows that
$\Delta pH/\Delta\delta^{13}C$ depends on background seawater acid/base chemistry, in ways that are
not yet fully understood. That said, the generally coherent nature of our results
confirms that we likely constrain the plausible range of $\Delta pH/\Delta\delta^{13}C$ for at least the
Neogene, if not the entire Cenozoic, outside of extreme events such as the
Palaeocene-Eocene Thermal Maximum.

**497    3.3 $\delta^{11}B_{sw}$ record through the Neogene**

Using input parameter uncertainties as described in section 2.5 yields individual
Monte Carlo member $\delta^{11}B_{sw}$ estimates between 30 ‰ and 43.5 ‰ at the overall
extreme points and typically ranging by ~10 ‰ (dashed in Fig. 10a) for each time
point, suggesting that the uncertainties we assign to the various input parameters are
generous enough not to predetermine the quantitative outcomes. However, for each
planktic/benthic time point most individual Monte Carlo $\delta^{11}B_{sw}$ estimates fall into a
much narrower central range (~1 ‰ to 4 ‰; thick black line showing interquartile
range in Fig. 10a). The $\delta^{11}B_{sw}$ for Plio-Pleistocene time-points cluster around ~40 ‰
while middle/late Miocene values cluster around ~36.5 ‰. The estimates at
individual time points are completely independent from each other, such that the
observed clustering is strong evidence for an underlying long-term signal in our data,
albeit one that is obscured by the uncertainties involved in our individual $\delta^{11}B_{sw}$
estimates. The same long-term signal is also evident when pooling the individual
Monte Carlo member $\delta^{11}B_{sw}$ estimates into 8 million year bins and evaluating the
mean and spread ($2\sigma$) in each bin (Fig. 10b).  This simple treatment highlights that
there is a significant difference between our Plio-Pleistocene and middle Miocene
data bins at the 95% confidence level and that $\delta^{11}B_{sw}$ appears to also have been
significantly lower than modern during the early Miocene.

**516    3.3.1   Data smoothing**

The ~1 to 4 ‰ likely ranges for $\delta^{11}B_{sw}$ would seem to be rather disappointing given
the goal to constrain $\delta^{11}B_{sw}$ for pH reconstructions. However, most of that uncertainty
is stochastic, random error that is uncorrelated from time point to time point.
Furthermore, we know from mass balance considerations that $\delta^{11}B_{sw}$ of seawater
should not change by more than ~0.1 ‰ per million years (Lemarchand et al., 2000),
because of the size of the oceanic boron reservoir compared the inputs and outputs
(see Fig. 1), and we use this as an additional constraint via the LOWESS smoothing
we apply to each Monte Carlo time series. One consideration is that every individual
Monte Carlo $\delta^{11}B_{sw}$ estimate is equally likely and the smoothing should therefore
target randomly selected individual Monte Carlo $\delta^{11}B_{sw}$ estimates, as we do here,
rather than smoothing over the likely ranges identified for each time point. In this
way the smoothing becomes integral part of our Monte Carlo uncertainty propagation
and the spread among the 10,000 individual smoothed $\delta^{11}B_{sw}$ curves carries the full
representation of propagated input uncertainty conditional on the boron cycle mass
balance constraint. A second consideration is that the smoothing should only remove
noise, not underlying signal. As detailed above, for this reason the smoothing
parameter we choose has enough freedom to allow the $\delta^{11}B_{sw}$ change to be dictated
by the data, with only the most extreme shifts in $\delta^{11}B_{sw}$ removed. We also tested the
robustness of the smoothing procedure itself (not shown) and found only marginal
changes when changing algorithm (LOESS versus LOWESS, with and without
robust option) or when reducing the amount of smoothing (i.e., increasing the
allowed rate $\delta^{11}B_{sw}$ change). The robustness of our smoothing is further underscored
by the good correspondence with the results of simple data binning (Fig.10b).

## 541 3.4 Comparison to other $\delta^{11}B_{sw}$ records

The comparison of our new $\delta^{11}B_{sw}$ record to those previously published reveals that
despite the differences in methodology the general trends in the records show
excellent agreement. The most dominant common feature of all the existing estimates
of Neogene $\delta^{11}B_{sw}$ evolution is an increase through time from the middle Miocene to
the Plio-Pleistocene (Fig. 11). While the model-based $\delta^{11}B_{sw}$ record of Lemarchand et
al. (2000) is defined by a monotonous and very steady rise over the entire study
interval, all three measurement-based records, including our own, are characterized
by a single dominant phase of increase between roughly 12 and 5 Ma. Strikingly, the
Pearson and Palmer (2000) record falls almost entirely within our 95% likelihood
envelope, overall displaying very similar patterns of long-term change but with a
relatively muted amplitude and overall rate of change relative to our reconstruction.
Conversely, some of the second-order variations in the reconstruction by Raitzsch
and Hönisch (2013) are not well matched by our reconstruction, but the dominant
episode of rapid $\delta^{11}B_{sw}$ rise following the middle Miocene is in almost perfect
agreement. We are encouraged by these agreements resulting from approaches based
on very different underlying assumptions and techniques, which we take as indication
for an emerging consensus view of $\delta^{11}B_{sw}$ evolution over the last 25 Ma and as a
pathway towards reconstructing $\delta^{11}B_{sw}$ further back in time. Below we discuss in
more detail the remaining discrepancies between our new and previously existing
$\delta^{11}B_{sw}$ reconstructions.
The record by Pearson and Palmer (2000) is well correlated to our reconstruction, but
especialy during the early Miocene there is a notable ~0.5 ‰ offset (Fig. 11). This
discrepancy could be due to a number of factors. Firstly, the applicability of this
$\delta^{11}B_{sw}$ record (derived from $\delta^{11}B$ data measured using NTIMS) to $\delta^{11}B$ records
generated using the MC-ICPMS is uncertain (Foster et al., 2013). In addition, this
$\delta^{11}B_{sw}$ record is determined using a fractionation factor of 1.0194 (Kakihana et al.,
1977), whereas recent experimental data have shown the value to be higher (1.0272 $\pm$
0.0006, Klochko et al., 2006), although foraminiferal vital effects are likely to mute
this discrepancy. Thirdly, given our understanding of the $\delta^{11}B$ difference between
species/size fractions (Foster, 2008; Henehan et al., 2013), the mixed species and size
fractions used to make the $\delta^{11}B$ measurements in that study may have introduced
some additional uncertainty in the reconstructed $\delta^{11}B_{sw}$. Conversely, there is
substantial spread between our three time points during the earliest Miocene which
combined with the edge effect of the smoothing gives rise to a widening uncertainty
envelope during the time of greatest disagreement with Pearson and Palmer (2000).
This could be taken as indication that our reconstruction, rather than that of Pearson
and Palmer, is biased during the early Miocene.

The $\delta^{11}B_{sw}$ record calculated using benthic $\delta^{11}B$ and assumed deep ocean pH changes
(Raitzsch and Hönisch, 2013) is also rather similar to our $\delta^{11}B_{sw}$ reconstruction. The
discrepancy between the two records in the early Miocene could plausibly be
explained by bias in our record (see above) or may in part be as a result of the
treatment of surface water pH in the study of Raitzsch and Hönisch (2013) and their
assumption of constant surface-deep pH gradient (see Fig 9). The combined output
from two carbon cycle box models is used to make the assumption that surface ocean
pH near-linearly increased by 0.39 over the last 50 Myrs. The first source of surface
water pH estimates is from the study of Ridgwell et al. (2005), where $CO_2$ proxy data
including some derived using the boron isotope-pH proxy is used, leading to some
circularity in the methodology. The second source of surface water pH estimates is
from Tyrrell & Zeebe (2004) and based on GEOCARB where the circularity problem
does not apply. While this linear pH increase broadly matches the $CO_2$ decline from
proxy records between the middle Miocene and present, it is at odds with the $CO_2$
proxy data during the early Miocene that show $CO_2$ was lower than the middle
Miocene during this interval (Beerling and Royer, 2011). Consequently the proxy
$CO_2$ and surface water pH estimates may not be well described by the linear change
in pH applied by Raitzsch and Hönisch (2013) across this interval, potentially
contributing to the discrepancy between our respective $\delta^{11}B_{sw}$ reconstructions.
Our new $\delta^{11}B_{sw}$ record falls within the broad uncertainty envelope of boron mass
balance calculations of Lemarchand et al. (2000), but those modelled values do not
show the same level of multi-million year variability of either Raitzsch and Hönisch
(2013) or our new record, therefore suggesting that the model does not fully account
for aspects of the changes in the ocean inputs and outputs of boron through time on
timescales less than ~10 million years.
In line with the conclusions of previous studies (e.g., Raitzsch and Hönisch, 2013),
our data show that the $\delta^{11}B_{sw}$ signal in the fluid inclusions (Paris et al., 2010) is most
likely a combination of the $\delta^{11}B_{sw}$ and some other factor such as a poorly constrained
fractionation factor between the seawater and the halite. Brine-halite fractionation
offsets of -20‰ to -30‰ and -5‰ are reported from laboratory and natural
environments (Vengosh et al., 1992; Liu et al., 2000). These fractionations and
riverine input during basin isolation will drive the evaporite-hosted boron to low-$\delta^{11}B$
isotope values such that the fluid inclusion record likely provides a lower limit for the
$\delta^{11}B_{sw}$ through time (i.e. $\delta^{11}B_{sw}$ is heavier than the halite fluid inclusions of Paris et al.
(2010)).  For this halite record to be interpreted directly as $\delta^{11}B_{sw}$, a better
understanding of the factor(s) controlling the fractionation during halite formation
and any appropriate correction need to be better constrained.

**3.5 Common controls on the seawater isotopic ratios of B, Mg, Ca and Li**

Our new record of $\delta^{11}B_{sw}$ has some substantial similarities to secular change seen in other marine stable isotope records (Fig. 12). The lithium isotopic composition of seawater ($\delta^7Li_{sw}$; Misra and Froelich, 2012) and the calcium isotopic composition of seawater as recorded in marine barites ($\delta^{44/40}Ca_{sw}$; Griffith et al., 2008) both increase through the Neogene, whereas the magnesium isotopic composition of seawater ($\delta^{26}Mg_{sw}$) decreases (Pogge von Strandmann et al., 2014) suggesting a similar control on the isotopic composition of all four elements across this time interval (Fig. 12). To further evaluate the correlation between these other marine isotope records and $\delta^{11}B_{sw}$, we interpolate and cross-plot $\delta^{11}B_{sw}$ and the $\delta^7Li_{sw}$, $\delta^{44/40}Ca_{sw}$ and $\delta^{26}Mg_{sw}$ records. This analysis suggests that the isotopic composition of $\delta^{11}B_{sw}$, $\delta^7Li_{sw}$, $\delta^{26}Mg_{sw}$ and $\delta^{44/40}Ca_{sw}$ are well correlated through the Neogene, although there is some scatter in these relationships (Fig. 13). Although the Sr isotope record shows a similar increase during the Neogene (Hodell et al., 1991), we focus our discussion on $\delta^{11}B_{sw}$, $\delta^7Li_{sw}$, $\delta^{26}Mg_{sw}$ and $\delta^{44/40}Ca_{sw}$ given that the factors fractionating these stable isotopic systems are similar (see below).

To better constrain the controls on $\delta^{11}B_{sw}$, $\delta^7Li_{sw}$, $\delta^{26}Mg_{sw}$ and $\delta^{44/40}Ca_{sw}$ it is instructive to compare the size and isotopic composition of the fluxes of boron, lithium, calcium and magnesium to the ocean (Table 3). The major flux of boron into the ocean is via riverine input (Lemarchand et al., 2000), although some studies suggest that atmospheric input may also play an important role (Park and Schlesinger, 2002). The loss terms are dominated by adsorption onto clays and the alteration of oceanic crust (Spivack and Edmond, 1987; Smith et al., 1995). Similarly, the primary inputs of lithium into the ocean come from hydrothermal sources and riverine input and the main outputs are ocean crust alteration and adsorption onto sediments (Misra and Froelich, 2012). The three dominant controls on magnesium concentration and isotope ratio in the oceans is the riverine input, ocean crust alteration and dolomitization (Table 3) (Tipper et al., 2006b). The main controls on the amount of calcium in the modern ocean and its isotopic composition is the balance between riverine and hydrothermal inputs and removal through $CaCO_3$ deposition and alteration of oceanic crust (Fantle and Tipper, 2014, Griffith et al., 2008). Dolomitization has also been cited as playing a potential role in controlling

$\delta^{44/40}Ca_{sw}$, although the contribution of this process through time is poorly constrained
(Griffith et al., 2008).
Analysis of the oceanic fluxes of all four ions suggests that riverine input may be an
important factor influencing the changing isotopic composition of B, Li, Ca and Mg
over the late Neogene (Table 3). In the case of all four elements, a combination of the
isotopic ratio of the source rock and isotopic fractionation during weathering
processes are typically invoked to explain the isotopic composition of a particular
river system. However, in most cases the isotopic composition of the source rock is
found to be of secondary importance (Rose et al., 2000; Kısakűrek et al., 2005;
Tipper et al., 2006b; Millot et al., 2010). For instance, the $\delta^{11}B$ composition of rivers
is primarily dependent on isotopic fractionation during the reaction of water with
silicate rocks and to a lesser extent the isotopic composition of the source rock (i.e.
the proportion of evaporites and silicate rocks; Rose et al., 2000). While some studies
have suggested that the isotopic composition of rainfall within the catchment area
may be an important factor controlling the $\delta^{11}B$ in rivers (Rose-Koga et al., 2006),
other studies have shown atmospheric boron to be a secondary control on riverine
boron isotope composition (Lemarchand and Gaillardet, 2006). The source rock also
appears to have limited influence on the $\delta^{7}Li$ composition of rivers and riverine $\delta^{7}Li$
varies primarily with weathering intensity (Kısakűrek et al., 2005; Millot et al.,
2010). The riverine input of calcium to the oceans is controlled by the composition of
the primary continental crust (dominated by carbonate weathering) and a recycled
component, although the relative influence of these two processes is not well
understood (Tipper et al., 2006a). In addition, vegetation may also play a significant
role in the $\delta^{44/40}Ca$ of rivers (Fantle and Tipper, 2014). For Mg, the isotopic
composition of the source rock is important for small rivers, however, lithology is of
limited significance at a global scale in comparison to fractionation in the weathering
environment (Tipper et al., 2006b). Given the lack of evidence of source rock as a
dominant control on the isotopic composition of rivers, here we focus on some of the
possible causes for changes in the isotopic composition and/or flux of riverine input
over the Neogene.
In this regard, of the four elements discussed here, the Li isotopic system is the most
extensively studied. Indeed, the change in $\delta^{7}Li_{sw}$ has already been attributed to an
increase in the $\delta^7 Li_{sw}$ composition of the riverine input (Hathorne and James, 2006;
Misra and Froelich, 2012). The causes of the shift in $\delta^7 Li$ riverine have been variably
attributed to: (1) an increase in incongruent weathering of silicate rocks and
secondary clay formation as a consequence of Himalayan uplift (Misra and Froelich,
2012; Li and West, 2014), (2) a reduction in weathering intensity (Hathorne and
James, 2006; Froelich and Misra, 2014; Wanner et al., 2014), (3) an increase in
silicate weathering rate (Liu et al., 2015), 4) an increase in the formation of
floodplains and the increased formation of secondary minerals (Pogge von
Strandmann and Henderson, 2014) and (5) a climatic control on soil production rates
(Vigier and Godderis, 2015). In all five cases the lighter isotope of Li is retained on
land in clay and secondary minerals. A mechanism associated with either an increase
in secondary mineral formation or the retention of these minerals on land is also
consistent across Mg, Ca and B isotope systems. For instance, clay minerals are
preferentially enriched in the light isotope of B (Spivack and Edmond, 1987; Deyhle
and Kopf, 2004; Lemarchand and Gaillardet, 2006) and Li (Pistiner and Henderson,
2003) and soil carbonates and clays are preferentially enriched in the light isotope of
Ca (Tipper et al., 2006a; Hindshaw et al., 2013; Ockert et al., 2013). The formation of
secondary silicate minerals, such as clays, is assumed to preferentially take up the
heavy Mg isotope into the solid phase (Tipper et al., 2006a; Tipper et al., 2006b;
Pogge von Strandmann et al., 2008; Wimpenny et al., 2014), adequately explaining
the inverse relationship between $\delta^{11}B_{sw}$ and $\delta^{26}Mg_{sw}$. Consequently the increased
formation or retention on land of secondary minerals would alter the isotopic
composition of the riverine input to the ocean in the correct direction to explain the
trends in all four isotope systems through the late Neogene (Fig. 13). While the
relationships between the different isotope systems discussed here suggest a common
control, the influence of carbonate and dolomite formation on Ca and Mg isotopes are
also likely to have played a significant role in the evolution of these isotope systems
(Tipper et al., 2006b; Fantle and Tipper, 2014). Consequently a future model of
seawater chemistry evolution through the Neogene must also include these additional
factors. Further exploration is also needed to determine the influence of residence
time on the evolution of ocean chemistry. Nonetheless, given the similarities between
the geochemical cycles of B and Li, and despite the large difference in residence time

714 (Li = 1 million years, B = 11-17 million years), the correlation between these two

715 records is compelling and would no doubt benefit from additional study.


## 4 Conclusions


718 Here we present a new $\delta^{11}B_{sw}$ record for the Neogene based on paired planktic-

719 benthic $\delta^{11}B$ measurements. Our new record suggests that $\delta^{11}B_{sw}$ (i) was ~ 37.5 ‰ at

720 the Oligocene-Miocene boundary, (ii) remained low through the middle Miocene,

721 (iii) rapidly increased to the modern value between 12 and 5 Ma, and (iv) plateaued at

722 modern values over the Plio-Pleistocene. Despite some disagreements, and different

723 uncertainties associated with each approach, the fact that our new record, and both of

724 the published data based reconstructions capture the first-order late Miocene $\delta^{11}B_{sw}$

725 rise suggests that consensus is building for the $\delta^{11}B_{sw}$ evolution through the Neogene.

726 This emerging view on $\delta^{11}B_{sw}$ change provides a vital constraint required to

727 quantitatively reconstruct Neogene ocean pH, ocean carbon chemistry and

728 atmospheric $CO_2$ using the $\delta^{11}B$-pH proxy. When our new $\delta^{11}B_{sw}$ record is compared

729 to changes in the seawater isotopic composition of Li, Ca and Mg the shape of the

730 records across the Neogene is remarkably similar. For all four systems, riverine input

731 is cited a common and key control of the isotopic composition of the respective

732 elements in seawater. When we compare the isotopic fractionation of the elements

733 associated with secondary mineral formation, the trends in the $\delta^{26}Mg_{sw}$, $\delta^{44/40}Ca_{sw}$

734 $\delta^{11}B_{sw}$ and $\delta^{7}Li_{sw}$ records are all consistent with an increase in secondary mineral

735 formation through time. While a more quantitative treatment of these multiple stable

736 isotope systems is required, the $\delta^{11}B_{sw}$ record presented here provides additional

737 constraints on the processes responsible for the evolution of ocean chemistry through

738 time.

739 **Acknowledgements:**

740 This work used samples provided by (I)ODP, which is sponsored by the U.S.

741 National Science Foundation and participating countries under the management of

742 Joint Oceanographic Institutions, Inc. We thank W. Hale and A. Wuelbers of the

743 Bremen Core Repository for their kind assistance. The work was supported by NERC

grants NE/I006176/1 (G.L.F. and C.H.L.), NE/H006273/1 (G.L.F), NE/I006168/1
and NE/K014137/1 and a Royal Society Research Merit Award (P.A.W), a NERC
Independent Research Fellowship NE/K00901X/1 (M.P.H.) and a NERC studentship
(R.G). Matthew Cooper, J. Andy Milton, and the B-team are acknowledged for their
assistance in the laboratory.  We thank two anonymous reviewers and Philip Pogge
von Strandmann for their helpful suggestions that improved the manuscript.

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

**Figure Captions:**
Figure 1: The oceanic boron cycle. Fluxes are from Lemarchand et al. (2000) and
Park and Schlesinger (2002). Isotopic compositions are from Lemarchand et al.
(2000), Foster et al., (2010) and references therein.
Figure 2: A compilation of published $\delta^{11}B_{sw}$ records. Seawater composition
reconstructed from foraminifera depth profiles (light blue squares and dark blue
cross) from Pearson and Palmer (2000) and Foster et al. (2012) respectively,
numerical modelling (green line), with additional green lines shows ± 1 ‰
confidence interval (Lemarchand et al., 2000), benthic $\delta^{11}B$ (purple diamonds and
dark purple line showing 5pt moving average is using the fractionation factor of
Klochko et al., 2006, light purple line showing 5pt moving average using an
empirical calibration) from Raitzsch and Hönisch (2013), and halites (orange crosses)

1123 from Paris et al. (2010). The orange crosses in brackets were discarded from the

1124 original study.

1125 Figure 3: Schematic diagram showing the change in pH gradient with a 3‰ change in

1126 $\delta^{11}$B for $\delta^{11}$B$_{sw}$ of a) 39.6‰ and b) 37.5‰. Arrows highlight the different pH

1127 gradients. Note how a $\delta^{11}$B difference of 3 ‰ is translated into different pH gradients

1128 depending on the $\delta^{11}$B$_{sw}$. Calculated using B$_T$= 432.6 μmol/kg (Lee et al., 2010) and

1129 $\alpha_B$= 1.0272 (Klochko et al., 2006). (c) The pH change for a $\delta^{11}$B change of 3 ‰ at a

1130 range of different $\delta^{11}$B$_{sw}$.

1131 Figure 4: Map of study sites and mean annual air-sea disequilibria with respect to

1132 $p$CO$_2$. The black dots indicate the location of the sites used in this study. ODP Sites

1133 758, 999, 926 and 761 used in this study are highlighted with water depth. Data are

1134 from (Takahashi et al., 2009) plotted using ODV (Schlitzer, 2016).

1135 Figure 5: Latitudinal cross-section through the Atlantic showing (a) pH variations;

1136 (b) the $\delta^{13}$C composition. Data are plotted using Ocean Data View (Schlitzer 2016).

1137 pH data are from the CARINA dataset (CARINA group, 2009) and the $\delta^{13}$C data are

1138 from the GLODAP data compilation (Key et al., 2004); (c) pH and $\delta^{13}$C$_{DIC}$

1139 relationships in the modern ocean adapted from Foster et al., (2012). Data are from

1140 all the ocean basins spanning approximately 40$^o$N to 40$^o$S. Because of anthropogenic

1141 acidification and the Suess effect only data from >1500 m are plotted. Also included

1142 in the plot are the data from a transect in the North Atlantic (from 0 to 5000 m) where

1143 the effects of anthropogenic perturbation on both parameters have been corrected

1144 (Olsen and Ninneman, 2010).

1145 Figure 6: $\delta^{11}$B$_{planktic}$, temperature and $\delta^{13}$C$_{DIC}$ estimates for the surface and deep

1146 ocean through the last 23 million years. (a) $\delta^{11}$B$_{planktic}$ surface; (b) $\delta^{11}$B$_{borate}$ deep from

1147 benthic foraminifera (blue) from this study and (green) Raitzsch and Hönisch, (2013).

1148 The error bars show the analytical external reproducibility at 95% confidence for this

1149 study. For the Raitzch & Hönisch (2013) data the error bars represent propagated

1150 uncertainties of external reproducibilities of time equivalent benthic foraminifer

1151 samples from different core sites in different ocean basins;  (c) Mg/Ca based

1152 temperature reconstructions of surface dwelling planktic foraminifera; (d) Deep water

1153 temperature estimates from Cramer et al. (2011); (e) $\delta^{13}$C$_{DIC}$ surface record; (f)

1154 $\delta^{13}$C$_{DIC}$ benthic record. Squares depict ODP Site 999, triangles are ODP Site 758,

diamonds are ODP Site 926, circles are ODP Site 761. Species are highlighted by
colour: Orange are *T. trilobus*, purple *G. ruber*, pink *G. praebulloides*, dark blue
*Cibicidoides wuellerstorfi* and light blue *Cibicidoides mundulus*. The two benthic-
planktic pairs that were removed prior to smoothing are highlighted with arrows.
Figure 7: The output from GENIE sensitivity analysis showing the warm-surface-to-
cold-deep $\Delta$pH-to-$\Delta\delta^{13}$C relationship. A pre-industrial model setup was taken and
perturbations were made to alkalinity inventory, carbon inventory, Antarctic shelf
fresh water flux (Sv), Atlantic-Pacific freshwater flux, S. Lim gas exchange (blocks
air-sea gas exchange south of the stated latitude), remineralisation depth scale (m)
and rain ratio – as described in the methods section. Blue circles depict the $\Delta$pH-to-
$\Delta\delta^{13}$C relationship (where the colours reflect the $CO_2$ level of each experiment) and
red open circles show the root mean square of the regression (RMSE). The green
stars are the $\Delta$pH-to-$\Delta\delta^{13}$C relationship for the control experiment conducted at
292.67 ppm $CO_2$. The green (open) points show the RMSE for this control run.
Inventories are dimensionless (1 is control). For the Atlantic-Pacific FWF 1 is
equivalent to 0.32 Sv. The alkalinity and carbon inventory experiments are very
extreme and inconsistent with geologic evidence.
Figure 8: The output from sensitivity analysis of the relationship between pH gradient
and $\delta^{13}$C gradient from the 13500 run CYCLOPS ensemble (see text for model
details). Panel (a) shows the mean gradient when the result from all 18 ocean boxes
are included in the regression. Panel (b) shows only the boxes from the low latitude
ocean from all basins and (c) shows the regression from only North Atlantic low
latitude boxes. Note the lower $\Delta$pH/$\Delta\delta^{11}$B slope at the lower latitudes due to the
effect of temperature. The 0.201 line in each panel is the mean gradient when all the
ocean boxes are included in the regression.
Figure 9: The pH gradient between surface and deep through time calculated from the
$\delta^{13}$C gradient and using a flat probability derived from the low latitude ensemble
regressions from the CYCLOPS model. The modern pH gradients at each site are
also plotted.
Figure 10: The calculated $\delta^{11}B_{sw}$ from the benthic-planktic $\delta^{11}$B pairs using a pH
gradient derived from $\delta^{13}$C . The uncertainty on each data point is determined using a
Monte Carlo approach including uncertainties in temperature, salinity, $\delta^{11}$B and the
pH gradient (see text for details). Data are plotted as box and whisker diagrams
where the median and interquartile range as plotted in the box and whiskers show the
maximum and minimum output from the Monte Carlo simulations. The line of best
fit is the probability maximum of a LOWESS fit given the uncertainty in the
calculated $\delta^{11}B_{sw}$. The darker shaded area highlights the 68% confidence interval and
the lighter interval highlights the 95% confidence interval. The bottom panel shows
box plots of the mean and 2 standard error (s.e.) of 'binning' the individual $\delta^{11}B_{sw}$
measurements into 8 Myr intervals. The middle line is the mean and the box shows
the 2 s.e. of the data points in that bin. The smoothed record is also plotted for
comparison where the line of best fit is the probability maximum of a LOWESS fit
given the uncertainty in the calculated $\delta^{11}B_{sw}$. The darker shaded area highlights the
68% confidence interval and the lighter interval highlights the 95% confidence
interval. The black dot is the modern value of 39.61 ‰ (Foster et al., 2010).
Figure 11: The $\delta^{11}B_{sw}$ curve calculated using the variable pH gradient derived from
$\delta^{13}C$. The median (red line), 68% (dark red band) and 95% (light red band)
confidence intervals are plotted. Plotted with a compilation of published $\delta^{11}B_{sw}$
records. Seawater composition reconstructed from foraminifera depth profiles (light
blue squares and dark blue cross) from Pearson and Palmer (2000) and Foster et al.
(2012) respectively, numerical modelling (green line), with additional green lines
shows ± 1 ‰ confidence interval (Lemarchand et al., 2000) and benthic $\delta^{11}B$ (purple
diamonds and dark purple line showing 5pt moving average is using the fractionation
factor of Klochko et al., 2006, light purple line showing 5pt moving average using an
empirical calibration) from Raitzsch and Hönisch (2013). All the published $\delta^{11}B_{sw}$
curves are adjusted so that at t=0, the isotopic composition is equal to the modern
(39.61 ‰).
Figure 12: a) The $\delta^{11}B_{sw}$ curve from this study plotted with other trace element
isotopic records. On the $\delta^{11}B_{sw}$ panel the darker shaded area highlights the 68%
confidence interval and the lighter interval highlights the 95% confidence interval),
$\delta^{26}Mg_{sw}$ record from Pogge von Strandmann et al. (2014) (error bars are ± 0.28 ‰
and include analytical uncertainty and scatter due to the spread in modern *O. universa*
and the offset between the two analysed species), $\delta^{44/40}Ca_{sw}$ record from Griffith et al.
(2008) (error bars show 2 σ uncertainty) and $\delta^7Li_{sw}$ record from Misra and Froelich
(2012) (error bars show 2 σ uncertainty). Blue dashed lines show middle Miocene
values, red dashed lines highlight the modern.
Figure 13: Crossplots of the records of $\delta^{11}B_{sw}$ using a variable pH gradient derived
from $\delta^{13}C$ (error bars show 2 σ uncertainty) with $\delta^{44/40}Ca_{sw}$ from Griffith et al. (2008)
(error bars show 2 σ uncertainty), $\delta^{7}Li_{sw}$ from Misra and Froelich (2012) (error bars
show 2 σ uncertainty) and $\delta^{26}Mg_{sw}$ from Pogge von Strandmann et al. (2014) (error
bars are ± 0.28 ‰ and include analytical uncertainty and scatter due to the spread in
modern *O. universa* and the offset between the two analysed species). The colour of
the data points highlights the age of the data points where red = modern and blue =
23 Ma.
Table 1: CYCLOPS model parameter values defining the ensemble of 13,500
simulations.
Table 2: Uncertainty inputs into the Monte Carlo simulations to calculate $\delta^{11}B$. The
sources of uncertainty are also added. All uncertainty estimates are 2σ.
Table 3: The average $\delta^{11}B$, $\delta^{26}Mg$, $\delta^{44/40}Ca$ and $\delta^{7}Li$ composition of major fluxes into
and out of the ocean. Colour coding reflects the relative importance of each the
processes (darker shading reflects greater importance). The colour coding for boron is
based on Lemarchand et al. (2000) and references therein, lithium from Misra and
Froelich (2012) and references therein, magnesium from Tipper et al. (2006b) and
calcium from Fantle and Tipper (2014) and Griffin et al. (2008) and references
therein. The isotopic ratio of each process is: (a) Lemarchand et al. (2000) and
references therein; b) Misra and Froelich (2012) and references therein; (c) Burton
and Vigier (2012); (d) Tipper et al. (2006b); e) Wombacher et al. (2011); f) includes
dolomitisation; g) removal through hydrothermal activity; h) Griffith et al. (2008); i)
Fantle and Tipper (2014) and references therein; j) dolomitisation may be an
important component of the carbonate flux. Modern $\delta^{26}Mg_{sw}$ and $\delta^{11}B_{sw}$ from Foster
et al. (2010), $\delta^{7}Li_{sw}$ from Tomascak (2004). The $\delta^{44/40}Ca$ presented here was
measured relative to seawater and hence seawater has a $\delta^{44/40}Ca_{sw}$ of 0 permil by
definition.

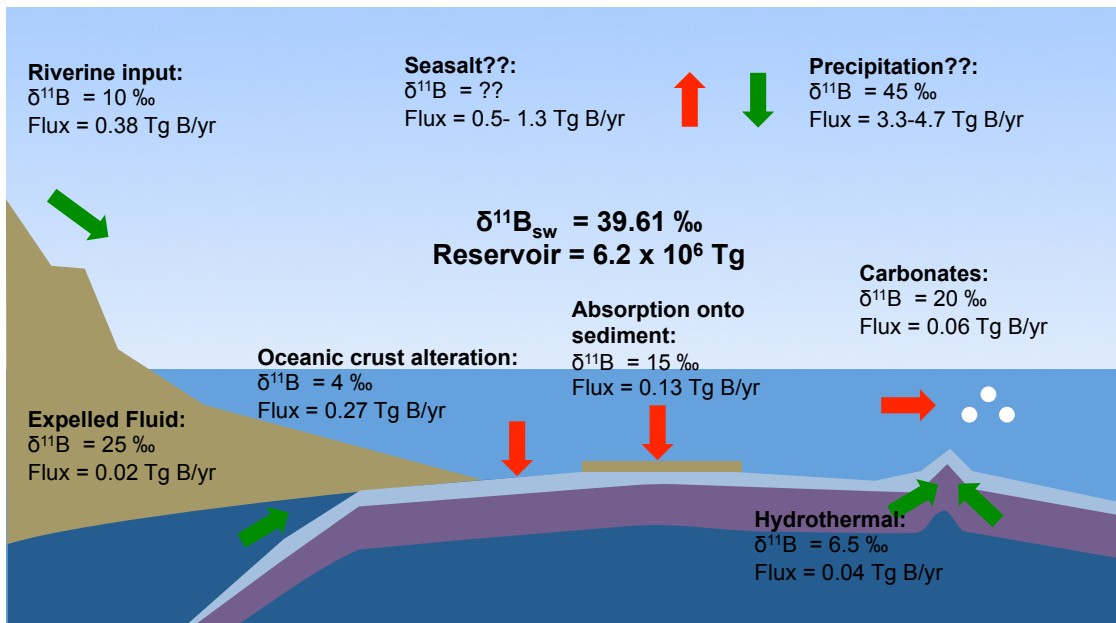

Figure 1

Figure 2

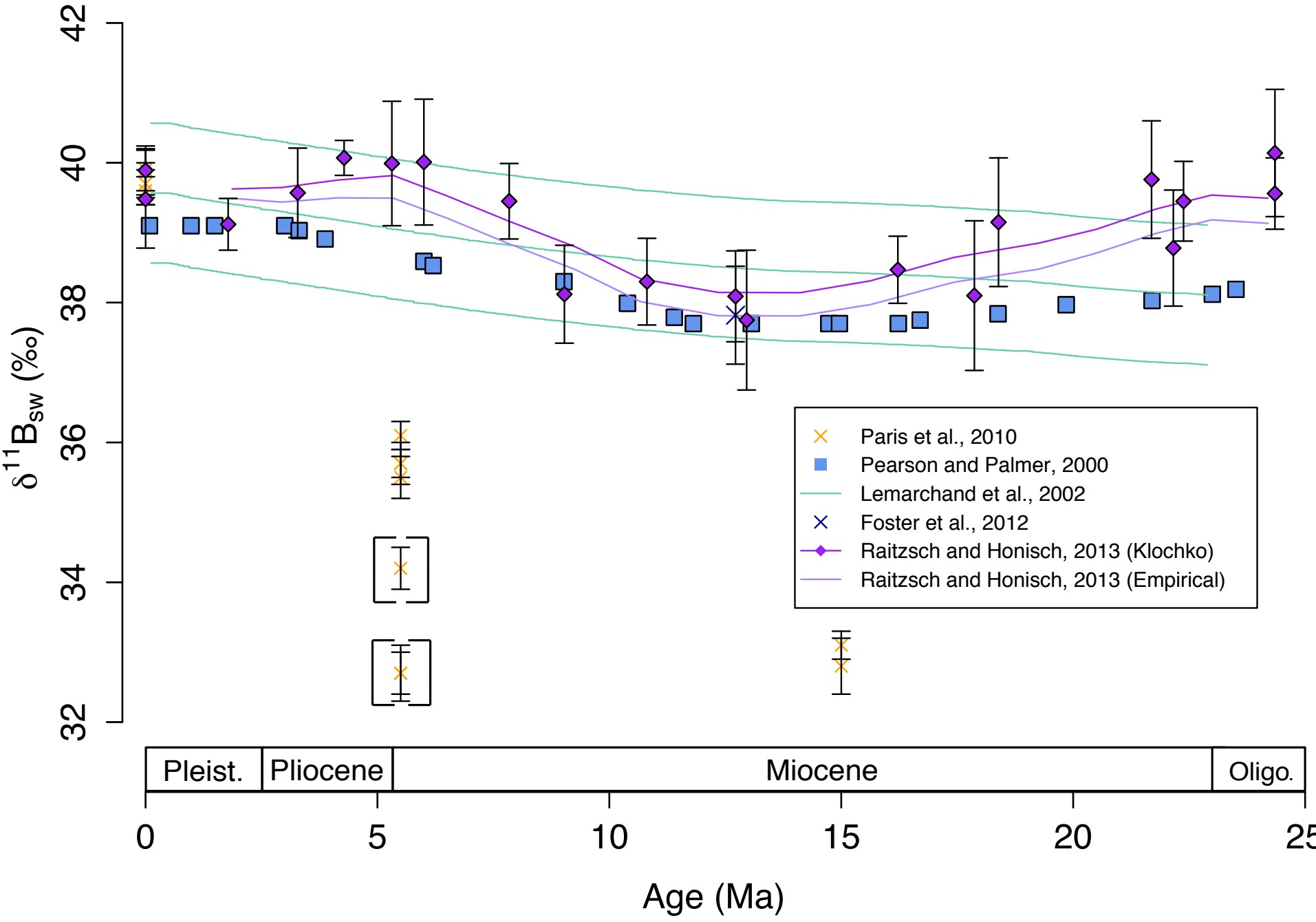

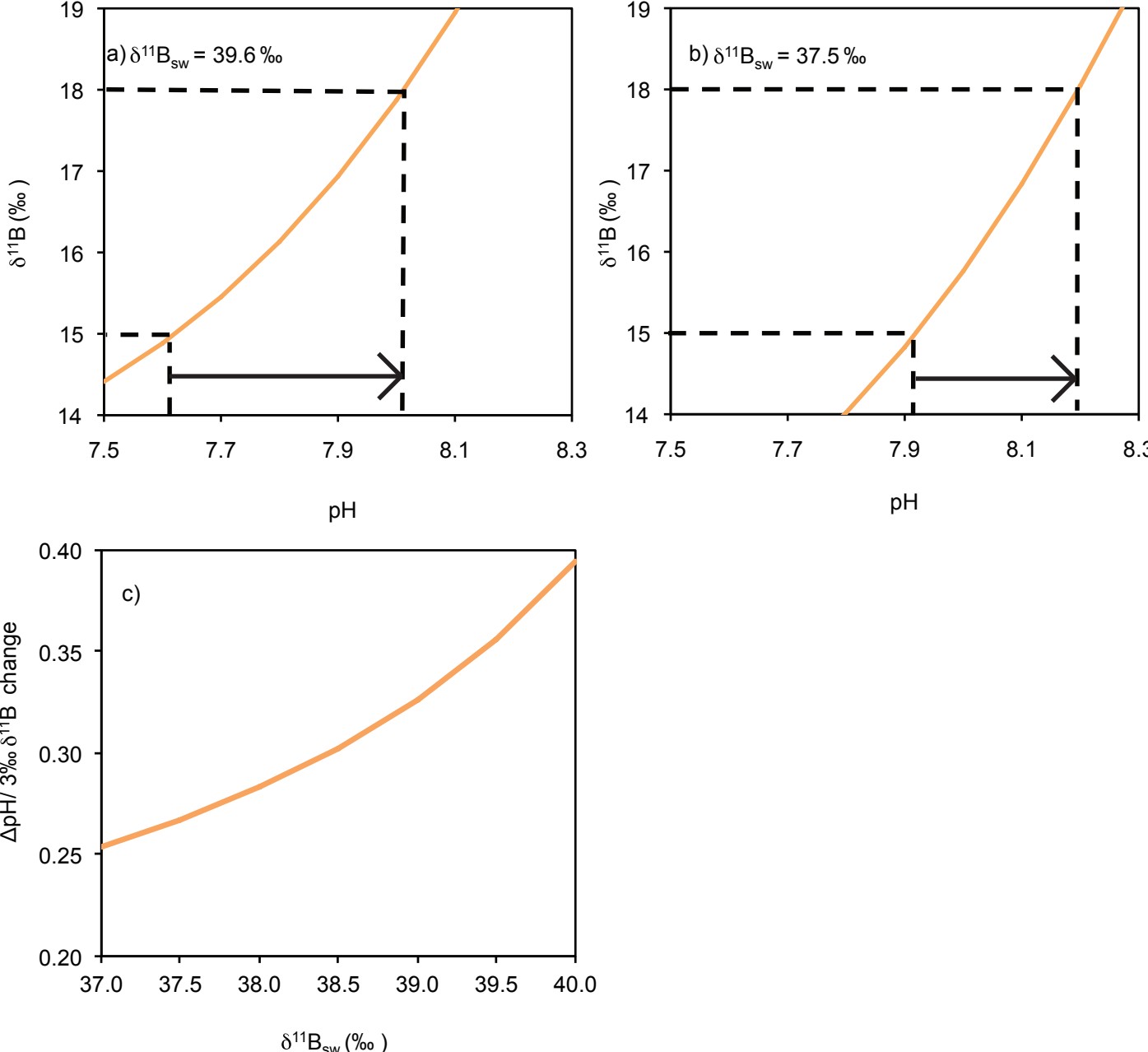

Figure 3

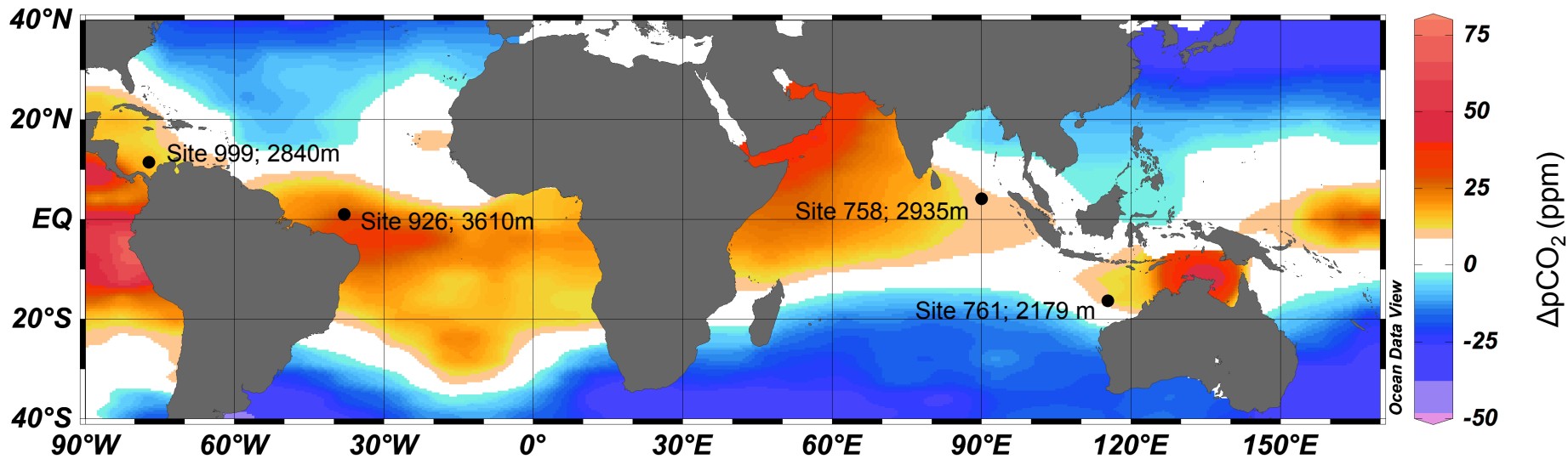

Figure 4

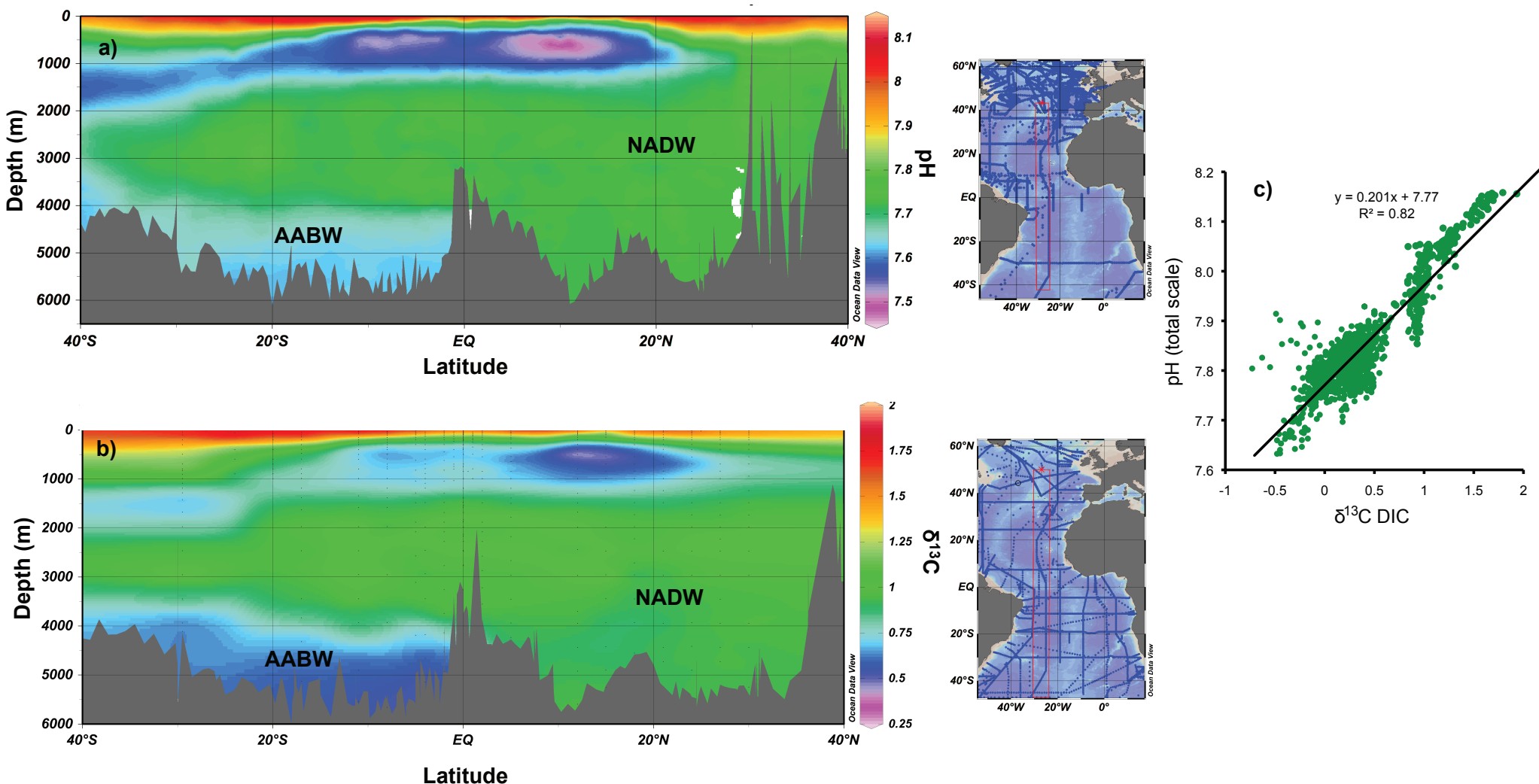

Figure 5

Figure 6

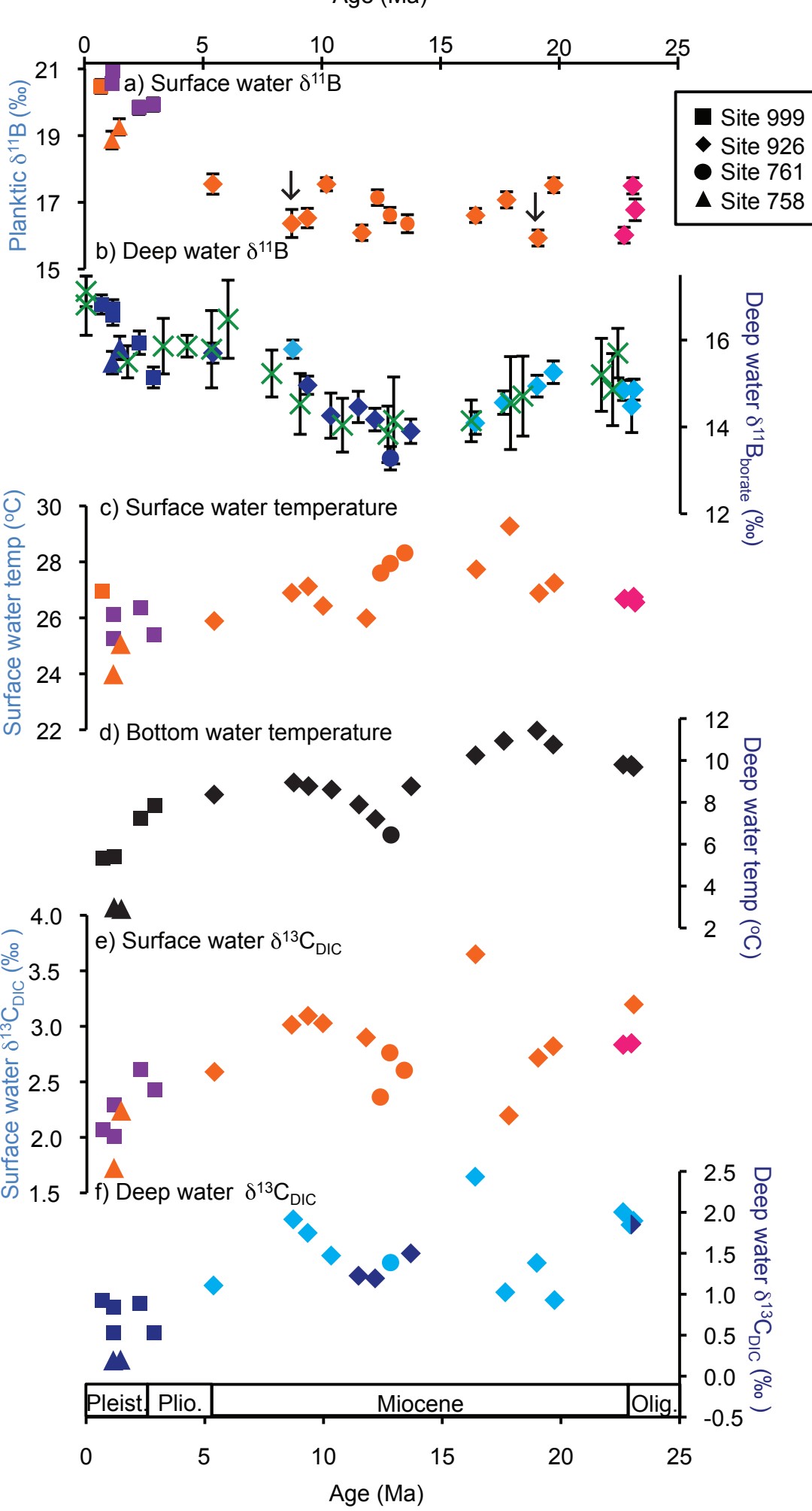

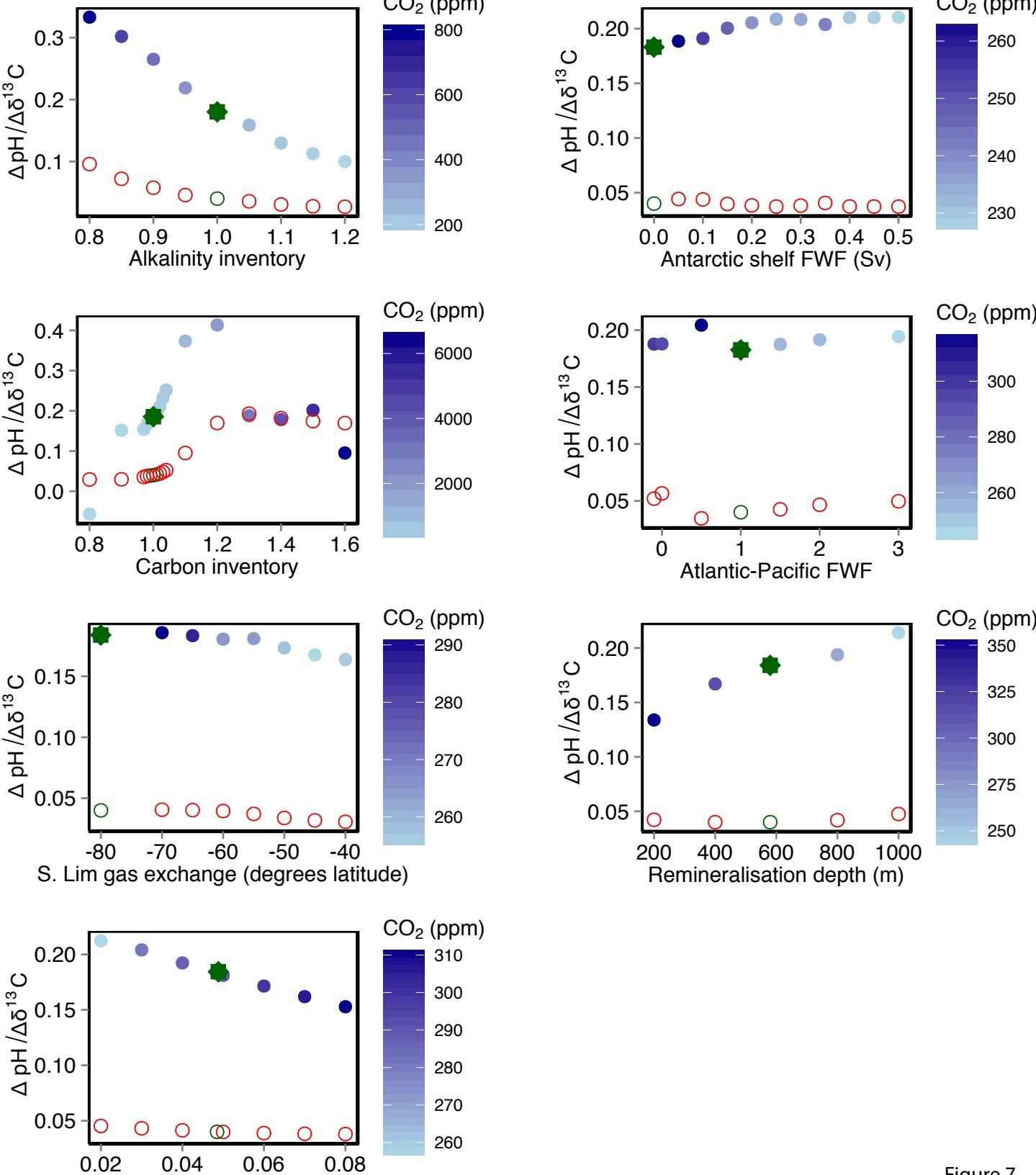

Figure 7

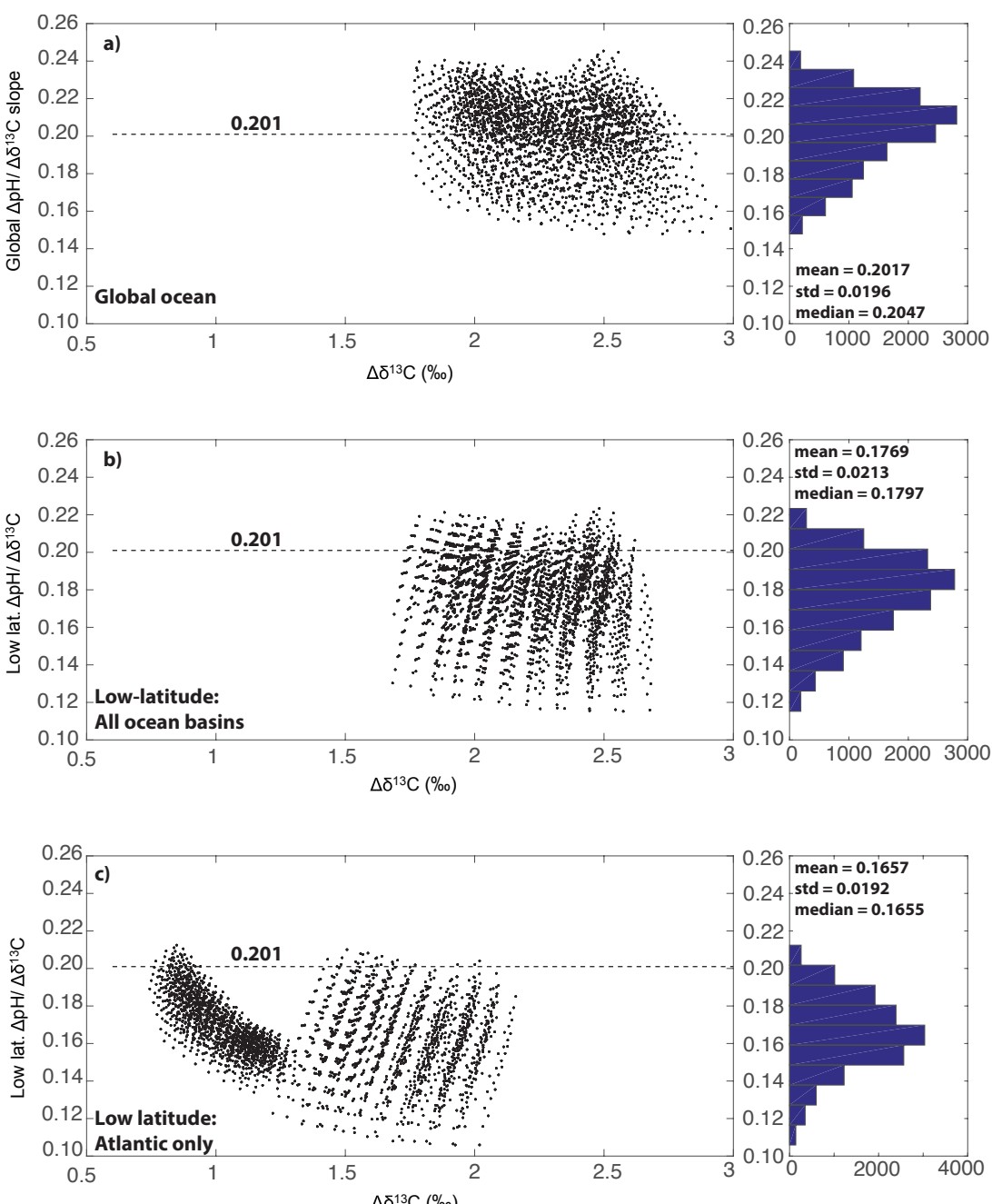

Figure 8

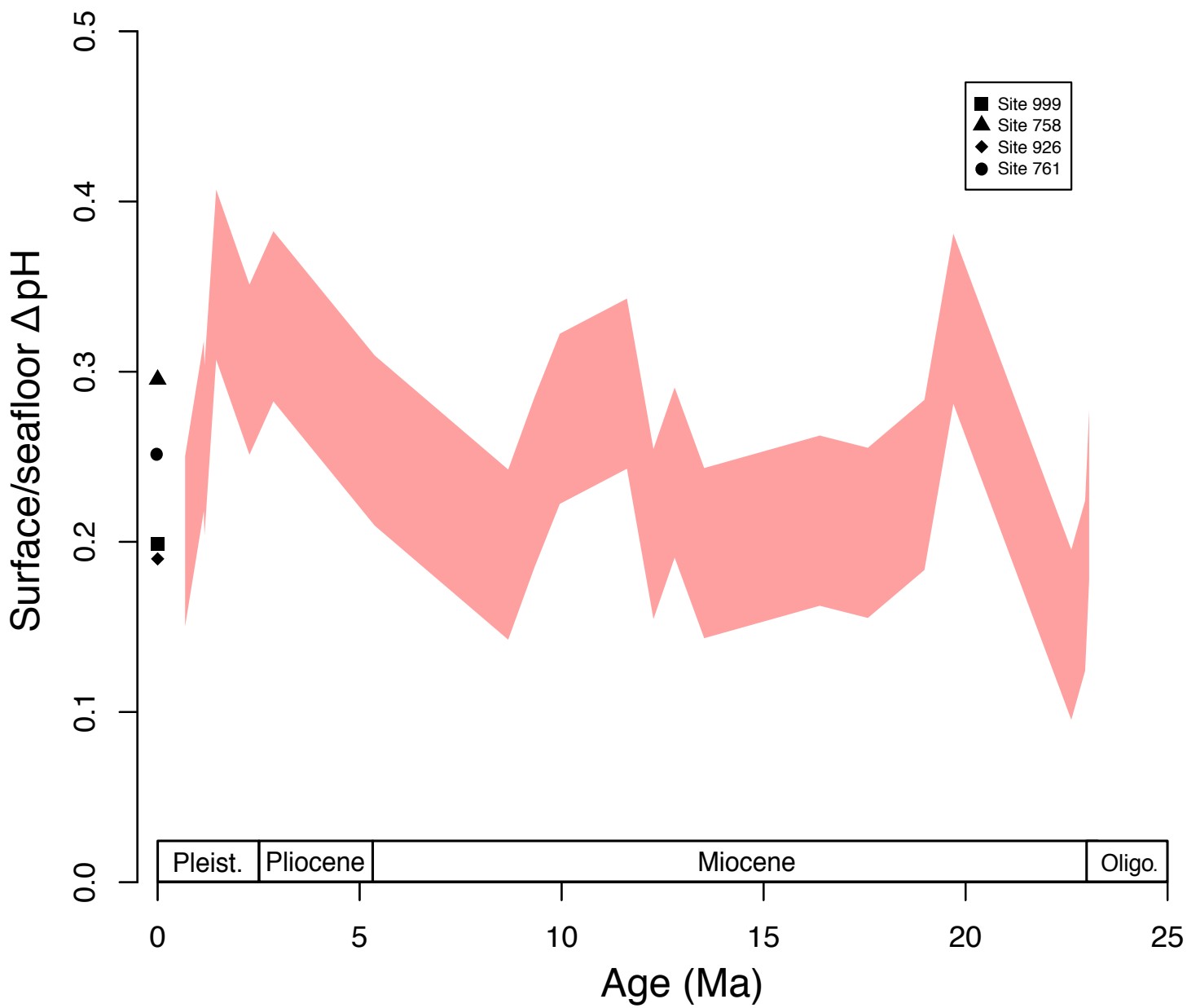

Figure 9

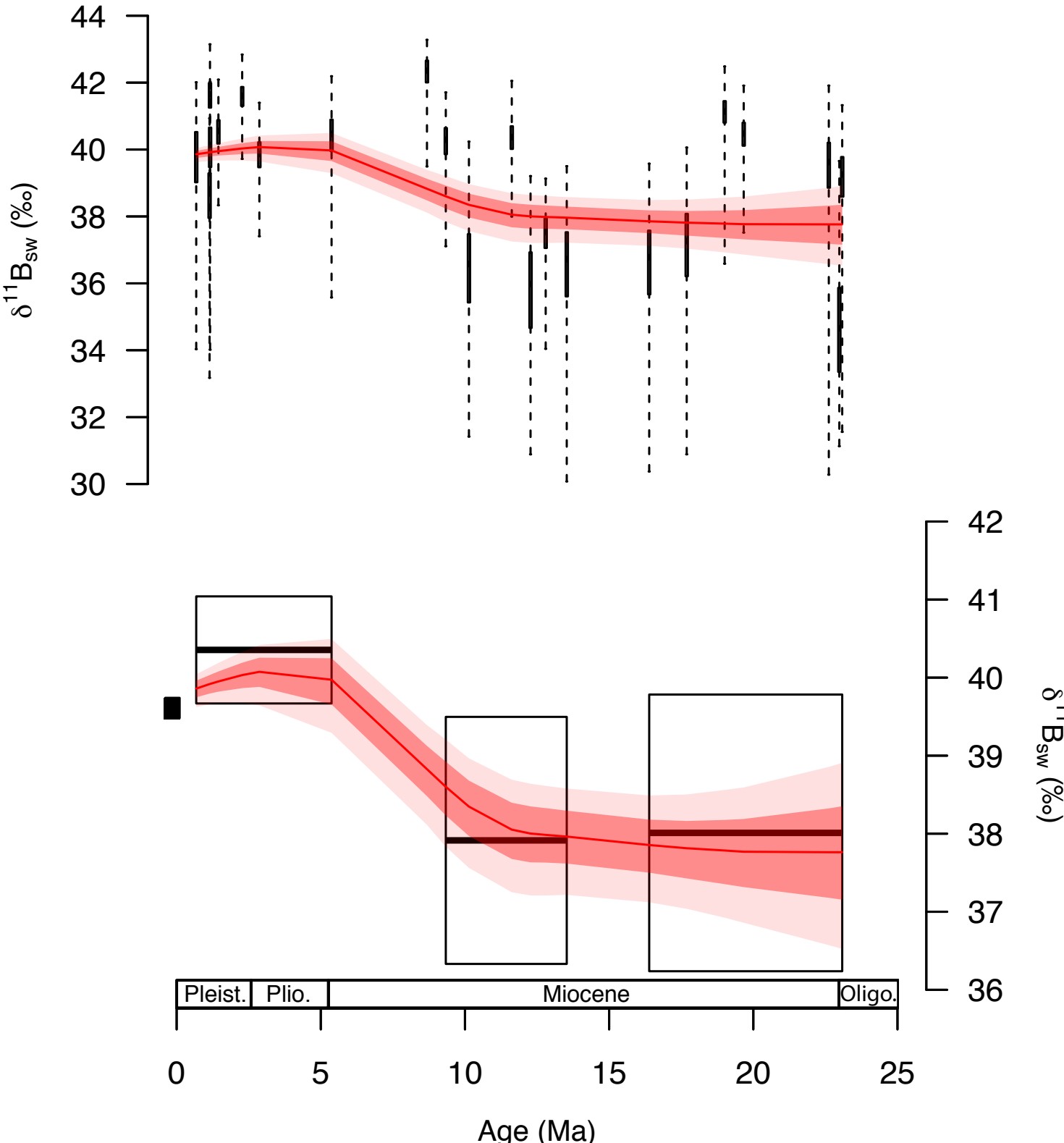

Figure 10

Figure 11

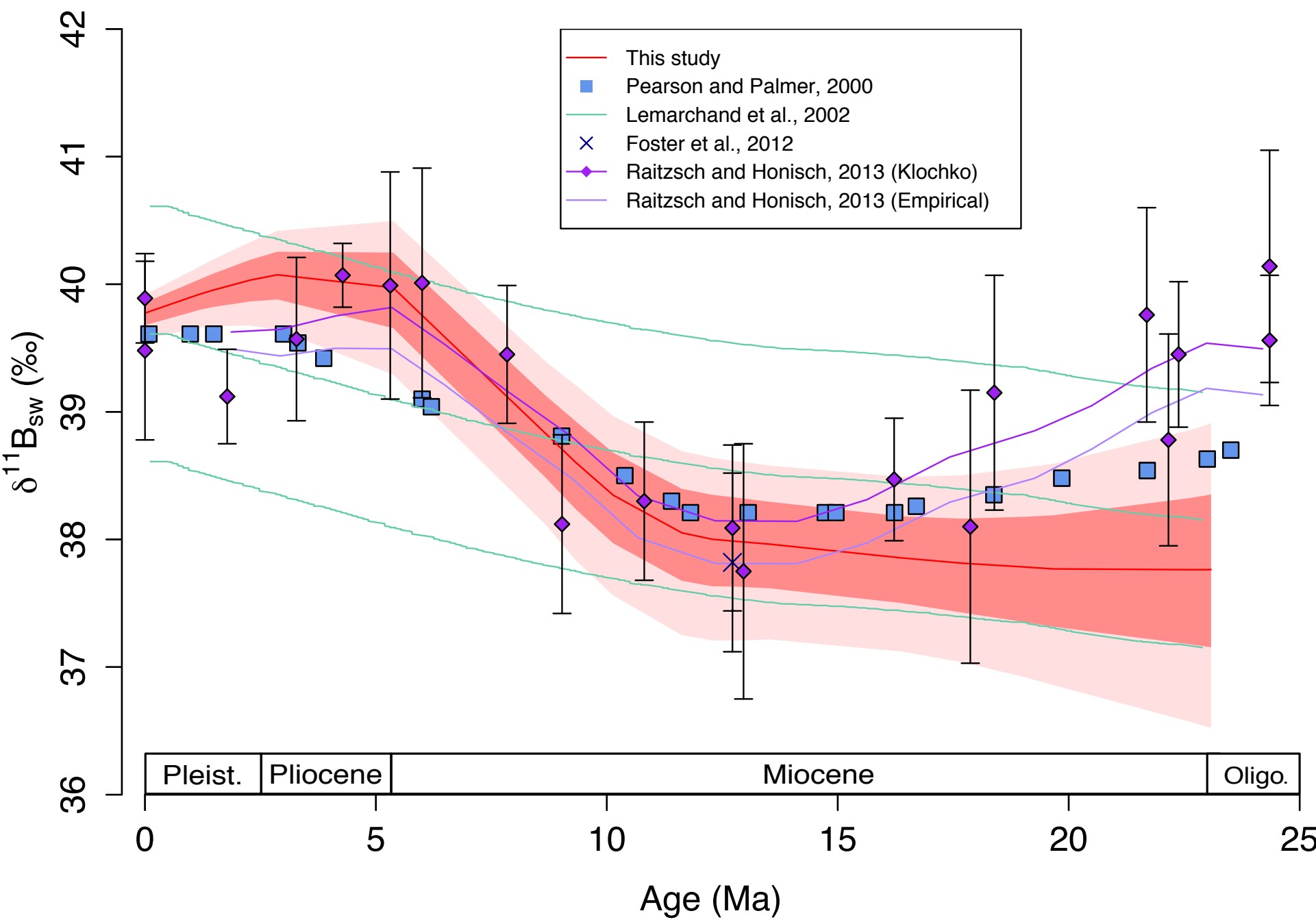

Figure 12

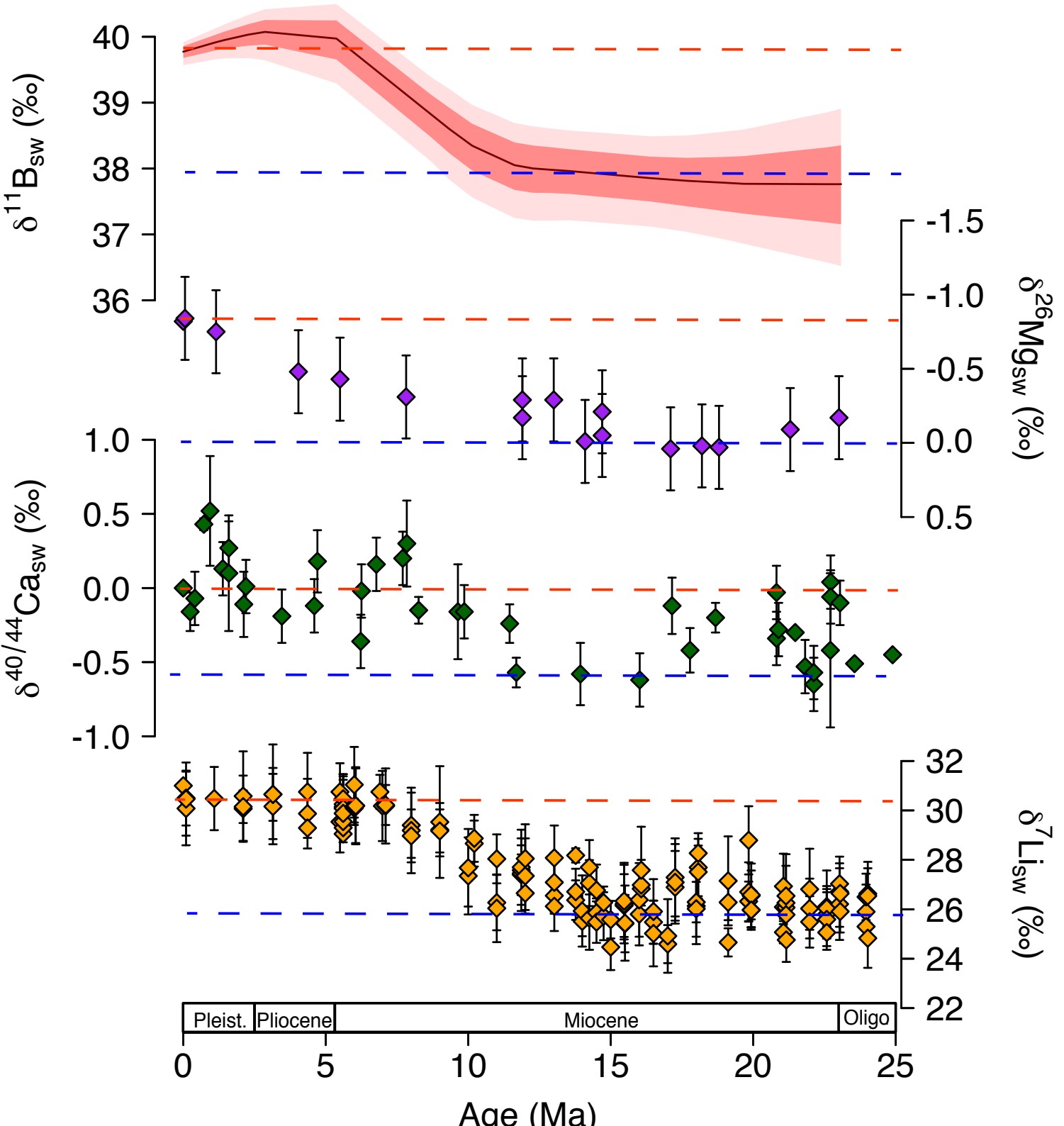

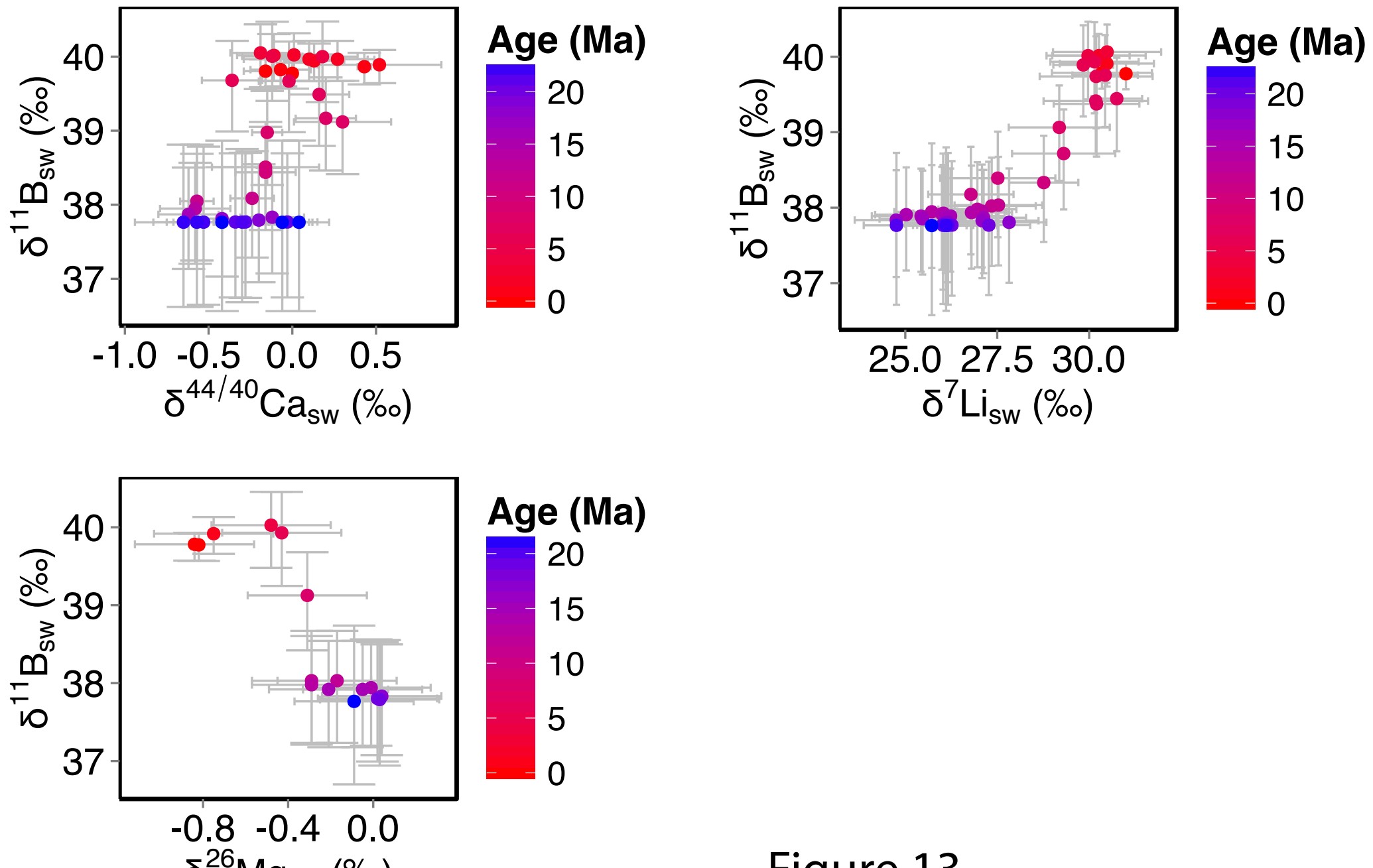

Figure 13

**Table 1.** CYCLOPS model parameter values defining the ensemble of 13,500 simulations*

| Parameter | Description | Values assumed |
|---|---|---|
| PAZ surface phosphate** | unutilized polar nutrient | 1µM, 1.25µM, 1.5µM, 1.75µM, 2µM |
| PAZ vertical exchange** | bottom water formation | 2Sv, 7.75Sv, 13.5Sv, 19.25Sv, 25Sv |
| SAZ surface phosphate** | unutilized polar nutrient | 0.7µM, 0.825µM, 0.95µM, 1.075µM, 1.2µM |
| AMOC circulation scheme** | deep vs. shallow overturning | NADW, GNAIW |
| representative timeslice*** | Age ($[Ca^{2+}]$/CCD); calcium set outright; CCD set via riverine $CaCO_3$ flux using inverse scheme | 0Myr (10.6mM, 4.65km), 9Myr (12.89mM, 4.4km), 11Myr (13.33mM, 4.9km), 16Myr (14.28mM, 4.7km), 18Myr (14.57mM, 4.25km), 20Myr (14.86mM, 4.7km) |
| atm. $CO_2$**** | set via silicate weatherability | 200ppm, 300ppm, 400ppm, 500ppm, 600ppm, 700ppm, 800ppm, 900ppm, 1000ppm |

*= The six parameters assume 5, 5, 5, 2, 9 and 6 values, yielding 13,500 distinct parameter combinations

** = These parameters are intended to span the full range of ocean carbon cycling over late Pleistocence glacial-interglacial cycles, as describe in more detail in Hain et al. (2010)

*** = We selected representative timeslices based on local extrema in the CCD reconstruction of Pälike et al. (2012) and we combine these with appropriate reconstructed calcium concentrations based on Horita et al. (2002). These choices are intended to capture the range of long-term steady state conditions of the open system $CaCO_3$ cycle relevant to our study interval

**** = These atmospheric $CO_2$ levels are chosen to span a range wider than expected for the study interval. Following silicate-weathering-feedback paradigm, long-term $CO_2$ is fully determined by the balance of geologic $CO_2$ sources and silicate weathering, whereby faster acting processes of the open system $CaCO_3$ cycle compensate relative to that $CO_2$ level. All else equal, high $CO_2$ levels, low calcium concentrations and deep CCD correspond to high bulk ocean carbon concentrations (Hain et al., 2015) with many of the individual simulations of this ensemble exeeding 4000µM DIC.

Table 2

| Input parameter | Uncertainty applied | Source of uncertainty estimate |
|---|---|---|
| Surface to sea floor $\Delta$pH | Uniform +/- 0.05 pH units | Plausible range of $\Delta$pH/$\Delta\delta^{13}$C in CYCLOPS and GENIE sensitivity tests; prediction error of linear $\Delta$pH/$\Delta\delta^{13}$C regression in GENIE |
| $\delta^{11}$B measurement | 0.15-0.61‰ | Long-term external reproducibility |
| Temperature | ±2°C | Uncertainty in the Mg/Ca measurement and Mg/Ca-temperature calibration |
| Salinity | ±2 psu | In the absence of a salinity proxy this uncertainty is applied to cover variations through time. |
| Seawater [Mg] | ± 4.5 mmol/kg | following Horita et al., (2002) |
| Seawater [Ca] | ± 4.5 mmol/kg | following Horita et al., (2002) |

# Table 3

| Sources | Isotopic Ratio | | | |
|---|---|---|---|---|
| **Oceanic Inputs** | $\delta^{11}B_{sw}$ 39.61 ‰ | $\delta^{7}Li_{sw}$ 31 ‰ | $\delta^{26}Mg_{sw}$ −0.83 ‰ | $\delta^{44/40}Ca_{sw}$ 0 ‰ |
| Input from hydrothermal | 6.5[a] | 8.3[b] | N/A | −0.96[h] |
| Fluid from accretionary prisms | 25[a] | 15[b] | N/A | N/A |
| Riverine Inputs | 10[a] | 23[b] | −1.09[d] | -1.28[h] |
| Groundwater | N/A | N/A | -0.82[d] | −1.02[i] |
| **Outputs** | | | | |
| Precipitation into carbonates | 20[a] | 29[c] | -3.5[d,e,f] | -1.15[h,j] |
| Ocean crust alteration | 4[a] | 15[b] | -0.83[d,g] | -1.2[h] |
| Absorption onto sediment | 15[a] | 15[b] | ?? | N/A |