# Peer review of "A record of Neogene seawater $\delta^{11}$ B reconstructed from paired $\delta^{11}$ B"

_Climate of the Past, 2015_

## Referee Comment (RC1) · Anonymous Referee #1 · 17 Mar 2016

Greenop et al. present a new d11Bsw-reconstruction over the past 23 million years, using the d13C difference between planktic and benthic foraminifers to predict the pH gradient between the surface and deep ocean, and then applying that pH gradient to infer d11Bsw from paired d11B analyses in planktic and benthic foraminifers from the same core sites and time intervals. The d13C vs pH gradient relationship in the modern is based on shipboard data, and the relative constancy of that relationship over the past 23 million years is estimated from carbon cycle models GENIE and CYCLOPS.

While this study is applaudable in the sense that it presents an elegant alternative approach to previous d11Bsw estimates, I have to admit that I find the practical results disappointing. Whereas the actual reconstruction results in a 10 permil d11Bsw spread

over the past 23 million years, it is only reduced to something more feasible after applying a hefty smoothing factor. However, this does not diminish the effort presented in this study, reconstructing d11Bsw is difficult, and we do not have any direct archives for it, so any new evidence that allows us to home in on a consensus view is highly appreciated. In this regard it is encouraging that the reconstruction finds some significant synergy with previous efforts, but given the large uncertainties presented herein, and the fact that this is yet another indirect estimate, I recommend a somewhat more inclusive final figure of the current state of understanding of this parameter. I personally would feel extremely uncomfortable to apply this reconstruction as the only estimate. To accomplish this goal, several significant revisions should be made to improve the presentation of these new data, but also the presentation of previously published data.

To start from the beginning, the authors use the introduction to discredit all previous d11Bsw reconstructions, except their own (Foster et al. 2012). I encourage the authors to consider that this may be a useful strategy when proposing a study, but at this point their data are available, and given the uncertainty associated with their results, I recommend toning down the arguments, providing solid arguments if some data are to be dismissed, and being less judgmental in the presentation of previously published data. For instance, Paris et al. 2010 presented d11B from modern halites that reflect d11Bsw=39.7 permil, so they did present a reasonable and indeed promising proof of concept, and this should be acknowledged. I agree with the authors that the implications of the paleo-d11Bsw reconstructions based on halites are highly unlikely, and the data do not match other estimates. However, coeval halite data are highly consistent in their d11Bsw, which suggests the data are likely due to a specific and ocean-wide cause, and less likely to a variety of causes, as currently suggested by the authors. I would recommend the authors look at the details of Liu et al. 2000 (their figure 6b) and consider the effect of decreasing [Ca] concentration throughout the Cenozoic. According to Liu's data this could explain a 3-4 permil difference in d11B recorded by the halites, which is somewhat consistent with their much lower d11Bsw estimates. More work clearly needs to be done to study the effect of [Ca] on halite precipitation, but

there is value in the study of Paris et al., and it should not be discredited so lightly.

Similarly, the data of Raitzsch & Hönisch 2013 are presented somewhat selectively in this study, and sometimes they are even misrepresented. Raitzsch & Hönisch 2013 used benthic foraminifers from all ocean basins for their study, to minimize the effect of local pH variations on their d11Bsw estimate. This fact alone deserves some acknowledgment and contrasts with the predominantly Atlantic focus presented by Greenop et al. Raitzsch & Hönisch's uncertainties (as displayed in Greenop's Fig. 5 but omitted in Greenop's Fig. 10) are averages based on 2-4 samples, where each individual sample is based on a monospecific benthic foraminifer sample from different ocean basins. The individual sample uncertainties are in fact comparable to the new individual data presented by Greenop et al., but what is presented in Greenop's Fig. 5b is the propagated uncertainty of all data normalized to the epibenthic C. wuellerstorfi. The statement on "large individual sample uncertainties" (Line 525) is therefore an inappropriate comparison. In contrast, these propagated uncertainties should in fact be shown in Greenop's Fig. 10, because they do reflect the individual data uncertainty associated with the fit (in contrast to Greenop's claim that individual data uncertainty is not accounted for (Line 526/527).

But more importantly, Greenop et al. selected only one of the two solutions presented by Raitzsch & Hönisch, and specifically the one that shows greater discrepancy from their data. The reason why Raitzsch & Hönisch presented two estimates is because (a) the d11B vs pH sensitivity of all marine carbonates calibrated over a wide pH range (>0.5 units) shows a sensitivity that is less than that predicted by the aqueous fractionation factor of Klochko et al. 2006, and (b) because models and experimental data suggest that boron isotope fractionation is affected by temperature (Zeebe 2005, Rustad et al. 2010, Dissard et al. 2012, Kaczmarek et al. 2015, Liu et al. 2015). These two factors were placed on hold by Rae et al. 2011 because they could not evaluate them within their 0.3 pH unit coretop calibration, and Rae et al admitted that their "closeness to inorganic theory may be fortuitous". We know now that the original ar-

guments of Foster (2008) and Rae et al. (2011), that all marine carbonates follow a pH sensitivity similar to aqueous boron isotope fractionation when measured by MC-ICP-MS, is false (e.g. Henehan et al. 2013, Krief et al, 2010), and evidence for a temperature effect under otherwise constant environmental conditions is accumulating (see above). Raitzsch & Hönisch 2013 therefore applied two sensitivities, one similar to empirical carbonate calibrations established over a wide pH range, and the other using the aqueous fractionation after Klochko et al. 2006. Only the latter is shown by Greenop et al., but the first, which is more and more confirmed by newer experimental data as described above, has been omitted. Remarkably, the omitted record matches the estimates of Greenop et al. much better than the one they chose to present in this manuscript. I would urge the authors to include both, and provide appropriate arguments for each of them. There are certainly shortcomings in the approach applied by Raitzsch & Hönisch, as there are with all indirect d11Bsw estimates, but looking for synchrony between different approaches, and learning from differences would be powerful way to improve the paleoceanographic community's confidence in this valuable proxy; insisting only on differences will do the opposite.

Finally, the estimates of Pearson & Palmer (1999, 2000) are dismissed based on their N-TIMS analytical procedure and the use of the boron isotope fractionation factor after Kakihana and Kotaka (1977). I disagree with both of these arguments: Foster et al. (2013) have shown that N-TIMS and MC-ICP-MS report different numbers in absolute terms, but the relative differences between samples are similar, such that they will yield the the same amplitude of d11B and pH change. The comparison of benthic d11B data measured by N-TIMS by Raitzsch & Hönisch (2013) and Greenop in Figure 5 confirms this notion and argues against the analytical procedure creating a systematic bias. Foster and his colleagues have discredited N-TIMS analyses long enough with this unfounded argument, and it should finally be put to rest. Similarly, the d11B vs. pH sensitivity implied by the Kakihana factor actually matches the sensitivity of empirical carbonates very well, both using N-TIMS (e.g. Sanyal et al. 1996, 2001, Hönisch et al. 2004) and MC-ICP-MS (e.g., Krief et al. 2010, Henehan et al.

2013). So even if the Kakihana fractionation factor incorrectly describes boron isotope fractionation in seawater, it accidentally describes marine carbonates, and in particular planktic foraminifers very well. There may be analytical issues with the specific N-TIMS method applied by Palmer, but we will not be able to evaluate this because that particular method is no longer applied in any labs reconstructing pH from marine carbonates. The only reasonable and significant argument that can and must be applied for the data of Pearson and Palmer (1999, 2000) is that they did not know about vital effect offsets between different foraminifer species at that time, and consequently applied the same calibration curve for a variety of them. Given what we know about vital effects on d11B today, that is a fundamental shortcoming of those studies, and I agree that they should be considered with caution on this basis.

Site locations: A map should be provided with the locations of the studied core sites, including a justification why these sites lend themselves to this particular study. Their water depths should be mentioned, and the preindustrial d13C and pH gradient for each core site should be added to Figure 4c, so the reader can evaluate how well these sites fit the fundamental premise of this study, i.e. that these sites can be used to reconstruct the d13C and pH gradients over the past 23 million years. It is particularly noteworthy here that sites 758 and 761 are from the Indian Ocean and therefore differ in their hydrography from Atlantic sites 926 and 999. Importantly, this difference is somewhat dismissed in the later part of the study, where the slope specific to Atlantic sites is selected and applied to estimate the pH gradients for all sites.

Boron isotope reconstructions: I have been looking for the original d11B data measured for each sample, but cannot find them in tabular form and Fig. 5 only displays them in the form of d11B borate, i.e. already one step removed from the original data. It is important to list and display the original data and their uncertainties. Have different foraminifer species been measured on the same samples, to cross-calibrate ancient species versus modern G. ruber? This is an important aspect for evaluating the appropriateness of the applied d11B vs d11b borate calibrations.

Modeling the d13C vs pH relationships: It would be nice to provide the exact parameters and their ranges applied for the model estimates in a table, so any reader could repeat these experiments without having to contact the authors. Also, references should be provided for all estimates, and please reconsider some of the wording. For instance, it is unclear whether atmospheric pCO2 was applied once at 200 ppm and once at 1000ppm, or across the entire range (Line 368). Please list studied ranges for pH, CaCO3 saturation states and the biological pump in tabular form.

The slopes of the d13C- pH relationships (Fig. 7) need to be better explained. I take from the text that the global estimate includes all data, then data are systematically restricted first to the low latitudes, and then to the low latitude Atlantic only. That should make the data density sparser from step to step, but while this seems somewhat correct for the dense cloud between 1.7-3 permil d13C, the low-latitude and Atlantic-only subsets (b and c) display data that are not included in Fig. 7a. Please explain where these data are coming from, and why they are not included in the complete data set (7a)? To allow the reader to better evaluate the consequence of this restriction, the data in Fig. 4c should be colored to highlight global low latitude and low latitude Atlantic data, and the respective regressions should be shown. Furthermore, it needs to be evaluated how the choice of slope affects the pH gradient estimates for the Indian Ocean cores. Specifically, the two 758 samples at around 2 Ma appear to have a large effect on the Pleistocene d11Bsw estimates, in particular after the smoothing function is applied. It seems like the Indian Ocean sites should be estimated using either the global low latitude slope (Fig. 7b) or a slope specific to the Indian Ocean, but not the one for the low latitude Atlantic.

Smoothing the d11Bsw estimates: Fig. 9 demonstrates that irrespective of the pH gradient chosen, the data uncertainty encompasses the entire record, and only application of the smoothing function allows to discern a d11Bsw trend. Greenop et al. argue that the smoothing function is justified because Lemarchand et al. 2002 calculated the d11Bsw rate of change is $\sim$0.1 permil per million years (line 398 in this manuscript).

However, Lemarchand et al. also estimated up to 0.6‰ d11Bsw change per million years for the Cretaceous, suggesting that the 0.1 permil/Myr change is not the limit and depends on the model parameterization. Remarkably, Greenop et al. eventually argue that their record shows more variability than the model estimates of Lemarchand et al., and suggest that the model relevant boron inputs and outputs are not fully understood (Line 532/533). This means that Greenop et al want it both ways – they use the small rate of change change to justify their hefty smoothing, but then cast doubt on Lemarchand's estimates when they show less variation than these new estimates. The smoothing obviously cannot be justified with Lemarchand's estimates, and whatever argument is brought up must be independent of an alternative d11Bsw estimate, or the argument becomes circular. Regardless, given that the smoothing function reduces the potential d11Bsw ranges estimated from this approach from 42-32 permil to ∼40-37 permil implies that application of the smoothing function largely dismisses the actual data, and therefore the heart of the reconstruction. This data treatment may be appropriate for a dataset with hundreds of data, but with just 20 data from not even a handful of sediment cores I am having a hard time finding the smoothed estimate convincing.

Similarly, Greenop et al. use the variability observed in their d13C and pH gradient estimates to argue the variable pH gradient estimates give a better result than the constant pH gradient approach. With error bars that essentially overlap for all samples, and two estimates randomly discarded, I am rather uncomfortable believing any arguments for preferring one estimate over the other. Given the large uncertainty of this approach, I would recommend Greenop et al. plot all their estimates, and do not bury half of them in the supplement. This entire range of possible d11Bsw estimates should be shown in Fig. 10, and not a subjective subset. Ideally, Figure 10 would present all available estimates not only from this study, but also from others (Raitzsch & Hönisch 2013, Lemarchand et al. 2000); and then focus the discussion on synergies between different data sets, find a consensus d11Bsw trend, and discuss whether the synergies can help us to further improve our understanding of the boron isotope proxy.

Comparison to other isotope systems: This is a nice comparison and it looks convincing after the true data variability has been discarded, but it would be nice to expand this analysis and include all possible d11Bsw scenarios in it, including the actual data before smoothing. Suggesting weathering and catchment patterns from correlating the smoothed record to actual data seems premature given the significant uncertainty of the original (unsmoothed) d11Bsw estimates.

Line 28: sp: consistent Line 46: sp: Martínez-Botí Line 54: please add Lemarchand et al. 2000. They were the first to note the change in d11Bsw and compare it to the analytical uncertainty of d11B Line 66: please add a reference for the d11B of B inputs Line 68: please specify whether mineral precipitation or rainfall is meant by "precipitation" Line 76: please replace Zeebe & Wolf-Gladrow by Hemming & Hanson 1992, the CO2 book only reports the earlier suggestion Line 83: please add Lemarchand et al. 2000 Line 90/91: please rephrase "currently...at present" Line 115/116 please rephrase: a small subset of the CO2 data used by Ridgwell et al. 2005 is derived using the boron isotope-pH proxy, leading to some circularity in the d11Bsw method. However, the pH estimates applied by Raitzsch & Hönisch (2013) from Tyrrell & Zeebe (2004) are based on GEOCARB, and the circularity problem does not apply. Line 216: Why is it important to specify "at 0.7 Ma" here? Please explain or rephrase Line 220: please specify that you assume there is no temperature effect on d11B. Ideally you should calculate d11B borate with and without the temperature effect, and estimate the associated uncertainty on the d11Bsw estimate Line 228/252: please specify whether trace element and d13C analyses were performed on the same foraminifer species and size classes as studied for d11B Line 231: please define "H" Line 249-251/281-286: please specify the influence of temperature on the d13C-DIC estimate Line 259: which benthic species was analyzed for d13C? Lines 261-265: symbiotic planktic foraminifers record heavier d13C than surrounding seawater, so the vital effect adjustment for estimating d13C-DIC should be negative, not positive Line 269: sp: outline Line 273: leads to a broad Lines 295/296: please rephrase, e.g. "CaCO3 dissolution increases CO32- and pH". The sentence is difficult to read. Line 302: it would be nice to anticipate at this

point which of the described effects will be considered in the remainder of the paper Line 304: please define the Suess effect and provide a reference Line 306: please add the uncertainty of the slope Line 399: please analyze and explain possible reasons for the small d11B differences between the planktic and benthic forams excluded here. Is there any indication/reason why these data are anomalous? Line 416/417: please rephrase to include that the benthic d11B record is a function of both pH and d11Bsw Fig. 4 caption: Please explain the difference between open and closed symbols, and different symbol shapes in 4c. It would be advisable to replot this figures according to recommendations made above, instead of just copying it from Foster et al. 2012. Line 1007: sp: Suess. To the best of my knowledge Dr. Seuss never wrote about carbon isotopes. Line 1015: please explain what is meant by "paired measurements" Line 1026: sp: colours reflect Line 1033: sp: tests Line 1049: insert period after "details)" Fig. 2: Please remove the two lower end Miocene d11Bsw estimates of Paris et al., which were identified as outliers by the original paper. Including them makes the record look worse than it is, which seems unfair. Fig. 3: It would be nice to insert a horizontal bar or arrow to indicate the smaller pH gradient in b. Fig. 5: Please add a second time axis at the top of the figure for better comparison throughout the figure. Horizontal lines in each panel might also help to estimate values at each time interval. Please also indicate the planktic foram species measured for each data point, e.g. by different color symbols. Are there any data pairs that allow cross-calibration of praebulloides and trilobus? Fig. 8: please indicate core site symbols in this figure. In particular, the large gradient at ∼2 Ma is due to two samples from one site in the Indian Ocean. It is unfortunate to select two samples that are close in time from one site and then no more for the rest of the record. Those samples have the potential to bias the record at that time. Fig. 10: please add both curves from Raitzsch & Hönisch, including their uncertainty estimates Fig. 12: at a minimum, I would suggest to remove the Indian Ocean cores from this estimate. I suspect the data will match much better without those estimates.
* * *

---

## Referee Comment (RC2) · Anonymous Referee #2 · 2 Apr 2016

this work aims at proposing a new approach to determine the B isotopic composition of the oceans over the past 25 Ma. This is one of the three pillars of the atmospheric pCO2 reconstruction over geological timescales from B isotopes in marin carbonates. The two other pillars are the pH dependence of the B isotopic fractionation and the change of seawater alkalinity. Even if the few d11Bsw reconstructions published so far all point to a slow increase during the last tens of Ma, they are all based on models and assumptions that are sometimes difficult to ascertain. In that, any effort to provide new and independent approach is the most welcome for the scientific community. Since it is definitely a tricky task, any effort of developing new approaches has therefore to be acknowledged. The past d11Bsw modeled in this study is based on new B isotopic

[Figure]

Interactive
comment

data on paired planktic and benthic foraminifera. Two scenarios have been tested to derive d11Bsw: the first one assumes a constant pH gradient along the seawater depth-profile and the second one uses d13C data to model possible variation of the seawater pH depth-profile. Even claimed to be a new approach by the authors, using paired foraminifera living at different depths in the water column was first introduced by Pearson and Palmer (1999). The originality of the present work is the attempt to anchor this approach with other data (d13C) and climate model. Finally, the present work ends with a model of evolution of the marine d11Bsw very close and consistent with the previous work, which is already fine, but the large scatter of data (Fig 9) makes possible any d11Bsw value between +42‰ and +32‰. After data smoothing (which statistical meaning being questionable from a so small database), the authors end with a narrow range of possible values which spreading approaches the uncertainty of previous approaches (see fig. 10). This is then a bit disappointing and this is could be easily corrected if the paper starts with a more upfront position and objective of their work in comparison to previous ones. This starts in the abstract: one of the most discussed issue of using B isotopes in carbonates to reconstruct past seawater pH is first the actual relationship between the B isotopic fractionation and the pH. So far, many works have provided "calibration curve" using different foraminifera species, and they all come with differences attributed to "biological effect". Another point is the B speciation in carbonates and the actual process of B incorporation. I personally think that most of the published d11B sw are actually consistent and what we need now is a more accurate model that will narrow the range of possible values. The present work also needs to acknowledge that the approach based on paired foraminifera living at different depths in the water column was first proposed by Pearson and Palmer in 1999! Later, sensitivity tests carried by Pagani et al. (2004) showed that this approach is rather inaccurate.

In Fig. 1 is presented a simplified B cycle in the ocean but, presented as it is, there is a large imbalance between inputs and outputs: the B inputs by precipitation is one order of magnitude higher that the sum of all the others. Then, either this inputs flux is balanced by output by seasalts or evaporation of gaseous B and then this loop is that fast that it must be taken into account, or atmospheric B derives from a continental source and a huge output flux is missing. One possibility is that the atmospheric B content (certainly in gaseous form and not seasalts) derives from anthropic activities, which are irrelevant for reconstructions on geological timescales. Whatever, the explanation is, the B marine cycle like presented in fig. 1 at least useless if not wrong. I would therefore strongly suggest to revise this figure and the corresponding text, in particular lines 66-71 where the atmospheric fluxes are discussed.

There is one thing that we can taken for sure is the very long B residence time in the oceans. This is held by the very high B concentration is seawater (about 500 times more concentrated that river waters). Therefore, whatever the model or indirect determination of the d11Bsw, large and rapid variation of d11Bsw like those mentioned line 86 are unrealistic as long as we do not identify a huge B flux, missed so far, that may have affected the B cycle over geological time scales.

A map of the sampling location would be appreciated

In equation (1), I think that epsilon(biological effect) would be appropriate and add in the text a discussion about the different calibration curves

section 2.2 is a succession of hypothesis and calculation made from a series of embedded models, which is certainly the best way to make estimation with our present knowledge, but a discussion of the possible errors propagated is critical here in a much more detailed and argued way than proposed in section 2.5. In particular, from what data is derived the estimation of $\pm 0.05$ pH of the error made on the d13C-pH relationship (line 381)?

It is difficult to follow the section 3.2. The important information is finally given in the last 5 lines of this section. Please shorten and clarify

This is a bit frustrating to see d11Bsw values and discussion of them only in the last 2

pages (lines 484-537). I would strongly suggest to shorten some of the previous parts and strengthen this last one.

One important point is the validation of the model and the statistical treatment made on the models d11Bsw values. At first sight, fig. 9. looks far from being convincing: a large scatter of the data, which looks like not providing strong new contraints and a significant restriction of the possible data by smoothing the small dataset?

---

## Author Comment (AC1) · 18 May 2016

RESPONSE TO REFEREE #1

Greenop et al. present a new d11Bsw-reconstruction over the past 23 million years, using the d13C difference between planktic and benthic foraminifers to predict the pH gradient between the surface and deep ocean, and then applying that pH gradient to infer d11Bsw from paired d11B analyses in planktic and benthic foraminifers from the same core sites and time intervals. The d13C vs pH gradient relationship in the modern is based on shipboard data, and the relative constancy of that relationship over the past 23 million years is estimated from carbon cycle models GENIE and CYCLOPS.
While this study is applaudable in the sense that it presents an elegant alternative approach to previous d11Bsw estimates, I have to admit that I find the practical results disappointing. Whereas the actual reconstruction results in a 10 permil d11Bsw spread over the past 23 million years, it is only reduced to something more feasible after applying a hefty smoothing factor. However, this does not diminish the effort presented in this study, reconstructing d11Bsw is difficult, and we do not have any direct archives for it, so any new evidence that allows us to home in on a consensus view is highly appreciated. In this regard it is encouraging that the reconstruction finds some significant synergy with previous efforts, but given the large uncertainties presented herein, and the fact that this is yet another indirect estimate, I recommend a somewhat more inclusive final figure of the current state of understanding of this parameter. I personally would feel extremely uncomfortable to apply this reconstruction as the only estimate.
To accomplish this goal, several significant revisions should be made to improve the presentation of these new data, but also the presentation of previously published data.

We thank the referee for the explicitly positive view on our work, and we particularly appreciate the thoughtful and constructive criticism voiced here. We agree that in our writing we were too much focused on shortcomings of previous approaches to reconstruct seawater boron isotopic composition, and in our discussion/conclusion we placed too much emphasis on the differences

between our results and previous reconstructions. We follow the referee's suggestion by significantly revising the manuscript in three specific ways: (1) in our description of the current state of the art we more clearly state that the description of shortcomings serves to motivate the more thorough approach that we describe rather than to invalidate previous results; (2) we include a wider set of reconstruction in our final figure that is central to our discussion; and (3) we now highlight the rather good agreement between our and the results of Raitzsch & Hönisch before discussing the differences. Overall, we have modified the tone of the manuscript with regard to prior work and we emphasize the progress towards a consensus on past seawater boron isotope change that is supported by different approaches, including our own.

To start from the beginning, the authors use the introduction to discredit all previous d11Bsw reconstructions, except their own (Foster et al. 2012). I encourage the authors to consider that this may be a useful strategy when proposing a study, but at this point their data are available, and given the uncertainty associated with their results, I recommend toning down the arguments, providing solid arguments if some data are to be dismissed, and being less judgmental in the presentation of previously published data.
For instance, Paris et al. 2010 presented d11B from modern halites that reflect d11Bsw=39.7 permil, so they did present a reasonable and indeed promising proof of concept, and this should be acknowledged.
We will acknowledge this point in the revised manuscript and as noted above we have changed the tone of the manuscript.

I agree with the authors that the implications of the paleo-d11Bsw reconstructions based on halites are highly unlikely, and the data do not match other estimates. However, coeval halite data are highly consistent in their d11Bsw, which suggests the data are likely due to a specific and oceanwide cause, and less likely to a variety of causes, as currently suggested by the authors.

In the study of Paris et al., (2010) coeval halites for the samples at 5.5 Ma show a spread of 3 per mil for a single location. While the authors of the paper suggest the lower values are attributable to a local fractionation following brine calcium enrichment or acidification, and discard the data, it is difficult to assess to what extent this process has occurred in each sample. However, the results do have utility as a minimum constraint. In the revised manuscript we will further stress this point.

I would recommend the authors look at the details of Liu et al. 2000 (their figure 6b) and consider the effect of decreasing [Ca] concentration throughout the Cenozoic. According to Liu's data this could explain a 3-4 permil difference in d11B recorded by the halites, which is somewhat consistent with their much lower d11Bsw estimates. More work clearly needs to be done to study the effect of [Ca] on halite precipitation, but there is value in the study of Paris et al., and it should not be discredited so lightly.

In the revised manuscript we will expand on reasons for the difference in d11Bsw estimates between halites and other estimates. However, in the case of changing [Ca] concentration, estimates suggest that the concentration of [Ca] has increase by approximately 7 mmol (0.28g/L) (Horita et al., 2002) over the Neogene. This change is very small compared to range of [Ca] explored in the work of Liu et al., 2000.

Similarly, the data of Raitzsch & Hönisch 2013 are presented somewhat selectively in this study, and sometimes they are even misrepresented. Raitzsch & Hönisch 2013 used benthic foraminifers from all ocean basins for their study, to minimize the effect of local pH variations on their d11Bsw estimate. This fact alone deserves some acknowledgment and contrasts with the predominantly Atlantic focus presented by Greenop et al.

We will acknowledge this is the revised manuscript and where appropriate we highlight the good agreement between the two datasets.

Raitzsch & Hönisch's uncertainties (as displayed in Greenop's Fig. 5 but omitted in Greenop's Fig. 10) are averages based on 2-4 samples, where each individual sample is based on a monospecific benthic foraminifer sample from different ocean basins. The individual sample uncertainties are in fact comparable to the new individual data presented by Greenop et al., but what is presented in Greenop's Fig. 5b is the propagated uncertainty of all data normalized to the epibenthic C. wuellerstorfi. The statement on "large individual sample uncertainties" (Line 525) is therefore an inappropriate comparison.

In the revised manuscript we will make it clear where we are referring to the uncertainty in $\delta^{11}B$ measurements (which is comparable to this study) and the uncertainty in $\delta^{11}B$ borate (where the uncertainty is larger than for this study as multiple species are used).

In contrast, these propagated uncertainties should in fact be shown in Greenop's Fig. 10, because they do reflect the individual data uncertainty associated with the fit (in contrast to Greenop's claim that individual data uncertainty is not accounted for (Line 526/527).

In the revised manuscript these propagated uncertainties will be added to the figure. It is important to note, however, the methodologies for propagating the uncertainty in the studies of Greenop et al., and Raitzsch & Honisch are different and consequently not directly comparable.

But more importantly, Greenop et al. selected only one of the two solutions presented by Raitzsch & Hönisch, and specifically the one that shows greater discrepancy from their data. The reason why Raitzsch & Hönisch presented two estimates is because:

(a) the d11B vs pH sensitivity of all marine carbonates calibrated over a wide pH range (>0.5 units) shows a sensitivity that is less than that predicted by the aqueous fractionation factor of Klochko et al. 2006, and (b) because models and experimental data suggest that boron isotope fractionation is affected by temperature (Zeebe 2005, Rustad et al. 2010, Dissard et al. 2012, Kaczmarek et al. 2015, Liu et al. 2015). These two factors were placed on hold by Rae et al. 2011 because they could not evaluate them within their 0.3 pH unit coretop calibration, and Rae et al admitted that their "closeness to inorganic theory may be fortuitous". We know now that the original arguments of Foster (2008) and Rae et al. (2011), that all marine carbonates follow a pH sensitivity similar to aqueous boron isotope fractionation when measured by MC-ICP-MS, is false (e.g. Henehan et al. 2013, Krief et al, 2010), and evidence for a temperature effect under otherwise constant environmental conditions is accumulating (see above). Raitzsch & Hönisch 2013 therefore applied two sensitivities, one similar to empirical carbonate calibrations established over a wide pH range, and the other using the aqueous fractionation after Klochko et al. 2006. Only the latter is shown by Greenop et al., but the first, which is more and more confirmed by newer experimental data as described above, has been omitted. Remarkably, the omitted record matches the estimates of Greenop et al. much better than the one they chose to present in this manuscript. I would urge the authors to include both, and provide appropriate arguments for each of them.

In our modified figure 10 we now show both scenarios from Raitzsch and Hönisch (2013). However, we disagree with a number of points raised above. Specifically, our reading of the literature suggests that whilst the aqueous fractionation factor for boron is well-known (Klochko et al. 2006, Nir et al., 2015) there is little consensus regarding the universality or not of the $\delta^{11}$B response of marine carbonates to changing pH. Furthermore, there is no experimental evidence supporting a *significant* effect of temperature on the aqueous fractionation factor (Klochko et al., 2006). Indeed, a recently published inorganic calcite precipitation study has shown no measureable effect (at the ± 0.3 ‰ level) of temperature on the fractionation factor when

growth rate is controlled (within the temperature range investigated 12 °C to 32 °C; Kaczmarek et al. 2016). This finding is entirely in agreement with the conclusions of Rae et al. (2011), Henehan et al. (2013), Martinez-Boti et al. (2015), Henehan et al. (in review) based on field calibrations. As noted by the reviewer, Dissard et al. (2012) do find that the $\delta^{11}B$ in *Acropora* coral cultured at 22 to 28 °C is temperature dependent. However, the observed dependency (~0.15 permil per °C) is within uncertainty of the expected relationship given the known dependence of pKB on temperature (~0.13 permil per °C) and hence is not evidence of a significant temperature influence on the isotopic fractionation factor. It is of course likely that the fractionation factor is temperature dependent, current evidence however suggests it is smaller than analytical uncertainty. In a comprehensive study of the theoretical determination of the boron isotopic fractionation factor Zeebe (2005) states that:

"Given the range of outcome for αB3–B4 [fractionation factor] at 300 K calculated in the current paper, no recommendation will be made regarding α's temperature dependence, which equally depends on the frequencies/methods chosen."

So we feel that currently the safest option, and one that is supported by the field calibration datasets (Rae et al., 2011; Henehan et al., 2013; Martinez-Boti et al., 2015; Henehan et al. in review) is to not apply a temperature correction. Nonetheless, for completeness we will show both curves from R&H (2013) whilst also reiterating these points in the revised manuscript.

There are certainly shortcomings in the approach applied by Raitzsch & Hönisch, as there are with all indirect d11Bsw estimates, but looking for synchrony between different approaches, and learning from differences would be powerful way to improve the paleoceanographic community's confidence in this valuable proxy; insisting only on differences will do the opposite.

.

The agreement between the two approaches is indeed encouraging.  In the revised manuscript we endeavour to find the common ground between the studies whilst also highlighting the uncertainties (including those in our own study).

Finally, the estimates of Pearson & Palmer (1999, 2000) are dismissed based on their N-TIMS analytical procedure and the use of the boron isotope fractionation factor after Kakihana and Kotaka (1977). I disagree with both of these arguments: Foster et al. (2013) have shown that N-TIMS and MC-ICP-MS report different numbers in absolute terms, but the relative differences between samples are similar, such that they will yield the the same amplitude of d11B and pH change. The comparison of benthic d11B data measured by N-TIMS by Raitzsch & Hönisch (2013) and Greenop in Figure 5 confirms this notion and argues against the analytical procedure creating a systematic bias. Foster and his colleagues have discredited N-TIMS analyses long enough with this unfounded argument, and it should finally be put to rest. Similarly, the d11B vs. pH sensitivity implied by the Kakihana factor actually matches the sensitivity of empirical carbonates very well, both using N-TIMS (e.g. Sanyal et al. 1996,2001, Hönisch et al. 2004) and MC-ICP-MS (e.g., Krief et al. 2010, Henehan et al. 2013). So even if the Kakihana fractionation factor incorrectly describes boron isotope fractionation in seawater, it accidentally describes marine carbonates, and in particular planktic foraminifers very well. There may be analytical issues with the specific N-TIMS method applied by Palmer, but we will not be able to evaluate this because that particular method is no longer applied in any labs reconstructing pH from marine carbonates. The only reasonable and significant argument that can and must be applied for the data of Pearson and Palmer (1999, 2000) is that they did not know about vital effect offsets between different foraminifer species at that time, and consequently applied the same calibration curve for a variety of them. Given what we know about vital effects on $\delta^{11}B$ today, that is a fundamental

shortcoming of those studies, and I agree that they should be considered with caution on this basis.

While the interlaboratory comparison of Foster et al., (2013) did suggest relative variation in $\delta^{11}$B are recorded by different analytical methods, the study also found some indication that for measurements by NTMIS the deviation from the interlaboratory mean increases with decreasing B/Ca ratio. This is important for the study of Pearson & Palmer (1999, 2000) where the whole foraminiferal assemblage is used, with a range of B/Ca, as the gradient between different species may in part be influenced by these analytical issues. While the lack of offset between the benthics from Raitzsch and Honisch (2013) and this study further confirms that the offset between MC-ICPMS and NTMIS is smaller in foraminifera with higher B/Ca, this good agreement is in contrast to that observed by Rae et al. (2011) who documented a ~1.2 permil difference between *C. wuellerstorfi* measured by MC-ICPMS and NTIMS (by Honisch et al. 2008). We see no offsets between our dataset and Rae et al. (2011) and so the NTIMS vs. MC-ICPMS offsets in absolute sense are clearly not that well understood. Similarly, at ODP 668 in the Equatorial Atlantic, we see a 2.6 permil difference between similar sized Holocene *G. sacculifer* (comparing data in Foster, 2008 and Honisch and Hemming, 2005), but the Atlantic benthic data in Raitzsch and Honisch (2013) are in good agreement between methods. It is for this reason that we are cautious about over interpreting the agreement between our estimates of $\delta^{11}$Bsw and those of Pearson and Palmer (1999, 2000). In addition, there are also uncertainties associated with the difficultly of assigning a depth habitat to individual foraminiferal species and species-dependent isotope effects (as mentioned above) Pagani et al., (2005). However, given the comments of this reviewer, in the revised manuscript we do include the Pearson and Palmer (1999,2000) estimates in Figure 10 and use them to support the emerging view of the evolution of $\delta^{11}$Bsw (albeit caveated with the points above).

Site locations: A map should be provided with the locations of the studied core sites, including a justification why these sites lend themselves to this particular study. Their water depths should be mentioned, and the preindustrial d13C and pH gradient for each core site should be added to Figure 4c, so the reader can evaluate how well these sites fit the fundamental premise of this study, i.e. that these sites can be used to reconstruct the d13C and pH gradients over the past 23 million years. It is particularly noteworthy here that sites 758 and 761 are from the Indian Ocean and therefore differ in their hydrography from Atlantic sites 926 and 999. Importantly, this difference is somewhat dismissed in the later part of the study, where the slope specific to Atlantic sites is selected and applied to estimate the pH gradients for all sites. A map will be added to the revised manuscript. These sites were initially chosen as they have been shown to faithfully reconstruct surface water pH in previous studies (Foster et al., 2008; Foster et al., 2012; Martinez-Boti et al., 2015). ODP Site 758 was used in Pleistocene to extend coverage outside of the Atlantic Ocean. In the supplement we present the data using both the "Atlantic only" and the "low latitude" gradient. In the revised manuscript we will plot the data from both these scenarios in the main part of the manuscript in order to more fully explore the data.

Boron isotope reconstructions: I have been looking for the original d11B data measured for each sample, but cannot find them in tabular form and Fig. 5 only displays them in the form of d11B borate, i.e. already one step removed from the original data. It is important to list and display the original data and their uncertainties. Have different foraminifer species been measured on the same samples, to cross-calibrate ancient species versus modern G. ruber? This is an important aspect for evaluating the appropriateness of the applied d11B vs d11b borate calibrations.
We are changing figure to show planktic $\delta^{11}$B foraminifera in the revised manuscript. No $\delta^{11}$B-pH calibration is applied to the benthic $\delta^{11}$B so this is the original data in agreement with Rae et al. (2011). For the majority of the

samples we use either the species-specific *G. ruber* or *G. sacculifer* calibration (from 0 to 22 Ma). In the case of samples at the Oligo-Miocene boundary the absence of either of these species means we are unable to do this. Here we use *G. praebulloides* making inferences about its life habit from d18O and d13C gradient with size fraction (Pearson and Wade, 2009). We currently do not have a sample where *G. praebulloides and G. sacculifer* co-exist and this is an area that needs to be further researched in the future. We hope to display this information clearer in the revised manuscript.

Modeling the d13C vs pH relationships: It would be nice to provide the exact parameters and their ranges applied for the model estimates in a table, so any reader could repeat these experiments without having to contact the authors. Also, references should be provided for all estimates, and please reconsider some of the wording. For instance, it is unclear whether atmospheric pCO2 was applied once at 200 ppm and once at 1000ppm, or across the entire range (Line 368). Please list studied ranges for pH, CaCO3 saturation states and the biological pump in tabular form.

We add this table to the revised manuscript.

The slopes of the d13C- pH relationships (Fig. 7) need to be better explained. I take from the text that the global estimate includes all data, then data are systematically restricted first to the low latitudes, and then to the low latitude Atlantic only. That should make the data density sparser from step to step, but while this seems somewhat correct for the dense cloud between 1.7-3 permil d13C, the low-latitude and Atlantic-only subsets (b and c) display data that are not included in Fig. 7a. Please explain where these data are coming from, and why they are not included in the complete data set (7a)? To allow the reader to better evaluate the consequence of this restriction, the data in Fig. 4c should be colored to highlight global low latitude and low latitude Atlantic data, and the respective regressions should be shown. Furthermore, it needs to be

evaluated how the choice of slope affects the pH gradient estimates for the Indian Ocean cores. Specifically, the two 758 samples at around 2 Ma appear to have a large effect on the Pleistocene d11Bsw estimates, in particular after the smoothing function is applied. It seems like the Indian Ocean sites should be estimated using either the global low latitude slope (Fig. 7b) or a slope specific to the Indian Ocean, but not the one for the low latitude Atlantic.

In Figure 7 there is one estimate of the slope per individual ensemble member simulation (i.e. the regression for that simulation). In the "global" case we regress all 18 model boxes, in the "low latitude" case we regress only the low latitude surface/deep boxes (4x surface and 4x deep; 8 boxes) and in the "Atlantic-only" calculate the slope from only the low latitude Atlantic surface and deep box (2 boxes). That is, each of the panels has the same number of estimates for the slope of the relationship, but they are evaluated from the model ensemble in different ways. All experiments are consistent between the three panels of Figure 7. To prevent any confusion we have modified the relevant text and expanded the caption of the figure.

Smoothing the d11Bsw estimates: Fig. 9 demonstrates that irrespective of the pH gradient chosen, the data uncertainty encompasses the entire record, and only application of the smoothing function allows to discern a d11Bsw trend. Greenop et al. argue that the smoothing function is justified because Lemarchand et al. 2002 calculated the d11Bsw rate of change is ~0.1 permil per million years (line 398 in this manuscript). However, Lemarchand et al. also estimated up to 0.6‰ d11Bsw change per million years for the Cretaceous, suggesting that the 0.1 permil/Myr change is not the limit and depends on the model parameterization. Remarkably, Greenop et al. eventually argue that their record shows more variability than the model estimates of Lemarchand et al., and suggest that the model relevant boron inputs and outputs are not fully understood (Line 532/533). This means that Greenop et al want it both ways – they use the small rate of change to justify their hefty smoothing, but then cast doubt on Lemarchand's estimates when they show less variation than these new estimates. The smoothing obviously

cannot be justified with Lemarchand's estimates, and what-ever argument is brought up must be independent of an alternative d11Bsw estimate, or the argument becomes circular. Regardless, given that the smoothing function reduces the potential d11Bsw ranges estimated from this approach from 42-32 permil to ~40-37 permil implies that application of the smoothing function largely dismisses the actual data, and therefore the heart of the reconstruction. This data treatment may be appropriate for a dataset with hundreds of data, but with just 20 data from not even a handful of sediment cores I am having a hard time finding the smoothed estimate convincing.

In order to justify our use of smoothing we refer to the study of Lemarchand et al., (2000). Here we determine an approximate rate of change for $\delta^{11}$Bsw from the inputs and outputs of boron into and out of the ocean relative to the boron concentration. This is therefore a fundamental constraint on the behaviour of the boron isotope system in the oceans given what we know about the modern inputs and outputs. Where we have concerns about the record of Lemarchand et al., (2000), is in regards to the extent to which changes in the magnitude of these inputs/outputs through time are understood. Many of the processes that determine the $\delta^{11}$Bsw are poorly constrained through time (e.g. crustal permeability, lifetime of water–rock interactions, and expansion rate of the oceanic ridge; Simon et al., 2006). We make minor changes to the text to clarify this distinction.

This reviewer also suggests that by smoothing is somehow dismisses the actual data. In the revised manuscript we will detail why smoothing is both needed (since the uncertainties in each $\delta^{11}$Bsw reconstruction is large due to propagation of uncertainties in all parameters involved) and justified. The latter is simply because our $\delta^{11}$Bsw data exhibit a larger variability than is possible given the amount of boron in the oceans and the likely inputs and outputs. As noted by reviewer 2, given the long residence time of boron in the oceans it should have a smooth evolution and one that, given what we know about the boron cycle currently, should not result in large changes (i.e. < 0.1 permil per Ma). In order to further support our decision to smooth the

estimates we explore a number of other scenarios (e.g. binning the data, using an algorithm to smooth, and assume a spline fit) and then focus the subsequent discussion on what aspects of the evolution of $\delta^{11}$Bsw are consistent across these scenarios and hence robust to the nature of our chosen smoothing (e.g. that $\delta^{11}$Bsw is around xx permil lighter than modern in the middle Miocene, and much of the change in ratio occurred during the interval xx to xx Ma).

Similarly, Greenop et al. use the variability observed in their d13C and pH gradient estimates to argue the variable pH gradient estimates give a better result than the constant pH gradient approach. With error bars that essentially overlap for all samples, and two estimates randomly discarded, I am rather uncomfortable believing any arguments for preferring one estimate over the other. Given the large uncertainty of this approach, I would recommend Greenop et al. plot all their estimates, and do not bury half of them in the supplement. This entire range of possible d11Bsw estimates should be shown in Fig. 10, and not a subjective subset. Ideally, Figure 10 would present all available estimates not only from this study, but also from others (Raitzsch & Hönisch 2013, Lemarchand et al. 2000); and then focus the discussion on synergies between different data sets, find a consensus d11Bsw trend, and discuss whether the synergies can help us to further improve our understanding of the boron isotope proxy.

Again, we want to thank the referee for the thoughtful comments along these lines, which we agree with and used to guide manuscript revision throughout. In the revised manuscript we will include a figure of all our $\delta^{11}$Bsw records using the full range of different assumptions compared with published data and now put emphasis on how a collective view of the available $\delta^{11}$Bsw records can aid our understanding.

Comparison to other isotope systems: This is a nice comparison and it looks convincing after the true data variability has been discarded, but it would be nice to expand this analysis and include all possible d11Bsw scenarios in it, including the actual data before smoothing. Suggesting weathering and catchment patterns from correlating the smoothed record to actual data seems premature given the significant uncertainty of the original (unsmoothed) d11Bsw estimates.

Clearly the database for this analysis is not fully satisfying at this point, both for boron isotopes and the other isotope systems. We will make it clearer in the revised manuscript that these interpretations are tentative and subject to further work, as the seawater records of all the isotope systems are improved. We take the position that the apparent correspondence among the isotopic change of the different elements is interesting and thought provoking, in particular with respect to future modelling work to better understand the evolution of $\delta^{11}$Bsw, even if it is still tentative.

Line 28: sp: consistent

Line 46: sp: Martínez-Botí

Line 54: please add Lemarchand et al. 2000. They were the first to note the change in d11Bsw and compare it to the analytical uncertainty of d11B

Line 66: please add a reference for the d11B of B inputs Line 68: please specify whether mineral precipitation or rainfall is meant by "precipitation"

Line 76: please replace Zeebe & Wolf-Gladrow by Hemming & Hanson 1992, the $CO_2$ book only reports the earlier suggestion Line 83: please add Lemarchand et al. 2000

Line 90/91: please rephrase "currently...at present"

Line 115/116 please rephrase: a small subset of the CO2 data used by Ridgwell et al. 2005 is derived using the boron isotope-pH proxy, leading to some circularity in the d11Bsw method. However, the pH estimates applied by

Raitzsch & Hönisch (2013) from Tyrrell & Zeebe (2004) are based on GEOCARB, and the circularity problem does not apply.

Line 216: Why is it important to specify "at 0.7 Ma" here? Please explain or rephrase

All of the above typos and revised sentencing will be carried out in the revised manuscript. We would argue, however, that the suggested rephrasing of Line 115/116 is slightly misleading. Despite the GEOCARB pH estimates not containing any estimates derived using the boron isotope pH proxy, the $\delta^{11}$Bsw record is going to be heavily dependent on the surface water pH input. Consequently, if this $\delta^{11}$Bsw record is going to be subsequently used in pH calculations, the output pH will strongly reflect the GEOCARB input.

Line 220: please specify that you assume there is no temperature effect on d11B. Ideally you should calculate d11B borate with and without the temperature effect, and estimate the associated uncertainty on the d11Bsw estimate

For the reasons described above we have elected not to apply a temperature correction to the calculated $\delta^{11}$B.

Line 228/252: please specify whether trace element and d13C analyses were performed on the same foraminifer species and size classes as studied for d11B

All trace element data was analysed on an aliquot of the $\delta^{11}$Bsw samples. In almost all cases d13C was also analysed on the same foraminifera. Where this was not possible another surface dweller/benthic foraminifera was used. All d13C will be made available on Pangaea on successful publication of the manuscript.

Line 231: please define "H"

This will be added to the revised manuscript.

Line 249-251/281-286: please specify the influence of temperature on the d13C-DIC estimate

We have not corrected the d13C-DIC estimates for temperature and will acknowledge this in the revised manuscript. Culture work by Bemis et al., 1999 suggests that in the symbiotic foraminifera *O. universa* the d13C is insensitive to temperature under low light levels and the d13C of high light shells decreased slightly with temperature (0.05 per mil/$^{o}$C). Consequently, we expect changes in temperature to have a minimal effect on the d13C-DIC estimates.

Line 259: which benthic species was analyzed for d13C?
*C. wuellestorfi* or *C. mundulus* was analysed for d13C

Lines 261-265: symbiotic planktic foraminifers record heavier d13C than surrounding seawater, so the vital effect adjustment for estimating d13C-DIC should be negative, not positive

This section will be rephrased as "We use a carbon isotope vital effect for *G. ruber* (+0.94 ‰; Spero et al., 2003), *T. sacculifer/G. praebulloides* (+0.46 ‰; Spero et al., 2003; Al-Rousan et al., 2004;), *C. mundulus* (+0.47 ‰; McCorkle et al., 1997) and *C. wuellestorfi* (+0.1 ‰; McCorkle et al., 1997) to calculate dissolved inorganic carbon (DIC).

Line 269: sp: outline
Line 273: leads to a broad
Lines 295/296: please rephrase, e.g. "CaCO3 dissolution increases CO32- and pH". The sentence is difficult to read. Line 302: it would be nice to anticipate at this point which of the described effects will be considered in the remainder of the paper
Line 304: please define the Suess effect and provide a reference
Line 306: please add the uncertainty of the slope
All of the above typos and revised sentencing will be carried out in the revised manuscript.

Line 399: please analyze and explain possible reasons for the small d11B differences between the planktic and benthic forams excluded here. Is there any indication/reason why these data are anomalous?

It is possible that preservation is not so good at these intervals in the core and the planktic foraminifera are affected by partial dissolution (Seki et al., 2010). Alternatively, localised circulation changes at the core site may be affecting the benthic $\delta^{11}B$ signal.

Line 416/417: please rephrase to include that the benthic d11B record is a function of both pH and d11Bsw
Will update in the revised manuscript.

Fig. 4 caption: Please explain the difference between open and closed symbols, and different symbol shapes in 4c. It would be advisable to replot this figures according to recommendations made above, instead of just copying it from Foster et al. 2012.
The figure will be replotted and the caption made clearer.

Line 1007: sp: Suess. To the best of my knowledge Dr. Seuss never wrote about carbon isotopes.
Line 1015: please explain what is meant by "paired measurements"
Mg/Ca measurements are conducted on an aliquot of the sample measured for $\delta^{11}B$.

Line 1026: sp: colours reflect Line 1033: sp: tests Line 1049: insert period after "details)"
This typo will be removed.

Fig. 2: Please remove the two lower end Miocene d11Bsw estimates of Paris et al., which were identified as outliers by the original paper. Including them makes the record look worse than it is, which seems unfair.

Following Paris et al., (2010) we will put these data points in brackets and explain their exclusion in the figure caption.

Fig. 3: It would be nice to insert a horizontal bar or arrow to indicate the smaller pH gradient in b.

This can be added to the figure.

Fig. 5: Please add a second time axis at the top of the figure for better comparison throughout the figure. Horizontal lines in each panel might also help to estimate values at each time interval. Please also indicate the planktic foram species measured for each data point, e.g. by different color symbols. Are there any data pairs that allow cross-calibration of praebulloides and trilobus?

The 2$^{nd}$ time axis and horizontal line will be added. Colour coding of species can also be applied. Unfortunately, there are no data pairs that allow cross-calibration of *G. praebulloides* and *G. trilobus* but this is a good avenue for study in the future.

Fig. 8: please indicate core site symbols in this figure. In particular, the large gradient at ~ 2 Ma is due to two samples from one site in the Indian Ocean. It is unfortunate to select two samples that are close in time from one site and then no more for the rest of the record. Those samples have the potential to bias the record at that time.

Core site samples can be added and, as noted above, we will fully explore the influence of these two samples on our smoothed record;

Fig. 10: please add both curves from Raitzsch & Hönisch, including their uncertainty estimates

We will add both curves from Raitzsch & Honisch, while explaining the caveats associated with including the record calculated with an empirical fractionation factor.

Fig. 12: at a minimum, I would suggest to remove the Indian Ocean cores from this estimate. I suspect the data will match much better without those estimates.
Here we elect to keep in the data from the Indian Ocean, however, we will put a greater emphasis in the manuscript on the record calculated using the global low latitude d13C-pH relationship and explore this further.

---

## Author Comment (AC2) · 18 May 2016

RESPONSE TO REFEREE #2

This work aims at proposing a new approach to determine the B isotopic composition of the oceans over the past 25 Ma. This is one of the three pillars of the atmospheric pCO2 reconstruction over geological timescales from B isotopes in marin carbonates. The two other pillars are the pH dependence of the B isotopic fractionation and the change of seawater alkalinity. Even if the few d11Bsw reconstructions published so far all point to a slow increase during the last tens of Ma, they are all based on models and assumptions that are sometimes difficult to ascertain. In that, any effort to provide new and independent approach is the most welcome for the scientific community. Since it is definitely a tricky task, any effort of developing new approaches has therefore to be acknowledged.

We thank the referee for the positive view on our new approach to reconstruct $\delta^{11}$Bsw, and we also appreciate the constructive criticism on the way that we describe and discuss prior results. In response to the concerns raised here and by the other referee we have modified the tone of the manuscript with regard to prior work and we emphasize the progress towards a consensus on past seawater boron isotope change that is supported by different approaches, including our own.

The past d11Bsw modeled in this study is based on new B isotopic data on paired planktic and benthic foraminifera. Two scenarios have been tested to derive d11Bsw: the first one assumes a constant pH gradient along the seawater depth-profile and the second one uses d13C data to model possible variation of the seawater pH depth-profile. Even claimed to be a new approach by the authors, using paired foraminifera living at different depths in the water column was first introduced by Pearson and Palmer (1999).

In the revised manuscript we will make it clear the approach we are using is an extension of that first introduced by Pearson and Palmer (1999). The

novelty of our approach, however, is in (a) combining measurements on planktic and deep ocean benthic foraminifera, and (b) correcting for change in the surface/bottom water pH gradient using d13C (as mentioned below).

The originality of the present work is the attempt to anchor this approach with other data (d13C) and climate model. Finally, the present work ends with a model of evolution of the marine d11Bsw very close and consistent with the previous work, which is already fine, but the large scatter of data (Fig 9) makes possible any d11Bsw value between +42‰ and +32‰. After data smoothing (which statistical meaning being questionable from a so small database), the authors end with a narrow range of possible values which spreading approaches the uncertainty of previous approaches (see fig. 10). This is then a bit disappointing and this is could be easily corrected if the paper starts with a more upfront position and objective of their work in comparison to previous ones. This starts in the abstract: one of the most discussed issue of using B isotopes in carbonates to reconstruct past seawater pH is first the actual relationship between the B isotopic fractionation and the pH. So far, many works have provided "calibration curve" using different foraminifera species, and they all come with differences attributed to "biological effect". Another point is the B speciation in carbonates and the actual process of B incorporation. I personally think that most of the published d11B sw are actually consistent and what we need now is a more accurate model that will narrow the range of possible values. The present work also needs to acknowledge that the approach based on paired foraminifera living at different depths in the water column was first proposed by Pearson and Palmer in 1999! Later, sensitivity tests carried by Pagani et al. (2004) showed that this approach is rather inaccurate.

In the revised manuscript we will make it clear the approach we are using is an extension of that first introduced by Pearson and Palmer (1999). The inaccuracies pointed out in Pagani et al., (2005) mainly refer to the difficult of assigning a depth habitat to individual foraminiferal species, species-dependent isotope effects, the analytical uncertainty of the carbonate. In our

study, by focusing on the surface to deep gradient we avoid the difficulty of defining calcification depths for multiple species. The analytical uncertainty and species-specific isotope effect are both better understood and accounted for in this work.

In Fig. 1 is presented a simplified B cycle in the ocean but, presented as it is, there is a large imbalance between inputs and outputs: the B inputs by precipitation is one order of magnitude higher that the sum of all the others. Then, either this inputs flux is balanced by output by seasalts or evaporation of gaseous B and then this loop is that fast that it must be taken into account, or atmospheric B derives from a continental source and a huge output flux is missing. One possibility is that the atmospheric B content (certainly in gaseous form and not seasalts) derives from anthropic activities, which are irrelevant for reconstructions on geological timescales. Whatever, the explanation is, the B marine cycle like presented in fig. 1 at least useless if not wrong. I would therefore strongly suggest to revise this figure and the corresponding text, in particular lines 66-71 where the atmospheric fluxes are discussed.

We acknowledge that our understanding of the modern boron cycle requires further work and the view we present is overly simplified. In the revised figure and text we will refer to what is presented here as the fluxes of B that are important on geological timescales following Lemarchand et al. (2002). We will also emphasis in the text, caption and figure itself that the view we present includes a large imbalance of inputs and outputs that will need to be addressed in future work.

There is one thing that we can taken for sure is the very long B residence time in the oceans. This is held by the very high B concentration is seawater (about 500 times more concentrated that river waters). Therefore, whatever the model or indirect determination of the d11Bsw, large and rapid variation of d11Bsw like those mentioned line 86 are unrealistic as long as we do not identify a huge B flux, missed so far, that may have affected the B cycle over

geological time scales.

We fully agree with this comment, and indeed this the fundamental rational behind our approach of smoothing. However, in the context of this comment, while changes mentioned in the study of Simon et al., (2006) (line 86) are highly unlikely, these authors illustrate the potential uncertainty involved in modeling the boron cycle with our current understanding of changes in oceanic crust alteration through time.

A map of the sampling location would be appreciated

We add a map to the revised manuscript.

In equation (1), I think that epsilon(biological effect) would be appropriate and add in the text a discussion about the different calibration curves

A description of how the calibration curves were constructed will be added to the text.

section 2.2 is a succession of hypothesis and calculation made from a series of embedded models, which is certainly the best way to make estimation with our present knowledge, but a discussion of the possible errors propagated is critical here in a much more detailed and argued way than proposed in section 2.5. In particular, from what data is derived the estimation of ±0.05 pH of the error made on the d13C-pH relationship (line 381)?

Also in response to the other referee's comments, we add a data table and additional description on this point to the updated manuscript. The uncertainty of the d13C-pH relationship is estimated from the model sensitivities and ensemble simulation spread. The uncertainty in d11B is described further down the section (lines 385-392). The ±2$^o$C uncertainty in temperature is a reflection of the uncertainty in the Mg/Ca measurement and the relationship between Mg/Ca$_{foraminifera}$ and temperature. No record of salinity exists through time. Consequently, we apply a ±2 psu unit uncertainty which we think we be sufficient to cover any potential variations. The uncertainty in [Mg] and [Ca] are derived from Horita et al., (2002).

It is difficult to follow the section 3.2. The important information is finally given in the last 5 lines of this section. Please shorten and clarify.

This section will be reworded and clarified in the revised manuscript.

This is a bit frustrating to see d11Bsw values and discussion of them only in the last 2 pages (lines 484-537). I would strongly suggest to shorten some of the previous parts and strengthen this last one.

While we acknowledge further discussion of the $\delta^{11}$Bsw values would certainly be of interest to the boron isotope community, we hesitate to expand this section as reviewer 1 suggests it is premature to do so. However, in the revised manuscript we endeavor to cut down the sections outlining our methodology and justification of approach.

One important point is the validation of the model and the statistical treatment made on the models d11Bsw values. At first sight, fig. 9. looks far from being convincing: a large scatter of the data, which looks like not providing strong new contraints and a significant restriction of the possible data by smoothing the small dataset?

As outlined by the reviewer, boron has a long residence time in the ocean and therefore rapid variations in $\delta^{11}$Bsw are unrealistic. We use this to justify our smoothing and provide an extra constraint on our record. This is necessary because our uncertainties in each $\delta^{11}$Bsw reconstruction are large as a consequence of our approach where we fully propagating the uncertainties in all parameters involved. In order to strengthen the justification for our approach in the revised manuscript we will explore a number of other scenarios (e.g. binning the data, using an algorithm to smooth, and assume a spline fit) and then focus the subsequent discussion on what aspects of the evolution of d11Bsw are consistent across these scenarios and hence robust to the nature of our chosen smoothing (e.g. that d11Bsw is around xx permil

lighter than modern in the middle Miocene, and much of the change in ratio occurred during the interval xx to xx Ma).

---

## Author Response (AR1)

We would like to take this opportunity to thank the editor and all reviewers for their thoughtful and constructive comments, from which our manuscript has greatly benefited. For the most part the changes that were made in response to the reviewer's comments are outline in the individual responses to reviewer's comments. All minor typos, rewording and changes to the figures have been made. Outlined below is a list of the major changes we have made to the manuscript and an instance where we found it was not possible to fulfill the request of the reviewer during revision of the manuscript.

**Major issues addressed:**

**1) Presentation of other datasets**

We have made considerable revisions to the introduction. Here, we now present the pre-existing $\delta^{11}B_{sw}$ records, focusing on the similarity between records. The discussion of these various records has been extended later in the manuscript in comparison to our new $\delta^{11}B_{sw.}$ As with the introduction, we follow the referee's suggestion and the revised discussion section now clearly emphasises the similarity between the various records, building towards consensus. Only in the case of specific discrepancies between the records do we explore the differences in methodology between the $\delta^{11}B_{sw}$ reconstructions as a way to explain any of the differences.

**2) More detailed discussion of uncertainties and modeling parameters.**

The uncertainties used in the Monte Carlo simulation and the variables changed in the modeling studies have now been tabulated in order to improve the clarity in these sections of the manuscript. Included in the table of uncertainties is also a justification of the $2\sigma$ that we apply.

**3) Simplifying the $\delta^{11}B_{sw}$ output and applying the most appropriate $\delta^{13}C/pH$ relationship**

Reviewer 1 suggested that given the overlap in uncertainties between the different records we presented, it wasn't appropriate to recommend the use of one of our $\delta^{11}B_{sw}$ record over another. This valuable comment led us to re-think the presentation of our central argument with regard to the uncertainty of $\Delta pH/\Delta\delta^{13}C$ relationship and thereby, we think, both simplified and improved our study. We now present a single record that captures a broader range of different $\delta^{13}C/pH$ relationships than any of our initial scenarios. In practical terms, this is done by applying a flat probability (ie. Equal rather than normally distributed) of ±0.05 to the $\Delta$pH estimate using the central $\Delta$pH/$\Delta\delta^{13}$C slope of 0.175/‰ diagnosed from our extensive sensitivity tests using both the CYCLOPS and GENIE models. This nominal uncertainty is equivalent to the broad range of $\Delta$pH/$\Delta\delta^{13}$C slopes of 0.14/‰ and 0.21/‰ – covering the vast majority of our model simulations and removing the need to present separate "slope scenarios". That is, because we have used a flat probability there is an equal likelihood of any value between about 0.14/‰ and 0.21/‰. To avoid confusion and given the evidence from our $\delta^{13}$C data and modeling work we have also now discarded the hypothetical scenario where the pH gradient was assumed to have remained the same as modern. The second motivation for following this broad approach is that it is not possible to test the $\delta^{13}$C/pH relationship at our specific sites as requested by reviewer 1, nor for the low latitudes. Currently pre-industrial surface water $\delta^{13}$C data is only available for the North Atlantic >20$^{o}$N (Olsen and Ninnemann, 2010). When a wider dataset of pre-industrial water column $\delta^{13}$C is available in the future, it will be possible to refine our $\delta^{11}$B$_{sw}$ record. We now make this point explicitly.

**4) Exploration of the smoothing parameter and the impact on the record**

In order to test the dependence of the output record on the smoother we have undertaken a binning exercise where we have averaged our data over 8 Myr intervals. The calculated mean and two standard errors of the data in each interval show that the difference between the middle Miocene $\delta^{11}$B$_{sw}$ and modern is significant. The presence of the rise in $\delta^{11}$B$_{sw}$ across this interval in a number of other published records suggests that our record adds to the growing consensus on the evolution of $\delta^{11}$B$_{sw}$ in the Neogene.

[revised manuscript text omitted]

which can be converted to $\delta^{11}B_{sw}$ based on

**Page 4: [9] Deleted**            **r.greenop**            **18/07/2016 15:00**

While this approach yields a qualitative independent check on other approaches (e.g. halite inclusions, geochemical modeling), as a quantitative record of $\delta^{11}B_{sw}$ through time, it has a number of drawbacks. Firstly, some of the $CO_2$ data used in the modeling studiesis derived using the boron isotope-pH proxy, leading to some circularity in the methodology.Secondly, given the structure in $CO_2$ proxy records, the assumption that surface ocean pH changed linearly through the Cenozoic is most likely an oversimplification (Beerling and Royer, 2011). Consequently, while this method may shed some light on the evolution of $\delta^{11}B_{sw}$ through time, it cannot be subsequently used to determine pH or atmospheric $CO_2$ from $\delta^{11}B$ of foraminiferal calcite because the $\delta^{11}B_{sw}$ record is itself based on assumptions of the secular evolution of pH and $CO_2$.

One of the big challenges of reconstructing a

**Page 4: [10] Deleted**            **r.greenop**            **11/08/2016 14:59**

record empirically is determining $\delta^{11}B_{sw}$

**Page 4: [11] Deleted**            **r.greenop**            **11/08/2016 15:00**

[revised manuscript text omitted]

Figure 1

[Figure]

Figure 2

[Figure]

Figure 3

[Figure]

Figure 4

[Figure]

Figure 5

Figure 6

[Figure]

[Figure]

Figure 7

[Figure]

Figure 8

[Figure]

Figure 9

[Figure]

Figure 10

Figure 11

[Figure]

Figure 12

[Figure]

[Figure]

Figure 13

**Table 1.** CYCLOPS model parameter values defining the ensemble of 13,500 simulations*

| Parameter | Description | Values assumed |
|---|---|---|
| PAZ surface phosphate** | unutilized polar nutrient | 1µM, 1.25µM, 1.5µM, 1.75µM, 2µM |
| PAZ vertical exchange** | bottom water formation | 2Sv, 7.75Sv, 13.5Sv, 19.25Sv, 25Sv |
| SAZ surface phosphate** | unutilized polar nutrient | 0.7µM, 0.825µM, 0.95µM, 1.075µM, 1.2µM |
| AMOC circulation scheme** | deep vs. shallow overturning | NADW, GNAIW |
| representative timeslice*** | Age ([$Ca^{2+}$]/CCD); calcium set outright; CCD set via riverine $CaCO_3$ flux using inverse scheme | 0Myr (10.6mM, 4.65km), 9Myr (12.89mM, 4.4km), 11Myr (13.33mM, 4.9km), 16Myr (14.28mM, 4.7km), 18Myr (14.57mM, 4.25km), 20Myr (14.86mM, 4.7km) |
| atm. $CO_2$**** | set via silicate weatherability | 200ppm, 300ppm, 400ppm, 500ppm, 600ppm, 700ppm, 800ppm, 900ppm, 1000ppm |

*= The six parameters assume 5, 5, 5, 2, 9 and 6 values, yielding 13,500 distinct parameter combinations

** = These parameters are intended to span the full range of ocean carbon cycling over late Pleistocene glacial-interglacial cycles, as describe in more detail in Hain et al. (2010)

*** = We selected representative timeslices based on local extrema in the CCD reconstruction of Pälike et al. (2012) and we combine these with appropriate reconstructed calcium concentrations based on Horita et al. (2002). These choices are intended to capture the range of long-term steady state conditions of the open system $CaCO_3$ cycle relevant to our study interval

**** = These atmospheric $CO_2$ levels are chosen to span a range wider than expected for the study interval. Following silicate-weathering-feedback paradigm, long-term $CO_2$ is fully determined by the balance of geologic $CO_2$ sources and silicate weathering, whereby faster acting processes of the open system $CaCO_3$ cycle compensate relative to that $CO_2$ level. All else equal, high $CO_2$ levels, low calcium concentrations and deep CCD correspond to high bulk ocean carbon concentrations (Hain et al., 2015) with many of the individual simulations of this ensemble exeeding 4000µM DIC.

Table 2

| Input parameter | Uncertainty applied | Source of uncertainty estimate |
|---|---|---|
| Surface to sea floor $\Delta$pH | Uniform +/- 0.05 pH units | Plausible range of $\Delta$pH/$\Delta\delta^{13}$C in CYCLOPS and GENIE sensitivity tests; prediction error of linear $\Delta$pH/$\Delta\delta^{13}$C regression in GENIE |
| $\delta^{11}$B measurement | 0.15-0.61‰ | Long-term external reproducibility |
| Temperature | ±2°C | Uncertainty in the Mg/Ca measurement and Mg/Ca-temperature calibration |
| Salinity | ±2 psu | In the absence of a salinity proxy this uncertainty is applied to cover variations through time. |
| Seawater [Mg] | ± 4.5 mmol/kg | following Horita et al., (2002) |
| Seawater [Ca] | ± 4.5 mmol/kg | following Horita et al., (2002) |

**Table 3**

| Sources | Isotopic Ratio | | | |
|---|---|---|---|---|
| **Oceanic Inputs** | $\delta^{11}B_{sw}$ 39.61 ‰ | $\delta^{7}Li_{sw}$ 31 ‰ | $\delta^{26}Mg_{sw}$ −0.83 ‰ | $\delta^{44/40}Ca_{sw}$ 0 ‰ |
| Input from hydrothermal | 6.5[a] | 8.3[b] | N/A | −0.96[h] |
| Fluid from accretionary prisms | 25[a] | 15[b] | N/A | N/A |
| Riverine Inputs | 10[a] | 23[b] | −1.09[d] | -1.28[h] |
| Groundwater | N/A | N/A | -0.82[d] | −1.02[i] |
| **Outputs** | | | | |
| Precipitation into carbonates | 20[a] | 29[c] | -3.5[d,e,f] | -1.15[h,j] |
| Ocean crust alteration | 4[a] | 15[b] | -0.83[d,g] | -1.2[h] |
| Absorption onto sediment | 15[a] | 15[b] | ?? | N/A |

---

## Author Response (AR2)

**Report #1**

Greenop et al. have revised their manuscript substantially, improved the model tests and tailored it to show the consensus with previously published studies, rather than the focus on minor discrepancies of the original manuscript. I find this manuscript very much improved, but it still needs a few minor but important revisions. For the sake of simplicity, I will list these in chronological order, which is not necessarily the order of importance.
We thank the referee for the positive view on our revised manuscript and the improvements outlined below.

Line 175: spelling of name: Martínez-Botí
This has been changed.

Lines 192/238/248…: wuellerstorfi
This has been changed

Lines 205/208: Müller
This has been changed.

Lines 274 ff: what makes these "minor effects"? In particular the Miocene is a period of intense organic matter deposition (Monterey Formation!), that indicates systematic and substantial removal of isotopically light carbon from the ocean. Systematic changes in boundary conditions such as d13C-DIC and pCO2 are not captured in the assumed constant uncertainty, which accounts only for random changes in boundary conditions. As such, these new d11Bsw estimates are still not the ultimate answer, which is generally nicely acknowledged in this manuscript, but should be repeated in the conclusions (see comment below).
In the section here we refer to dissolution of calcium carbonate shells as a minor effect on the timescales we are concerned with as the CCD record of Palike et al. (2012) suggests there has not been substantial changes through time in this parameter. While the carbon isotopic composition of the whole ocean may have change, this was most likely accompanied by a pH change, and based on the modeling we present, the d13C-pH relationship has not changed significantly within the uncertainty bounds we use. We will reiterate the drawbacks to our methodology in the conclusion section as suggested below.

Line 460: its changes

This has been changed.

Lines 474/475: what is the "plausible" range and why? Please define "plausible".
See response to reviewer 2. This has now been expanded on at the end of the section (lines 370-374).

Lines 584-587: the pH decrease may be at odds with the pCO2 estimates shown in Beerling & Royer (2011), but more estimates have since been published that support the greater pCO2 reconstructed from stomata (as already shown in Beerling & Royer). For instance, Zhang et al. (2013) show Miocene pCO2 from alkenones >400 µatm, Bolton et al. (2016) went a step further and considering calcification changes in coccolithophores, they suggest Miocene pCO2 may have been at least 50% higher compared to the Plio-Pleistocene. Foster et al. (2012) and Greenop et al. (2014) also suggest Miocene pCO2 ~ 400 µatm, although they used an exceedingly low alkalinity of 1292 µmol/kg to yield these estimates, a value that is even more extreme than the alkalinity inventories tested in Fig. 7 (~1800-2600, considering that modern alk is somewhere around 2200 µmol/kg), which are considered "extreme and inconsistent with geological evidence" (Line 1138). Using more reasonable alkalinity values of ~2000 µmol/kg (Ridgwell 2005, Tyrrell & Zeebe 2004), the Miocene boron isotope pCO2 estimates range closer to 500-600 µatm, clearly consistent with global evidence for warmer temperatures, and acidification as implied by models. There is uncertainty in all of these pCO2 estimates but growing evidence for higher Miocene pCO2 should be acknowledged and the argument that "proxy CO2 and surface water pH estimates are not well described by the linear change in pH" should be revised in the light of this growing proxy data evidence.
We agree with the reviewer that there is now good evidence for high $CO_2$ during the middle Miocene climatic optimum. We would question, however, the evidence that the early Miocene (17-23 Ma) has higher $CO_2$ than the Miocene climatic optimum (15-17 Ma; as would be suggested by a linear change in pH). There is currently a lack of $CO_2$ data for the early Miocene and consequently we will change the wording of the sentence to "proxy $CO_2$ and surface water pH estimates may not be well described by the linear change in pH" to reflect this uncertainty.

Line 704: please augment this sentence to "Despite some disagreements, and different uncertainties associated with each approach, the fact that…".
This is just to reiterate that Greenop's approach, like all others, has large uncertainties as well, albeit due to different factors.
This has been done.

Line 712: controls
This has been done.

Line 1109: please specify where the "modern ocean" data are coming from,
i.e. they are globally distributed and exceed the data shown in the Atlantic
profiles. The reference to Foster et al. (2012) is not sufficient.
The data are from all the ocean basins spanning the latitudes of 40N to 40S.
This detail has been added to the figure caption.

Line 1117: Please clarify that the error bars with 95% confidence due to
external reproducibility only apply to the new data by Greenop et al.
(this study), but not to Raitzch & Hönisch (2013), where they represent
propagated uncertainties of external reproducibilies of time equivalent
benthic foraminifer samples from different core sites in different ocean
basins.
This has been done.

Line 1122: the color choice for the planktic foraminifera symbols is
unfortunate and should be changed - the orange and red colors are nearly
indistinguishable.
This has been changed.

Figure 8 caption: Please explain the 0.201 line
This has been done.

Figures 2 and 11: The d11Bsw data of Raitzsch & Hönisch (2013) are not
plotted correctly, the data are all lower than originally presented, and so are
the polynomial fits. The polynomial fits should probably be replaced by
something like a 5-point running mean anyway, but I am puzzled why the
data are lower than presented in the original study? This is particularly
striking in Fig. 11, where the data are shifted even lower than in Fig. 2,
below the 39.61‰ modern seawater d11Bsw estimate that the caption claims
the data have been adjusted to (Lines 1175/1176). This must be rectified.
The original publication presented the data originating at a modern d11Bsw
average of 39.6‰, so no adjustment of the original is necessary or justified.
The polynomial fits have now been replaced by a 5-pt running average. The data were erroneously adjusted to 39.61‰ based on a single value for modern d11Bsw rather than an average of the two data points at 0 Ma in the Raitzsch & Hönisch (2013) dataset. We thank the reviewer for pointing this out and now no adjustment has now been made the d11Bsw data. There was also an error in the original dataset provided by the author of the study.

**Report #2 Submitted on 18 Nov 2016**
This study presents a new boron isotope curve of the Neogene based on foraminifera. The authors make the point that the d11B of forams is not an unfractionated archive for seawater, but is instead pH dependent, and therefore must be corrected using modelled ocean pH. The study goes into some depth on this modelling, and the assumptions made, and the results broadly agree with several other studies, although this study has considerably more data. Interestingly, the d11B seawater data also co-vary with records from other isotopic systems like Li, Mg and ca.
Overall this is a well-written manuscript, with a detailed discussion on modelling, and some interesting outcomes in terms of seawater records, and I recommend it be published after some minor amendments listed below.
We thank the referee for taking the time to review our manuscript and the insightful comments outlined below.

Line 61: isn't carbonate weathering also a B source? Presumably they must be if carbonates are a major sink.
Carbonate weathering is also a source of boron to the ocean, however, in the river systems currently studied (Rose et al., 2000; Lemarchand and Gaillardet 2006) the weathering of silicate rocks dominates the riverine signal. The text has been updated to reflect this.

Line 63: is this low or high temperature ocean crust alteration?
Low temperature alteration. This has been updated in the text.

Line 116/Fig 3c: I would draw this as a line, rather than a series of data points, to make reading the graph easier.
This has been done.

Lines 11-125: for the non-B person, explain how this is not affected by CCD changes and associated pH changes?

This section refers to the d13C-pH gradient between the surface and thermocline depth so won't be affected by CCD change. The surface-to-deep gradient may be affected and this is something that is explored in the CYCLOPS modeling. When the CCD depth is changed within the bounds set out by Palike et al. (2012) we find the d13C-pH relationship does not vary outside the assigned uncertainty.

Line 145: referencing a manuscript in preparation is a little odd.
Instead of referring to a manuscript in prep, we will describe the paper as "a follow up study by *Sosdian et al.,*".

Line 147: "nano" rather than "nanno"
This has been corrected.

Lines 154-161: what secondary standards were measured to ensure precision/accuracy? What is the precision of the d11B measurements?
The secondary standard used to ensure precision/accuracy is the Japanese Geological Survey *Porites* coral standard JCP. The details of this standard and the precision of the d11B measurements are outlined in section 2.5: Assessing uncertainty.

Section 2.4 and 2.5: you've listed all the parameters you used, but it would also be useful if you explained why you selected those parameters for the models.
These parameters were used to exceed any plausible changes within the Cenozoic. This is stated on lines 331-332, and is now, for improved clarity, reiterated at the end of section 2.4.

Line 513: explain this 0.1‰/Myr a bit more, especially given rapid Neogene weathering (and therefore input) changes.
The rate at which the $\delta^{11}B_{sw}$ can change is limited by the size of the oceanic boron reservoir compared the magnitude of the inputs and outputs. Therefore, even in the case of a large change in the weathering input, the residence time of boron in the ocean will exert a strong control on the rate at which $\delta^{11}B_{sw}$ can change. This has now been emphasized in the text.

Line 634: you say there are two controls on Mg, but then list three…
This has been changed.

Line 674-680: if you want to be inclusive about all the Li hypotheses, there are also modelling papers on the Cenozoic: Li and West 2014, Wanner et al., 2014, Vigier and Godderis, 2015.

These references have been added to the appropriate sections. Thank you for this comment.

Line 671-698: this could be considered a bit oversimplified, given controls by carbonate and dolomite formation on Ca and Mg isotopes, and in turn their link with enhanced cation supply through weathering, and also potential temperature controls on dolomite formation. Also, Li is only affected by silicate weathering, whereas Ca and Mg are not – and Sr isotopes tell us that the Himalayas are dominantly (metamorphosed) carbonates. However, I agree that the cross-plots are compelling – but I think probably worth backing off a bit from the statements that it's only rivers, given, for example, that it's very hard to actually model Li by jut invoking rivers (see Li and West 2014).

The final statement of this paragraph has been changed to be more inclusive of other factors that could control the Neogene major ion composition.

It's also interesting that there is an order of magnitude difference in residence time of the different elements – perhaps something worth mentioning and discussing. Why, for example, does the correlation between Li and B seem best, given the widest residence time difference?

This is a very interesting point. Here (and in the text) we hypothesize that the correlation between B and Li is best because the processes controlling the fluxes into and out of the ocean are similar. A more sophisticated approach would no doubt be necessary to fully unpick the trends we highlight here, but this is beyond the scope of this current manuscript but is something we will be following up on in due course.

In terms of the Ca record, I would take a look at Fantle and Tipper 2014 – especially their compilation of Neogene Ca isotopes (and their corrigendum to that graph), which suggests that the Griffith data are offset from others.

Many thanks for pointing us towards the Fantle and Tipper corrigendium. Here we chose to use the marine barite record of Griffith et al., 2008 as it is the archive that is the most likely to be a passive trace of Ca isotopes (Fantle, 2010). However, we acknowledge that the extent to which this is the case is still unknown and will make the distinction that the calcium isotopic composition of seawater is based on the marine barites.

P. Pogge von Strandmann, UCL, UK

[revised manuscript text omitted]

r.greenop 4/12/2016 10:58

[Figure]

Figure 1

Figure 2

[Figure]

[Figure]

Figure 3

[Figure]

Figure 4

[Figure]

Figure 5

Figure 6

[Figure]

[Figure]

Figure 7

[Figure]

Figure 8

[Figure]

Figure 9

[Figure]

Figure 10

[Figure]

Figure 11

Figure 12

[Figure]

[Figure]

Figure 13

**Table 1.** CYCLOPS model parameter values defining the ensemble of 13,500 simulations*

| Parameter | Description | Values assumed |
|---|---|---|
| PAZ surface phosphate** | unutilized polar nutrient | 1μM, 1.25μM, 1.5μM, 1.75μM, 2μM |
| PAZ vertical exchange** | bottom water formation | 2Sv, 7.75Sv, 13.5Sv, 19.25Sv, 25Sv |
| SAZ surface phosphate** | unutilized polar nutrient | 0.7μM, 0.825μM, 0.95μM, 1.075μM, 1.2μM |
| AMOC circulation scheme** | deep vs. shallow overturning | NADW, GNAIW |
| representative timeslice*** | Age ($[Ca^{2+}]$/CCD); calcium set outright; CCD set via riverine $CaCO_3$ flux using inverse scheme | 0Myr (10.6mM, 4.65km), 9Myr (12.89mM, 4.4km), 11Myr (13.33mM, 4.9km), 16Myr (14.28mM, 4.7km), 18Myr (14.57mM, 4.25km), 20Myr (14.86mM, 4.7km) |
| atm. $CO_2$**** | set via silicate weatherability | 200ppm, 300ppm, 400ppm, 500ppm, 600ppm, 700ppm, 800ppm, 900ppm, 1000ppm |

*= The six parameters assume 5, 5, 5, 2, 9 and 6 values, yielding 13,500 distinct parameter combinations

** = These parameters are intended to span the full range of ocean carbon cycling over late Pleistocence glacial-interglacial cycles, as describe in more detail in Hain et al. (2010)

*** = We selected representative timeslices based on local extrema in the CCD reconstruction of Pälike et al. (2012) and we combine these with appropriate reconstructed calcium concentrations based on Horita et al. (2002). These choices are intended to capture the range of long-term steady state conditions of the open system $CaCO_3$ cycle relevant to our study interval

**** = These atmospheric $CO_2$ levels are chosen to span a range wider than expected for the study interval. Following silicate-weathering-feedback paradigm, long-term $CO_2$ is fully determined by the balance of geologic $CO_2$ sources and silicate weathering, whereby faster acting processes of the open system $CaCO_3$ cycle compensate relative to that $CO_2$ level. All else equal, high $CO_2$ levels, low calcium concentrations and deep CCD correspond to high bulk ocean carbon concentrations (Hain et al., 2015) with many of the individual simulations of this ensemble exeeding 4000μM DIC.

Table 2

| Input parameter | Uncertainty applied | Source of uncertainty estimate |
| --- | --- | --- |
| Surface to sea floor $\Delta$pH | Uniform +/- 0.05 pH units | Plausible range of $\Delta$pH/$\Delta\delta^{13}$C in CYCLOPS and GENIE sensitivity tests; prediction error of linear $\Delta$pH/$\Delta\delta^{13}$C regression in GENIE |
| $\delta^{11}$B measurement | 0.15-0.61‰ | Long-term external reproducibility |
| Temperature | ±2$^{\circ}$C | Uncertainty in the Mg/Ca measurement and Mg/Ca-temperature calibration |
| Salinity | ±2 psu | In the absence of a salinity proxy this uncertainty is applied to cover variations through time. |
| Seawater [Mg] | ± 4.5 mmol/kg | following Horita et al., (2002) |
| Seawater [Ca] | ± 4.5 mmol/kg | following Horita et al., (2002) |

**Table 3**

| Sources | Isotopic Ratio | | | |
|---|---|---|---|---|
| **Oceanic Inputs** | $\delta^{11}B_{sw}$ 39.61 ‰ | $\delta^7Li_{sw}$ 31 ‰ | $\delta^{26}Mg_{sw}$ −0.83 ‰ | $\delta^{44/40}Ca_{sw}$ 0 ‰ |
| Input from hydrothermal | 6.5[a] | 8.3[b] | N/A | −0.96[h] |
| Fluid from accretionary prisms | 25[a] | 15[b] | N/A | N/A |
| Riverine Inputs | 10[a] | 23[b] | −1.09[d] | -1.28[h] |
| Groundwater | N/A | N/A | -0.82[d] | −1.02[i] |
| **Outputs** | | | | |
| Precipitation into carbonates | 20[a] | 29[c] | -3.5[d,e,f] | -1.15[h,j] |
| Ocean crust alteration | 4[a] | 15[b] | -0.83[d,g] | -1.2[h] |
| Absorption onto sediment | 15[a] | 15[b] | ?? | N/A |

---

## Author Response (AR3)

**Response to Editor's comments:**

We would like to thank the Editor for these improvements to the manuscript.

Line 17 - 21. Use Boron isotope composition or Boron isotopic composition (both fine with me but please use consistently; if a chemist would mind which one you use, please use the most correct one)

We have used boron isotope composition consistently through this section, and use a combination through the manuscript depending on context.

Abstract might be phrased more directly. Here a proposal:

The boron isotopic composition (d11B) of foraminiferal calcite reflects the pH and the boron isotopic composition of the seawater the foraminifer grew in. For pH reconstructions, the d11B of seawater must therefore be known but information on this parameter is limited. Here we reconstruct Neogene seawater d11B based on the d11B difference between paired measurements of planktic and benthic foraminifera and an estimate of the coeval water column pH gradient from their d13C values. Carbon cycle model simulations underscore that the DpH/Dd13C relationship is relatively insensitive to ocean and carbon cycle changes, validating our approach. Our reconstructions suggest that d11Bsw was ~37.5 ‰ during the early and middle Miocene (roughly 23 - 12 Ma) and rapidly increased during the late Miocene (between 12 and 5 Ma) towards the modern value of 39.61 ‰. Strikingly, this pattern is similar to the evolution of the seawater isotopic composition of Mg, Li and Ca, suggesting a common forcing mechanism. Based on the observed direction of change, we hypothesize that an increase in secondary mineral formation during continental weathering affected the isotopic composition of riverine input to the ocean since XXX Ma.

We thank the Editor for taking the time to edit the abstract, and we are happy to accept this more succinct version.

Please make sure you indicate one actual age for the XXX. As it stands now, it seems like you are referring to a time interval rather than one point in time with an uncertainty in the age.

**This has been changed.**

91. new paragraph starting at "Geochemical modelling"
109. new paragraph starting at "The fourth approach"
173. new paragraph starting at "Although"
356. new paragraph starting at "The ensemble spans"
the first paragraph of 3.2 might be split in at least 2 to optimize readability

These additional paragraphs have been added.

559-561. sentence unnecessary could be deleted. This sentence has been deleted.

In Conclusions, make sure to name both the epochs and absolute ages according to Gradstein et al for both the lower and upper part of the studied interval. Both the epochs and the upper and lower bounds of the time intervals have now been added to the conclusions.

- A record of Neogene seawater  $\delta^{11}$ B reconstructed from paired  $\delta^{11}$ B 1
- analyses on benthic and planktic foraminifera. 2
- 3
- Greenop Rosanna1,2\*, Hain, Mathis P.1, Sosdian, Sindia M.3, Oliver, Kevin I.C.1, 4
- Goodwin, Philip1, Chalk, Thomas B.1,4, Lear, Caroline H.3, Wilson, Paul A.1, Foster, 5 Gavin L.1, 6
- 7 \*Corresponding author
- 8 1Ocean and Earth Science, National Oceanography Centre Southampton, University
- 9 of Southampton, Waterfront Campus, European Way, Southampton SO14 3ZH, UK
- 10 2School of Geography & Geosciences, Irvine Building, University of St Andrews,
- North Street, St Andrews, KY16 9AL, UK 11
- 3School of Earth & Ocean Sciences, Cardiff University, Cardiff, CF10 3AT, UK 12
- 13 4 Department of Physical Oceanography, Woods Hole Oceanographic Institution,
- 14 Woods Hole, Massachusetts, USA
- 15

**16 Abstract:**

- The boron isotope composition ( $\delta^{11}$ B) of foraminiferal calcite reflects the pH and the 17 18
- boron isotope composition of the seawater the foraminifer grew in. For pH
- reconstructions, the  $\delta^{11}$ B of seawater must therefore be known but information on this 19
- 20 parameter is limited. Here we reconstruct Neogene seawater  $\delta^{11}$ B based on the  $\delta^{11}$ B
- 21 difference between paired measurements of planktic and benthic foraminifera and an
- 22 estimate of the coeval water column pH gradient from their  $\delta^{13}$ C values. Carbon
- cycle model simulations underscore that the  $\Delta p H / \Delta \delta^{13} C$  relationship is relatively 23
- insensitive to ocean and carbon cycle changes, validating our approach. Our 24
- reconstructions suggest that  $\delta^{11}B_{sw}$  was ~37.5 % during the early and middle 25
- Miocene (roughly 23 12 Ma) and rapidly increased during the late Miocene 26
- 27 (between 12 and 5 Ma) towards the modern value of 39.61 ‰. Strikingly, this pattern
- 28 is similar to the evolution of the seawater isotope composition of Mg, Li and Ca,

[revised manuscript text omitted]

- 1292 definition.
- 1293